# Understanding Scaling Laws with Statistical and Approximation Theory for Transformer Neural Networks on Intrinsically Low-dimensional Data

**Alex Havrilla**
Department of Mathematics
Georgia Institute of Technology
Atlanta, GA, 30308
ahavrilla3@gatech.edu

**Wenjing Liao**
Department of Mathematics
Georgia Institute of Technology
Atlanta, GA, 30308
wliao60@gatech.edu

## Abstract

When training deep neural networks, a model's generalization error is often observed to follow a power scaling law dependent both on the model size and the data size. Perhaps the best known example of such scaling laws are for transformer-based large language models (**LLMs**), where networks with billions of parameters are trained on trillions of tokens of text. Yet, despite sustained widespread interest, a rigorous understanding of why transformer scaling laws exist is still missing. To answer this question, we establish novel statistical estimation and mathematical approximation theories for transformers when the input data are concentrated on a low-dimensional manifold. Our theory predicts a power law between the generalization error and both the training data size and the network size for transformers, where the power depends on the intrinsic dimension $d$ of the training data. Notably, the constructed model architecture is shallow, requiring only logarithmic depth in $d$. By leveraging low-dimensional data structures under a manifold hypothesis, we are able to explain transformer scaling laws in a way which respects the data geometry. Moreover, we test our theory with empirical observation by training LLMs on natural language datasets. We find the observed empirical scaling laws closely agree with our theoretical predictions. Taken together, these results rigorously show the intrinsic dimension of data to be a crucial quantity affecting transformer scaling laws in both theory and practice.

## 1 Introduction

Deep learning has made remarkable breakthroughs in various real-world applications, such as natural language processing [Graves et al., 2013, Bahdanau et al., 2014, Liu et al., 2023, Vaswani et al., 2017], computer vision [Krizhevsky et al., 2012, Goodfellow et al., 2014, Song et al., 2020], healthcare [Miotto et al., 2018], and robotics [Gu et al., 2017]. A neural scaling law between the generalization error (or test loss) and several quantities, including the model size, the training data size, and the amount of compute, plays a key role in the performance of neural networks. Perhaps the best known example of such scaling laws are for transformer-based LLMs. Recent works in Hestness et al. [2017], Rosenfeld et al. [2019], Kaplan et al. [2020], Bahri et al. [2021] demonstrated a power law between the test loss and the network size, the training data size, and the amount of compute for transformer-based LLMs. Yet, despite sustained widespread interest, a rigorous understanding of why *transformer scaling laws* exist is still missing.

Understanding the theory behind neural scaling laws provides invaluable insights into practical applications of deep learning. A mathematical principal of neural scaling laws enables researchers and practitioners to describe and analyze the performance of neural networks with precision and

rigor. The neural scaling law between the generalization error and the network size can be partially explained via neural network representation theory [Yarotsky, 2016]. Further, the neural scaling law between the generalization error and the training data size $n$ can be explained via statistical estimation theory. For feedforward neural networks [Schmidt-Hieber, 2020] and convolutional residual networks [Oono and Suzuki, 2019], a generalization error bound has been established for regression. Schmidt-Hieber [2020], Oono and Suzuki [2019] predicted Generalization Error $\sim n^{-c/D}$ where $n$ is the training data size, $D$ is the data dimension and $c$ is a constant. This predicted rate of convergence is extremely slow for high dimensional data when $D$ is large, while the rate of convergence observed in real-world applications is significantly faster, which reveals a gap between theory and practice.

This gap can be bridged by exploiting low-dimensional structures of data. Real-world data sets often exhibit low-dimensional geometric structures due to rich local regularities, global symmetries, or repetitive patterns [Tenenbaum et al., 2000, Roweis and Saul, 2000]. According to Min et al. [2023, Figure 1], the intrinsic dimension of CIFAR-100, CelebA and ImageNet datasets are about $20, 20$ and $40$ respectively. When the low-dimensional geometric structure of data is modeled by a manifold, the predicted scaling for regression, classification and distribution estimation becomes Generalization Error $\sim n^{-c/d}$, where $n$ is the training data size, $d$ is the intrinsic dimension of the data manifold, and $c$ is a constant [Chen et al., 2022, Liu et al., 2021, Dahal et al., 2022, Nakada and Imaizumi, 2020]. In Sharma and Kaplan [2022], the neural scaling law between the test loss and the network size was predicted to be Test loss $\sim (size)^{-4/d}$ where $d$ is the intrinsic dimension of data. While the theoretical studies focus on feedforward neural networks [Chen et al., 2022, Nakada and Imaizumi, 2020] and convolutional residual networks [Liu et al., 2021], a generalization to transformer-based neural networks [Vaswani et al., 2017] is of great interest but widely open.

This paper establishes mathematical approximation and statistical estimation theories to predict and justify the scaling law between the generalization error and the model/data size for transformer neural networks. We consider regression of a $\beta$-*Hölder continuous function* $f : \mathcal{M} \to \mathbb{R}$ where $\mathcal{M}$ is a $d$-dimensional compact Riemannian manifold isometrically embedded in $\mathbb{R}^D$. After embedding the input $x \in \mathcal{M} \subset \mathbb{R}^D$ to a proper sequence, we apply a transformer network on the embedded sequence to learn the function $f$. Our main results are on the statistical estimation and universal approximation theories of Hölder continuous functions on $\mathcal{M}$ by transformer neural networks.

**Statistical Theory:** In Theorem 1, we consider the global empirical risk minimizer $\hat{\mathrm{T}}_n$ from $n$ i.i.d. training data $\{(x_i, f(x_i))\}_{i=1}^n$, given by

$$\hat{\mathrm{T}}_n = \arg\min_{\mathrm{T} \in \mathcal{T}} \frac{1}{n} \sum_{i=1}^n \left( \mathrm{T}(x_i) - f(x_i) \right)^2, \tag{1}$$

under a properly chosen transformer network architecture $\mathcal{T}$. We prove that, the generalization error of $\hat{\mathrm{T}}_n$ satisfies

$$\mathbb{E}\|\hat{\mathrm{T}}_n - f\|_{L^2(Q)}^2 \leq \tilde{O}\big(Dd^2 n^{-\frac{2\beta}{2\beta+d}}\big) \tag{2}$$

where $Q$ denotes the distribution of $x$, and $\tilde{O}$ hides constants and $\log n$ terms.

**Approximation Theory:** In Theorem 2, we construct a transformer network to universally approximate $\beta$-Hölder continuous functions on $\mathcal{M}$ with an arbitrarily given accuracy $\varepsilon$. Notably, the network is shallow, requiring only $O\big(\log(d)\big)$ independent of the desired accuracy $\epsilon$ to approximate $f$ locally. This highlights a major advantage of Transformers over feed-forward ReLU networks, which require $O\big(\log(\frac{1}{\epsilon})\big)$ layers to achieve the same accuracy.

In our proof, we embed the entries of $x = [x^1, \ldots, x^D] \in \mathcal{M}$ into tokens such that the $x^i$'s appear in a sequence. Our proof for the approximation theory explicitly constructs transformers to realize the interaction between different tokens efficiently via a crucial *Interaction Lemma* 3. This lemma allows us to flexibly implement many common operations including addition, multiplication, and parallelization, and so may of independent interest. In our proof for the statistical theory, we calculate the covering number of our transformer network class, which is also of independent interest.

**Neural Scaling Laws and the Intrinsic Dimension:** Our generalization error bound in (2) predicts the following neural scaling law between the generalization error and the data size $n$:

$$\text{Squared Generalization Error} := \mathbb{E}\|\hat{\mathrm{T}}_n - f\|_{L^2(Q)}^2 \lesssim n^{-\alpha_D}, \quad \text{where } \alpha_D = \frac{2\beta}{2\beta+d}, \tag{3}$$

with sufficient data. Our approximation theory in Theorem 2 predicts the following neural scaling law between the approximation error and the network size $N$:

$$\text{Squared Approximation Error} := \inf_{\mathrm{T} \in \mathcal{T}} \|\mathrm{T} - f\|_{L^{\infty}(\mathcal{M})}^2 \lesssim N^{-\alpha_N}, \text{ where } \alpha_N = \tfrac{2\beta}{d} \quad (4)$$

for a sufficiently large network class $\mathcal{T}$. Our prediction of the power scaling law is consistent with our own empirical observations, and those in Kaplan et al. [2020] and Biderman et al. [2023]. More importantly, our theory quantifies the power $\alpha_D, \alpha_N$ in terms of the intrinsic dimension of data.

**Experimental Validation on LLMs:** After establishing our theory we seek to validate it in practice by predicting empirical scaling laws for LLMs trained on natural language data. To test our predictions for the data scaling law, we pretrain a series of small (125 million parameter) LLMs on three datasets [Gokaslan et al., 2019, Eldan and Li, 2023, Kocetkov et al., 2022]. We find close agreement ($\pm 0.02$) between our predicted scaling exponent $\alpha_D$ and the observed exponents $\hat{\alpha}_D$. To evaluate our predictions for the model scaling exponent $\alpha_N$, we rely on publicly available scaling suites [Biderman et al., 2023, Radford et al., 2019] whose intrinsic data dimensions we can estimate. We find our predictions are still close but less accurate for $\alpha_N$. Finally, we carry out a series of ablations investigating factors impacting the estimated intrinsic data dimension $d$. For a fixed dataset, we find the estimated $d$ is stable with respect to several factors including the model size, model embedding dimension, and context length[1].

In summary, we make the following contributions:

- A novel approximation theory for transformers approximating Hölder continuous functions on a $d$-dimensional manifold, requiring $O(\log(d))$ depth independent of the accuracy $\epsilon$.

- A novel computation of the covering number of our transformer network class. This is used to establish generalization bounds exponentially depending on the intrinsic dimension $d$.

- Empirical experiments demonstrating our theory predicts data scaling laws for LLMs as a function of the estimated intrinsic data dimension $d$.

- An empirical study of several factors affecting the estimated intrinsic data dimension for transformers including model size, embedding dimension, layer depth, and context length.

We will present our main theory in Section 2, numerical validation of our theory and the prediction of neural scaling laws in Section 3. We will discuss related work in Section 4 and conclude our paper in Section 5. Our pre-training hyperparameters are given in Appendix A. The derivation of neural scaling laws is presented in Appendix B. Our notation is given in Appendix C, and proofs are presented in Appendix E and F.

## 2 Transformer Generalization and Approximation Theory

This paper establishes statistical estimation and mathematical approximation theory of transformers for the regression of Hölder functions on a low-dimensional manifold. We start by defining transformer neural networks.

### 2.1 Transformer Neural Networks

**Definition 1** (Transformer Neural Network)**.** *We define a transformer neural network* $\mathrm{T}$ *as a composition of functions of the form*

$$\mathrm{T}(x) = \mathrm{D} \circ \mathrm{B}_{L_T} \circ ... \circ \mathrm{B}_1 \circ (\mathrm{PE} + \mathrm{E}(x)) \quad (5)$$

*which is parameterized by*

- $L_T$*: The number of transformer blocks* $\mathrm{B}_i$ *in* $\mathrm{T}$.

- $m$*: The maximum number of attention heads per transformer block.*

- $L_{\text{FFN}}$*: The max depth of the feed-forward layers per block.*

- $w_{\text{FFN}}$*: The max width of the feed-forward layers per block.*

---

[1]Code is available at `https://github.com/Dahoas/transformer_manifolds_learning`

- $d_{embd}$: *The token embedding dimension.*

- *D: The input dimension.*

- *l: The number of hidden tokens.*

- $\kappa$: *A bound on the magnitude of network parameters.*

*We define each component of the composition as*

- $x \in \mathbb{R}^D$ *is the input.*

- *A linear embedding layer* $E : \mathbb{R}^D \to \mathbb{R}^{d_{embd} \times l}$. *In this work we will always take* $E = E' \circ U$ *where* $U \in \mathbb{R}^{l \times D}$ *and* $E' \in \mathbb{R}^{d_{embd} \times 1}$ *applied columnwise is fixed. We call embedded output* $H = E(x)$ *the first **embedding matrix** whose columns are referred to as **tokens**.*

- $PE \in \mathbb{R}^{d_{embd} \times l}$ *is a fixed matrix implementing the transformer **positional encoding**.*

- *Transformer blocks* $B_i : \mathbb{R}^{d \times l} \to \mathbb{R}^{d \times l}$ *for* $i \in \{1, ..., L_T\}$ *which are residual compositions of multi-headed attention (**MHA**) layers* MHA *and feed-forward layers* FFN *acting token-wise.*

- *A decoding layer* $D : \mathbb{R}^{d_{embd} \times l} \to \mathbb{R}$ *which is fixed to outputing the first element of the last column.*

*We use the ReLU activation function* $\sigma(x) = \max(0, x)$ *in the network.*

For a complete definition of the components of the transformer neural networks, we refer to Appendix D. The transformer network T may sometimes be written $T_\theta$ which makes explicit the dependence on learnable weights $\theta$. We can also define a class of transformer neural networks $\mathcal{T}$ of interest.

**Definition 2** (Transformer Network Class). *We define a class of transformer networks as*

$$\mathcal{T}(L_T, L_{\text{FFN}}, w_{\text{FFN}}, l, d_{embd}, m, R, \kappa) = \Big\{ T_\theta \mid T_\theta \text{ is in (5) with at most } m \text{ attention heads in each block,}$$

$$L_{\text{FFN}} \text{ layer feed-forward networks with hidden width } w_{\text{FFN}},$$

$$d_{embd} \text{ token dimension, } l \text{ hidden tokens },$$

$$\text{and have } \|T_\theta\|_{L^\infty(\mathbb{R}^D)} \leq R, \|\theta\|_\infty \leq \kappa \Big\}$$

*where* $\|T_\theta\|_{L^\infty(\mathbb{R}^D)} \leq R$ *bounds the output of* T *and* $\|\theta\|_\infty$ *bounds the weight magnitude of* $T_\theta$.

## 2.2 Assumptions

We consider a manifold $\mathcal{M}$ and the target function $f : \mathcal{M} \to \mathbb{R}$, with the following assumptions:

**Assumption 1** (Manifold). *Let* $\mathcal{M}$ *be a compact Riemannian manifold with intrinsic dimension* $d$ *isometrically embedded in* $\mathbb{R}^D$. *Because* $\mathcal{M}$ *is compact, there exists* $M > 0$ *such that* $\|x\|_\infty \leq M$ *for* $x \in \mathcal{M}$. *Additionally, we assume* $\mathcal{M}$ *has positive **reach** $\tau > 0$.*

**Assumption 2** (Target function). *The target function* $f : \mathcal{M} \to \mathbb{R}$ *is* $\beta$-*Hölder continuous on* $\mathcal{M}$, *for some* $0 < \beta \leq 1$ *and Hölder constant* $H_f > 0$, *and in addition* $\|f\|_{L^\infty(\mathcal{M})} \leq R$ *for some* $R > 0$.

In Assumption 1, the reach [Federer, 1959, Aamari et al., 2019] $\tau$ of $\mathcal{M}$ can be defined as

$$\tau = \inf\{r > 0 : \exists x \neq y \in \mathcal{M}, v \in \mathbb{R}^D \text{ such that } r = \|x - v\| = \|y - v\| = \inf_{z \in \mathcal{M}} \|z - v\|\}.$$

Informally, reach is the smallest distance at which a projection onto the manifold is no longer unique. In practice this can be used to establish a bound on the number of charts covering the manifold.

## 2.3 Transformer Generalization Theory

Given $n$ training samples $\{(x_i, f(x_i))\}_{i=1}^n$ where $\{x_i\}_{i=1}^n$ are i.i.d. samples of a distribution $Q$ supported on $\mathcal{M}$, we aim to *learn* an approximation $\hat{T}_n$ to $f$ by minimizing the empirical risk (1) over

a class of a transformer neural networks $\mathcal{T}(L_T, L_{\text{FFN}}, w_{\text{FFN}}, l, d_{embd}, m, R, \kappa)$. The corresponding generalization error is given by

$$\mathbb{E}\|\hat{\mathrm{T}}_n - f\|_{L^2(Q)} = \mathbb{E}\sqrt{\int_{\mathcal{M}} \left(\hat{\mathrm{T}}_n(x) - f(x)\right)^2 dQ(x)}. \tag{6}$$

If $\mathcal{M}$ and $f$ satisfy Assumptions 1 and 2, we prove the following generalization error bound.

**Theorem 1.** *Let $M, \tau, R, H_f > 0$, $0 < \beta \leq 1$, $d, D \in \mathbb{N}$, $\mathcal{M}$ and $f$ satisfy Assumption 1 and 2 respectively. Given $n$ training samples $\{(x_i, f(x_i))\}_{i=1}^n$ where $\{x_i\}_{i=1}^n$ are i.i.d. samples of a distribution $Q$ supported on $\mathcal{M}$, if we use the transformer neural network class $\mathcal{T}(L_T, L_{\text{FFN}}, w_{\text{FFN}}, l, d_{embd}, m, R, \kappa)$ with parameters*

$$L_T = O\big(\log(d)\big), \quad L_{\text{FFN}} = O\big(\log(n)\big), \quad w_{\text{FFN}} = O(1), \quad l = O\big(dn^{\frac{d}{2\beta+d}}\big)$$

$$d_{embd} = O(1), \quad m = O\big(dn^{\frac{d}{2\beta+d}}\big), \quad \kappa = O\big(d^2 n^{\frac{2d}{2\beta+d}}\big)$$

*in the empirical risk minimization* (1)*, where $O(\,\cdot\,)$ hides terms in $C_{\mathcal{M}}$ (the number of charts), $D, H_f, M$, then the empirical risk minimizer $\hat{\mathrm{T}}_n$ given by* (1) *satisfies*

$$\mathbb{E}\int_{\mathcal{M}} \left(\hat{\mathrm{T}}_n(x) - f(x)\right)^2 dQ \leq \tilde{O}\big(Dd^2 n^{\frac{-2\beta}{2\beta+d}}\big)$$

*where $\tilde{O}$ hides logarithmic terms in $n, d$ and linear terms in $C_{\mathcal{M}}$.*

Theorem 1 is proved in Appendix F, via a bias-variance decomposition. The bias represents the approximation error of $f$ by transformer neural networks, and the variance represents the stochastic error in the parameter estimation of transformer neural networks. To quantify the bias, we explicitly construct a transformer neural network to universally approximate $\beta$-Hölder continuous functions on $\mathcal{M}$, to be detailed in Section 2.4. The variance is bounded by a novel calculation of the covering number of the transformer network class used in Theorem 1.

## 2.4 Transformer Approximation Theory

**Theorem 2.** *Let $M, \tau, R, H_f > 0$, $0 < \beta \leq 1$, $d, D \in \mathbb{N}$ and $\mathcal{M}$ satisfy Assumption 1. For any $\epsilon \in (0, 1)$, there exists a transformer neural network class $\mathcal{T}(L_T, L_{\text{FFN}}, w_{\text{FFN}}, l, d_{embd}, m, R, \kappa)$ with parameters*

$$L_T = O\big(\log(d)\big), \quad L_{\text{FFN}} = O\big(\log(\epsilon^{-1})\big), \quad w_{\text{FFN}} = O(1), \quad l = O\big(d\epsilon^{-\frac{d}{\beta}}\big)$$

$$d_{embd} = O(1), \quad m = O\big(d\epsilon^{-\frac{d}{\beta}}\big), \quad \kappa = O\big(d^2\epsilon^{-\frac{2d}{\beta}}\big),$$

*where $O(\,\cdot\,)$ hides terms in $C_{\mathcal{M}}, D, H_f, \tau$, such that for any target function $f$ satisfying Assumption 2, if the network parameters $\theta$ are properly chosen, then the network yields a function $\mathrm{T}_\theta \in \mathcal{T}(L_T, L_{\text{FFN}}, w_{\text{FFN}}, l, d_{embd}, m, R, \kappa)$ with the approximation error*

$$\|\mathrm{T}_\theta - f\|_{L^\infty(\mathcal{M})} \leq \epsilon$$

Theorem 2 is proved in Appendix E.2. In our proof, we decompose $f(x)$ as a sum of terms over local neighborhoods $U_1, ..., U_{C_{\mathcal{M}}} \subseteq \mathcal{M}$ covering $\mathcal{M}$. Approximations on overlapping neighborhoods containing $x$ will then be combined via a partition of unity (**PoU**) $\{\rho_n\}_{n=1}^{C_{\mathcal{M}}}$ which subordinates $\{U_n\}_{n=1}^{C_{\mathcal{M}}}$. This will give us the expression $f(x) = \sum_{n=1}^{C_{\mathcal{M}}} f_n(x)\mathbf{1}_{U_n}(x)$ with $f_n = f\rho_n : \mathcal{M} \to \mathbb{R}$. On each local neighborhood $U_n$, we project the input $x \in \mathcal{M} \subseteq \mathbb{R}^D$ to the tangent coordinate in $[0, 1]^d$. This will give us the following *local decomposition* of the target function:

$$f(x) = \sum_{n=1}^{C_{\mathcal{M}}} \tilde{f}_n \circ \phi_n(x)\mathbf{1}_{U_n}(x) \tag{7}$$

where $\tilde{f}_n = f_n \circ \phi_n^{-1} : [0, 1]^d \to \mathbb{R}$ and $\phi_n : \mathcal{M} \to [0, 1]^d$ is a projection onto the local tangent space. We then construct transformers to approximate the $\tilde{f}_n, \phi_n, \mathbf{1}_{U_n}$ components in (7). A diagram of the constructed transformer network approximating $f : \mathcal{M} \to \mathbb{R}$ is given in Figure 1. The following key lemma is used to efficiently approximate each low-dimensional function $\tilde{f}_n$ on $d$-dimensional coordinates.

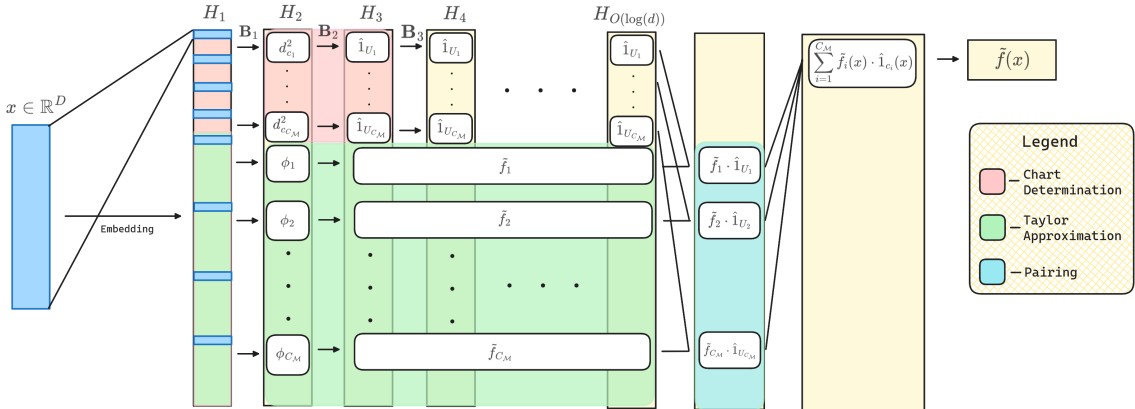

Figure 1: Diagram of the transformer architecture constructed in Theorem 2. T computes approximations of $f(x)$ on each local chart $U_n \subseteq \mathcal{M}$ by first projecting $x$ to the tangent coordinates in $\mathbb{R}^d$ via $\phi_n(x)$ and then approximating $f(x)$ with local Taylor polynomials. A shallow sub-network computes indicators $\mathbf{1}_{U_n}$ for each local chart in parallel. The results of the two sub-networks are then multiplied together and summed to produce the final result. Here $H_i$ denotes the embedding matrix before the $i$th transformer block $\mathbf{B}_i$.

**Lemma 1.** *Let $H_f, R > 0$, $d \in \mathbb{N}$ and $0 < \beta \le 1$. For any $\epsilon \in (0,1)$, there exists a transformer neural network class $\mathcal{T}(L_T, L_{\text{FFN}}, w_{\text{FFN}}, l, d_{embd}, m, R, \kappa)$ with parameters*

$$L_T = O\big(\log(d)\big), \quad L_{\text{FFN}} = O\big(1\big), \quad w_{\text{FFN}} = O\big(1\big), \quad l = O\big(d\epsilon^{-\frac{d}{\beta}}\big)$$

$$d_{embd} = O\big(1\big), \quad m = O\big(d\epsilon^{-\frac{d}{\beta}}\big), \quad \kappa = O\big(d^2\epsilon^{-\frac{2d}{\beta}}\big),$$

*where $O(\cdot)$ hides terms in $H_f$, such that, for any $\beta$-Hölder continuous function $f : [0,1]^d \to \mathbb{R}$, with Hölder constant no more than $H_f$ and $\|f\|_{L^\infty([0,1]^d)} \le R$, if the network parameters $\theta$ are properly chosen, this transformer network yields a function $\mathrm{T}_\theta \in \mathcal{T}(L_T, L_{\text{FFN}}, w_{\text{FFN}}, l, d_{embd}, m, R, \kappa)$ such that*

$$\|\mathrm{T}_\theta - f\|_{L^\infty([0,1]^d)} \le \epsilon.$$

Lemma 1 is proved in Appendix E.1. We develop a novel lemma - *Interaction Lemma* 3, implementing a highly-sparse pairwise interaction between two arbitrary tokens $h_{t_1}, h_{t_2}$, as a crucial architectural building block allowing us to easily implement more complex functions, architecture serialization, and parallelization (7). This result highlights a distinct advantage of transformer function approximation over ReLU function approximation [Yarotsky, 2016]: **A transformer network only needs a constant** $O\big(\log(d)\big)$ **number of layers to approximate** $f : [0,1]^d \to \mathbb{R}$ independent of the desired accuracy $\epsilon$. In contrast, the depth of ReLU feed-forward networks is in the order of $\log(\epsilon^{-1})$ [Yarotsky, 2016] This is desirable from an empirical point of view, where wider networks instead of deeper ones tend to achieve superior performance [Kaplan et al., 2020, Lee et al., 2020].

## 3  Predicting Empirical Scaling Laws and Validation on LLMs

Our theory provides practical insights by predicting neural scaling laws for transformers, as given in (3) and (4), by explicitly quantifying the data scaling exponent $\alpha_D$ and the model scaling exponent $\alpha_N$ as a function of the intrinsic dimension (**ID**) $d$. If we assume the language modeling objective has Lipschitz regularity such that $\beta = 1$ in Assumption 2, then Theorem 1 predicts the scaling law between the squared generalization error and the data size $n$, as given in (3), with $\alpha_D = \frac{2}{2+d}$, and the model scaling law with exponent given by $\alpha_N = \frac{2}{d}$. For a full derivation refer to Section B. We will observe how well our theory predicts these exponents both by pretraining small models from scratch and evaluating existing open-source model suites [Biderman et al., 2023].

In the following, we denote $\alpha_D$ and $\alpha_N$ as the scaling exponents predicted by our theory, where we numerically estimate the intrinsic dimension of data, denoted by $d_{\mathcal{D}}$. The empirical exponents are

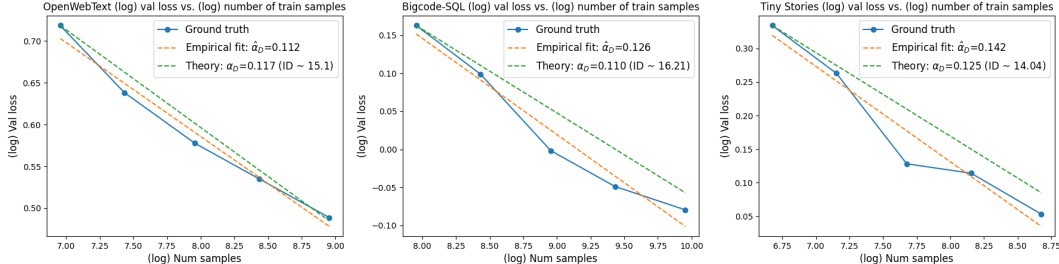

Figure 2: Observed and predicted data scaling laws on OpenWebText, The Stack-SQL, and Tiny Stories pretraining datasets. All estimates are close ($\pm 0.02$) and appear to reflect varying levels of pretraining data complexity. **Note:** $\hat{\alpha}_D$ denotes the empirically observed data scaling exponent and $\alpha_D$ denotes the theoretically estimated exponent.

denoted by $\hat{\alpha}_D$ and $\hat{\alpha}_N$. To obtain the data scaling exponent $\hat{\alpha}_D$, we plot the test loss (comparable to squared error) versus the data size $n$ in log-log scale, fit the log-log curve with a line, and then obtain $\hat{\alpha}_D$ from the magnitude of the slope. The model scaling exponent $\hat{\alpha}_N$ is obtained similarly.

**Estimating the ID of Text** To predict scaling exponents, we must first estimate the intrinsic dimension of our pretraining dataset $\mathcal{D}$. While we can do this directly for image datasets [Russakovsky et al., 2014, Pope et al., 2021], we cannot do this directly for textual datasets. Instead, we will estimate the intrinsic dimension of the input data by estimating the intrinsic dimension of token embeddings. Specifically, we will represent each input token with its corresponding final-layer token embedding. Given a pretraining test set $\mathcal{D}_{\text{test}}$ we embed a random $l = 1024$ length subsequence from each document $D_k \in \mathcal{D}_{\text{test}}$. We then randomly sub-sample 32 final-layer tokens from the embedded subsequence and shuffle together all the embeddings. To estimate the ID of the embeddings we use the Maximum Likelihood Estimation ID algorithm [Levina and Bickel, 2004, Pedregosa et al., 2011] with $K = 20$ neighbors. We split the sampled embedding tokens into batches of 4096 and run the MLE estimator on each batch, averaging together for the final result. Unless otherwise specified, we embed each document $D_k \in \mathcal{D}_{\text{test}}$ using a 125 million parameter model $M$ with $L_T = 12$ layers and embedding dimension $d_{embd} = 768$. We first pretrain $M$ on the full $\mathcal{D}_{\text{train}}$ for $200,000$ steps or until convergence.

**Intrinsic dimension predicts the empirical data scaling exponent** $\hat{\alpha}_D$ To validate our prediction of the dataset scaling exponent $\alpha_D$ we pretrain a series of 125 million parameter GPT-style LLMs on three different datasets: OpenWebText [Gokaslan et al., 2019], the SQL portion of The Stack [Kocetkov et al., 2022], and Tiny Stories [Eldan and Li, 2023]. We train across three orders of dataset size to fit scaling laws. Detailed hyperparameters can be found in the Appendix A. We report the observed scaling laws in log scale in Figure 2. In addition, we plot our predicted test loss whose slope is given by $\alpha_D = \frac{2}{2+d_\mathcal{D}}$ where $d_\mathcal{D}$ is the our estimated intrinsic dimension.

Empirically, we find all three datasets produce nearly log-linear laws whose exponents lie between $0.1 < \hat{\alpha}_D < 0.15$. Tiny Stories has the largest exponent, indicating the fastest rate of convergence, followed by the SQL dataset, followed by OpenWebText. This matches our rough intuition, since Tiny Stories is a synthetically generated dataset with less complexity than the other two datasets and thus easier to learn. The predicted exponents $\alpha_D$ generally over-estimate $\hat{\alpha}_D$ but otherwise closely match up to $\pm 0.02$ absolute error. Additionally, the predicted $\alpha_D$ reflect the previouly mentioned differences in complexity of pretraining datasets. In particular, Tiny Stories has a smaller estimated ID than both OpenWebText and SQL, resulting in a larger predicted $\alpha_D$ as desired.

**Predicting empirical model scaling exponent** $\hat{\alpha}_N$ **with intrinsic dimension** To validate our predictions of the model scaling exponent $\alpha_N = \frac{2}{d_\mathcal{D}}$, we evaluate two model scaling suites: GPT2 [Radford et al., 2019] and Pythia [Biderman et al., 2023]. We refer to Kaplan et al. [2020] for GPT2's $\hat{\alpha}_N$ and estimate $\hat{\alpha}_N$ using OpenWebText as a proxy for GPT2's pretraining data. We compute $\alpha_N$ for Pythia by evaluating each model on The Pile's test set [Gao et al., 2021]. We also estimate $d_\mathcal{D}$ on the publicly available pretraining data to predict $\alpha_N$. The results are reported in Figure 3. Our predicted $\alpha_N$ under-estimates the empirical $\hat{\alpha}_N$. We conjecture this to be due to a number of factors including possible under-training of the largest models and the intrinsic entropy of the data distribution.

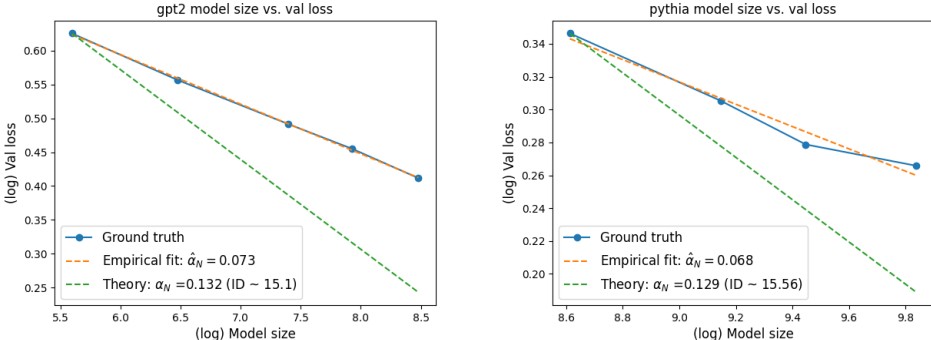

Figure 3: Observed and predicted model scaling laws in model size on GPT2 and Pythia scaling suites. $\alpha_N$ denotes the empirically observed scaling exponent, and $\hat{\alpha}_N$ denotes the theoretically predicted exponent. **Note:** we estimate $\alpha_N$ for GPT2 using OpenWebText.

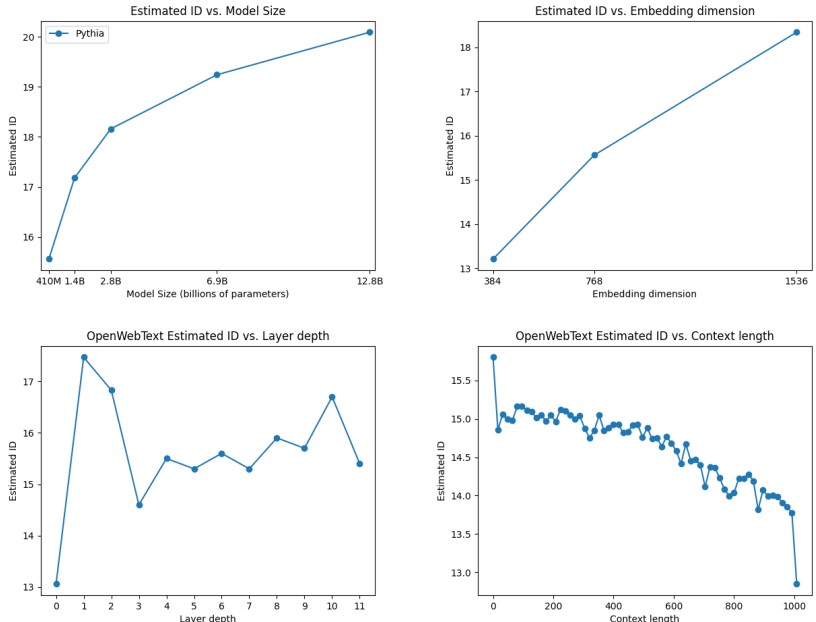

Figure 4: **Top left:** Estimated ID vs. number of parameters. **Top right:** Estimated ID vs. the embedding dimension. **Bottom left:** Variation of estimated ID across model layers. **Bottom right:** Variation of estimated ID across context position.

**Ablating the impact of model architecture on the estimated ID** Practical application of our theory relies on a good estimate of the intrinsic dimension. However, there are many factors potentially biasing our estimate. Of particular interest is the embedding model's embedding dimension, depth, context length, and number of parameters. We ablate these factors in Figure 4, plotting estimated ID against each factor.

Overall, we find the estimated ID is fairly stable across each factor. As the number of parameters increases, the estimated ID of The Pile slightly increases from 15.56, via a 410 million parameter model, to 20.02 with a 12.8 billion parameter model. ID on OpenWebText behaves similarly when increasing the embedding dimension, increasing from 15.56 when $d_{embd} = 768$ to 18.68 when $d_{embd} = 1536$. When fixing a single model, we find the ID across intermediate embedding layers is small initially but then increases and decreases again, stabilizing around the ID of the final layer. We observe that the ID appears to inversely correlate with sequence length, decreasing from 15.86 for very short sequences to 12.9 for sequences around 1024 tokens.

**Predicting $\alpha_N$ from $\alpha_D$ (and vice versa) without estimating ID**  Above we estimated $\alpha_D$ and $\alpha_N$ by first estimating the intrinsic dimension $d$ for a model's pretraining dataset. However, estimating $d$ may not always be possible when pretraining data is not public. Alternatively, we can predict $\alpha_D$ in terms of $\alpha_N$ (and vice versa) without ever needing to estimate $d$:

$$\alpha_D = \frac{2}{2+d} = \frac{2\frac{1}{d}}{2\frac{1}{d} + 1} = \frac{\alpha_N}{\alpha_N + 1}, \quad \alpha_N = \frac{\alpha_D}{1 - \alpha_D}$$

See Table 1 for ID-free estimations of empirically observed exponents in the literature [Sharma and Kaplan, 2022, Hoffmann et al., 2022].

| GPT-2 | | Chinchilla | |
|---|---|---|---|
| $\hat{\alpha}_N = 0.076$ | $\alpha_D = 0.070$ | $\hat{\alpha}_N = 0.34$ | $\alpha_D = 0.25$ |
| $\hat{\alpha}_D = 0.095$ | $\alpha_N = 0.106$ | $\hat{\alpha}_D = 0.28$ | $\alpha_N = 0.33$ |

Table 1: ID-free estimation of scaling exponents for GPT-2 and Chinchilla.

## 4  Related Work

The theoretical properties and advantages of transformers have been studied from many different perspectives [Jelassi et al., 2022, Zhang et al., 2022, Bai et al., 2023, Pérez et al., 2021, Sanford et al., 2024]. Most related to us are Yun et al. [2019], Edelman et al. [2022], Wei et al. [2022], Takakura and Suzuki [2023] in which transformers were studied from an approximation viewpoint. The work in Yun et al. [2019] proved that transformer models are universal approximators of continuous permutation equivariant sequence-to-sequence functions with compact support, while the network size suffers from the curse of dimensionality (the number of entries in the input sequence). Takakura and Suzuki [2023] studied the approximation and estimation ability of Transformers as seq-to-seq functions with infinite dimensional input, where anisotropic smoothness avoids the curse of dimensionality.

In the applications of Large Language Models (LLMs), empirical findings have demonstrated some correlation between the performance of transformers and the low-dimensional data structures [Razzhigaev et al., 2023, Min et al., 2023, Aghajanyan et al., 2020, Pandey, 2024]. Razzhigaev et al. [2023] investigated the intrinsic dimension of embeddings in transformer architectures, and suggested an encoder and decoder embedding property. Most similar to our work is Sharma and Kaplan [2022] which demonstrates an empirical connection between neural scaling laws and the intrinsic dimension of data. While they briefly discuss predictions for LLMs, their theory works best for predicting student-teacher model setups and image classification tasks which seem to enjoy more regularity (and a faster rate of convergence) than language modeling. Despite these empirical findings, we are not aware of any rigorous theoretical justification connecting the scaling laws of transformers with the intrinsic dimension of data. Our paper complements this line of research with statistical estimation and mathematical approximation theories which well-predict the behavior observed in practice.

## 5  Conclusion

**Conclusion**  This paper establishes statistical and approximation theory results for transformers approximating Hölder continuous functions on low-dimensional data manifolds. The resulting bound on the generalization error suffers from exponential dependence only in the intrinsic dimension $d$. The constructed approximations of low-dimensional functions are shallow, requiring only $O\big(\log(d)\big)$ layers independent of the desired accuracy. We demonstrate this theory is accurate in practice by predicting scaling laws in both model size and data size for LLMs trained on natural language datasets. We pay careful attention to the sensitivity of the estimated intrinsic data dimension, finding it is relatively stable with respect to several relevant hyperparameters.

**Limitations and Broader Impact**  One important question unanswered by this work is how the intrinsic data dimension may affect the computational scaling exponent $\alpha_C$. Future work may investigate this direction. Additionally, our empirical experiments make the simplifying assumption that the underlying target function possesses Lipschitz regularity ($\beta = 1$). Better estimates of the correct regularity would likely improve the accuracy of our predictions. More broadly, our work improves fundamental understanding of transformer-based LLMs and improves our ability to theoretically and safely predict future capabilities.

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

# A    Pretraining Hyperparameters

| Architecture hparams | |
|---|---:|
| num layers | 12 |
| num attention heads | 12 |
| embedding dimension | 768 |
| context length | 1024 |
| **Optimization hparams** | |
| Optimizer | AdamW |
| max lr | 6e-4 |
| min lr | 6e-5 |
| lr schedule | linear warmup + cosine decay |
| warmup steps | 2000 |
| max steps | 200,000 |
| global batch size | 1920 |

Table 2: Default hyperparameters for all training jobs. All training was done on four RTX 6000s.

Table 2 lists the default hyper-parameters we use for pretraining. All training was done on four RTX 6000s.

# B    Deriving Lanugage Model Scaling Laws from Statistical and Approximation Theory

## B.1    Squared Regression Error

First we extract bounds on scaling laws in the case of regression squared error. We have the bound from the proof of Theorem 1

$$\int_{\mathcal{M}} \big(f(x) - \hat{\mathrm{T}}_n(x)\big)^2 dQ(x) \le \tilde{O}\big(\epsilon^2 + \frac{Dd^2\epsilon^{-\frac{d}{\beta}}}{n}\big)$$

where $\epsilon$ is the approximation error such that

$$\inf_{\mathrm{T}\in\mathsf{T}} \|f - \mathrm{T}\|_{L^{\infty}(\mathcal{M})} < \epsilon$$

Let the *model size* $N$ of a transformer $\mathrm{T} \in \mathcal{T}$ be $N = L_T(d_{embd}^2(3m + L_{\mathrm{FFN}})) = \log(d)(25(3\epsilon^{-\frac{d}{\beta}} + \log(\epsilon^{-1}))) = \tilde{O}\big(\epsilon^{-\frac{d}{\beta}}\big)$. Write the squared generalization error as

$$L_{\mathrm{sq}}(N,n) = \int_{\mathcal{M}} \big(f(x) - \hat{\mathrm{T}}_n(x)\big)^2 dQ(x).$$

Then

$$L_{\mathrm{sq}}(N,n) \le \tilde{O}\big(\epsilon^2 + \frac{Dd^2\epsilon^{-\frac{d}{\beta}}}{n}\big) = \tilde{O}\big(N^{-\frac{2\beta}{d}} + \frac{N}{n}\big)$$

In the model scaling regime, when data is plentiful, we have $\frac{N}{n} << N^{-\frac{2\beta}{d}}$ which implies the behavior of $L_{\mathrm{sq}}(N,n)$ is dominated by $N^{-\frac{2\beta}{d}}$. This gives us the model scaling exponent as

$$\alpha_N = \frac{2\beta}{d}$$

For the data scaling exponent $\alpha_D$ we will choose $N$ to balance both error terms. The will predict how data size and model size should scale together to achieve a minimal generalization error. We should have

$$N^{-\frac{2\beta}{d}} \asymp \frac{N}{n} \iff n \asymp N^{1+\frac{2\beta}{d}} \asymp N^{\frac{2\beta+d}{d}} \iff N \asymp n^{\frac{d}{2\beta+d}}$$

where $\asymp$ denotes in the same order. Substituting this into the error bound gives

$$L_{\mathrm{sq}}(N, n) \leq \tilde{O}\big(2\frac{N}{n}\big) = \tilde{O}\big(n^{\frac{d}{2\beta+d}-1}\big) = \tilde{O}\big(n^{-\frac{2\beta}{2\beta+d}}\big),$$

which gives rise to

$$\alpha_D = \frac{2\beta}{2\beta + d}.$$

## B.2 From Regression to Classification

After establishing statistical and approximation theory for transformers for regression, we now seek to apply our theory to classification. In language models, the next-token prediction task is a multi-class classification problem, where transformers are trained to predict the probability of the next word out of a large dictionary.

For simplicity, we consider binary classification here. Let $(x, y)$ be a random couple taking values in $\mathcal{M} \times \{0, 1\}$ with joint distribution $P$, where $\mathcal{M}$ is a manifold satisfying Assumption 1. Let $P_x$ be the marginal distribution of $x$. Here $x$ stands for the input feature and $y$ is the corresponding label. The classification goal is to predict the label $y$ given the value of $x$. A decision rule is given by a function $f : \mathcal{M} \to \{0, 1\}$. The performance of a decision rule $f$ is measured by the misclassification error

$$R(f) := P(y \neq f(x)).$$

The Bayes decision rule has the form

$$f^*(x) = \mathbf{1}\{\eta(x) \geq 1/2\}$$

where $\mathbf{1}$ denotes the indicator function and

$$\eta(x) := P(y = 1|x) \tag{8}$$

is the regression function of $y$ on $x$. For binary classification, the goal is to estimate the probability function $\eta(x)$. If $\eta$ is a $\beta$-Hölder function satisfying Assumption 2, our approximation theory in Theorem 2 gives rise to a transformer neural network $\eta_\theta$ such that

$$\|\eta_\theta - \eta\|_{L^\infty(\mathcal{M})} \leq \epsilon \tag{9}$$

for an arbitrary small $\epsilon > 0$.

We next consider the plug-in estimate of $\eta_\theta$ [Audibert and Tsybakov, 2007]: $f_\theta(x) = \mathbf{1}\{\eta_\theta \geq 1/2\}$. The excess risk of the plug-in estimate $f_\theta$ is

$$
\begin{aligned}
\mathcal{E}(f_\theta) &= R(f_\theta) - R(f^*) \\
&= P(y \neq f_\theta(x)) - P(y \neq f^*(x)) \\
&= P(y = f^*(x)) - P(y = f_\theta(x)) \\
&= \mathbb{E}_x[\mathbf{1}\{f^*(x) = 1\}\eta(x) + \mathbf{1}\{f^*(x) = 0\}(1 - \eta(x)) \\
&\quad - \mathbf{1}\{f_\theta(x) = 1\}\eta(x) - \mathbf{1}\{f_\theta(x) = 0\}(1 - \eta(x))] \\
&= \mathbb{E}_x\left[|2\eta(x) - 1|\mathbf{1}\{f_\theta(x) \neq f^*(x)\}\right].
\end{aligned}
$$

We next discuss how the regression error in (9) impacts the excess risk $\mathcal{E}(f_\theta)$. For classification problems, the classification error depends on how well the conditional probability $\eta(x)$ is estimated, and how many points are close to the decision boundary. Following Audibert and Tsybakov [2007], we assume the following margin condition: There exist $c_M > 0$ and $\gamma \geq 0$, such that for any $t > 0$,

$$P_x(0 < |\eta(x) - 1/2| \leq t) \leq c_M t^\gamma. \tag{10}$$

A smaller $\gamma$ means more samples cluster around the decision boundary with a higher likelihood of falling into either class. This is not expected to be the case for natural data, where for most samples it is easy to tell to which class they belong [Kim et al., 2019, Figure 2]. As a result, we assume $\gamma \geq 1$.

Under the margin condition in (10), we follow Audibert and Tsybakov [2007] to decompose the excess risk $\mathcal{E}(f_\theta)$ such that for any $\delta > 0$:

$$
\begin{aligned}
\mathcal{E}(f_\theta) &= \mathbb{E}_x\left[|2\eta(x) - 1|\mathbf{1}\{f_\theta(x) \neq f^*(x)\}\mathbf{1}\{|\eta(x) - 1/2| \leq \delta\}\right] \\
&\quad + \mathbb{E}_x\left[|2\eta(x) - 1|\mathbf{1}\{f_\theta(x) \neq f^*(x)\}\mathbf{1}\{|\eta(x) - 1/2| > \delta\}\right] \\
&\leq 2\delta P_x(|\eta(x) - 1/2| \leq \delta) + 2\mathbb{E}_x\left[|\eta_\theta(x) - \eta(x)|\mathbf{1}\{|\eta_\theta(x) - \eta(x)| > \delta\}\right] \\
&\leq 2c_M \delta^{1+\gamma} + 2\mathbb{E}_x\left[|\eta_\theta(x) - \eta(x)|\mathbf{1}\{|\eta_\theta(x) - \eta(x)| > \delta\}\right]. \tag{11}
\end{aligned}
$$

Plugging the regression error in (9) to the excess risk bound in (11) with $\delta = \epsilon$, we obtain

$$\mathcal{E}(f_\theta) \leq 2c_M \epsilon^{1+\gamma}.$$

When the margin assumption satisfies $\gamma = 1$, the excess risk bound becomes

$$\mathcal{E}(f_\theta) \leq 2c_M \epsilon^2,$$

which demonstrates a connection between the classification risk bound and the squared regression error for the $\eta$ function in (8). This argument partially justifies the connection between empirical neural scaling laws in large language models and the squared regression error. We will leave a rigorous mathematical argument as future work.

### B.3 Cross-Entropy Based Language Model

In practice, language models are trained and evaluated using cross-entropy loss. We consider the next-token prediction in language models. Per-sample (token), language models are trained to minimize the multi-class cross-entropy loss over a large number of classes/tokens:

$$L_{\mathrm{cr}}(n) = -\frac{1}{n} \sum_{i=1}^{n} \sum_{y=1}^{V} \mathbf{1}_{y=y_i^*} \ln(P(y|x_i)) \tag{12}$$

where $V$ is the vocabulary size of the model (number of classes), $y_i^*$ is the ground-truth label of $x_i$, $\mathbf{1}_{y=y_i^*}$ is the indicator function for the event $y = y_i^*$, and $P(y|x)$ is the conditional probability of labels given $x$. For the next-token prediction, transformers are trained to estimate the probability function $P(y|x)$, and the next token is predicted to the word with the highest probability. The test loss is evaluated using cross-entropy on test data. We will leave the error bound on the cross-entropy loss as future work.

## C  Notation

$H$ will represent the *embedding matrix* of a transformer and $h_i$ will represent the $i$th *token* (column) of the embedding matrix. $h_i^j$ will denote the $j$th component of the vector $h_i$. $\|\cdot\|_p, p > 1$, will denote the $p$th norm of vectors, and $\|\cdot\|_{p,q}$ will denote the component-wise matrix norm. For a matrix $A \in \mathbb{R}^{m \times n}$, $\|A\|_{p,q} = (\sum_{j=1}^{n}(\sum_{i=1}^{m}|A_{ij}|^p)^{\frac{q}{p}})^{\frac{1}{q}}$. In particular, we denote $\|A\|_\infty = \|A\|_{\infty,\infty}$ and $\|A\|_1 = \|A\|_{1,1}$ for matrices. $\|\cdot\|_{L^p(U)}$ will denote the $L^p$ norm of a function on $U \subseteq \mathbb{R}^D$. Neural network blocks will always be **bolded**. Sometimes neural network blocks will also be sub-scripted with their corresponding vector of *weights* $\theta$. We use $\|\theta\|_\infty$ to denote the largest magnitude of the weight parameters. However we will often omit $\theta$ and leave the dependence as implicit. In many places, we will also write $\theta_{\mathbf{NN}}$ to denote the weights for a neural network **NN**. $e_i$ will denote the $i$th standard basis vector where the $i$th component is 1 and all other entries are 0. $\sigma$ will always denote the ReLU activation. $B(x, r)$ denotes the Euclidean ball of radius $r$ centered at $x \in \mathbb{R}^D$. For a vector $h_t$, $h_t^{i:j} \in \mathbb{R}^{j-i}$ will denote the vector such that $(h_t^{i:j})^k = h_t^k$ i.e. the contiguous components of $h_t$ from $i, ..., j - 1$.

## D  Details about Transformer Neural Networks

We have defined transformer neural networks in Definition 1. Some detailed definitions are given below.

**Definition 3** (Transformer Block). *We define a transformer block* B *as a residual composition of the form*

$$\mathrm{B}(H) = \mathrm{FFN}(\mathrm{MHA}(H) + H) + \mathrm{MHA}(H) + H \tag{13}$$

*taking as input an embedding matrix* $H \in \mathbb{R}^{d_{embd} \times l}$ *and* MHA *is a multi-headed attention layer and* FFN *is a feed-forward layer. To define the multi-headed attention layer we first define the attention mechanism.*

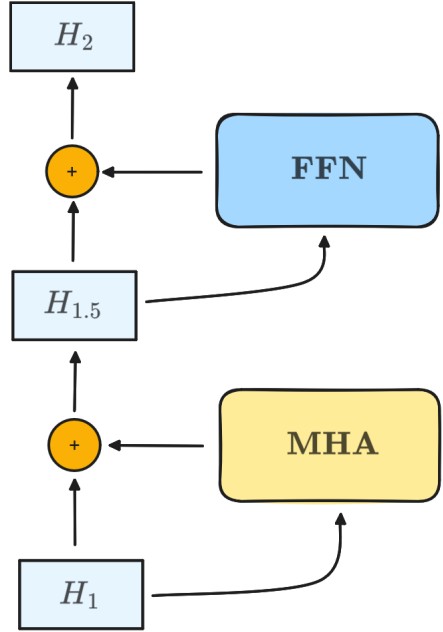

Figure 5: Diagram of transformer block.

**Definition 4** (Attention).

$$\mathrm{A}_{Q,K,V}(H) = VH\sigma((KH)^T QH) \tag{14}$$

*where $Q, K, V \in \mathbb{R}^{d_{embd} \times d_{embd}}$ are referred to as the query, key, and value matrices and $\sigma(x) = \max(0, x)$ is the ReLU activation function. Often we will omit the dependence of $A$ on $Q, K, V$. We note it will be convenient to write the action of $A$ on the $i$th column $h_i$ of $H$ as*

$$\mathrm{A}(h_i) = \sum_{j=1}^{l} \sigma(\langle Qh_i, Kh_j \rangle) V h_j \tag{15}$$

*. This allows us to interpret the $i$th column of the output of $A$ as a linear combination of the values weighted by the interaction of $h_i$th query and the $h_j$th key. Multi-headed attention can then be defined as*

$$\mathrm{MHA}(H) = W_O(\mathrm{concat}_j(V_j H \sigma((K_j H)^T Q_j H)) \tag{16}$$

*where the output of each of $j \in \{1, ..., M\}$ attention heads is concatenated and $W_O \in \mathbb{R}^{d_{embd} \times m d_{embd}}$. Frequently we will simply take $W_O$ to be a sum so that*

$$\mathrm{MHA}(H) = \sum_{j=1}^{m} V_j H \sigma((K_j H)^T Q_j H). \tag{17}$$

We also formally define the feed-forward layer FFN:

**Definition 5** (Feed-forward Layer). *A feed-forward layer of depth $L_{\mathrm{FFN}}$ and width $w_{\mathrm{FFN}}$ is of the form*

$$\mathrm{FFN}(h) = W_{L_{\mathrm{FFN}}} \sigma(W_{L_{\mathrm{FFN}}-1}...\sigma(W_1 h + b_1)... + b_{L_{\mathrm{FFN}}-1}) + b_{L_{\mathrm{FFN}}}$$

*where $W_2, ..., W_{L_{\mathrm{FFN}}-1} \in \mathbb{R}^{d_{embd} \times d_{embd}}$, $W_1 \in \mathbb{R}^{w_{\mathrm{FFN}} \times d_{embd}}$, $W_{L_{\mathrm{FFN}}} \in \mathbb{R}^{d_{embd} \times w_{\mathrm{FFN}}}$, $b_1, ..., b_{L_{\mathrm{FFN}}} \in \mathbb{R}^{d_{embd}}$ and the activation is ReLU. Note, each feed-forward layer is applied tokenwise to an embedding matrix $H$.*

We can also define a *class $\mathcal{FFN}$* of feed-forward networks:

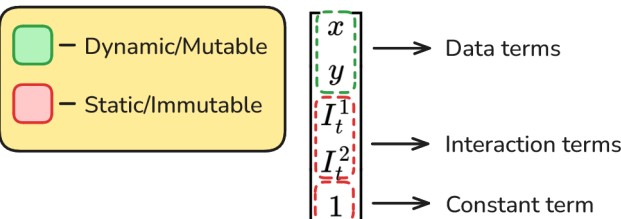

Figure 6: Diagram of a structured token. The first two rows contain mutable data used to compute the target function. The remaining rows are never changed after initialization.

**Definition 6** (Feed-forward Network Class)**.**

$$\mathcal{FFN}(L_{\text{FFN}}, w_{\text{FFN}}) = \big\{\text{FFN}_\theta \mid \text{FFN}_\theta \text{ is a feed-forward network with weights } \theta$$
$$\text{with at most } L_{\text{FFN}} \text{ layers with width } w_{\text{FFN}}\big\}.$$

Sometimes we may omit the dependence of $\mathcal{F}$ on $w_{\text{FFN}}$. In these cases we assume $w_{\text{FFN}} = d_{embd}$. We similarly define multi-headed attention and transformer block classes:

**Definition 7** (Multi-headed Attention and Transformer Block Classes)**.**

$$\mathcal{MHA}(m) = \big\{\text{MHA}_\theta \mid \text{MHA}_\theta \text{ is a multi-headed attention network with weights } \theta$$
$$\text{with at most } m \text{ attention heads.}\big\}$$

$$\mathcal{B}(m, L_{\text{FFN}}, w_{\text{FFN}}) = \big\{\text{B}_\theta \mid \text{B}_\theta \text{ is a transformer block with weights } \theta, \text{ a } m\text{-headed multi-headed attention layer,}$$
$$\text{and a } L_{\text{FFN}} \text{ deep feed-forward layer with width } w_{\text{FFN}}\big\}.$$

As with $\mathcal{F}$, we may sometimes omit the dependence of B on $w_{\text{FFN}}$. In these cases we take $w_{\text{FFN}} = d_{embd}$. We can parameterize the class of transformer neural networks $\mathcal{T}$ as Definition 2.

Lastly, our constructions will rely heavily on a particular structuring of the tokens/columns in the transformer's hidden states. The first two rows will contain mutable data used to compute the target function. The remaining rows will be immutable, containing positional/interaction vectors allowing us to uniquely identify each row and a constant term serving as a kind of "scratchpad". See Figure 6 for a diagram.

# E   Proof of Transformer Approximation Theory

We first prove the approximation theories in Section 2.4. We first prove Lemma 1 in Section E.1, and then prove Theorem 2 in Section E.2.

## E.1   Proof of Lemma 1

*Proof of Lemma 1.* Fix $f : [0,1]^d \to \mathbb{R}$ and $\epsilon > 0$. We aim to approximate $f$ efficiently with a transformer $\text{T} \in \mathcal{T}(L_T, L_{\text{FFN}}, w_{\text{FFN}}, l, d_{embd}, m, R, \kappa)$. We will proceed similarly to Yarotsky [2016] by approximating $f$ locally with piecewise constant functions. First, we partition the domain uniformly into blocks indexed by $n \in \{1, ..., N\}^d$ for some $N \geq 1$ with $U_n = \{x \in [0,1]^d : \forall i \in \{1, ..., d\}, \left| x^i - \frac{n^i}{N-1} \right| \leq \frac{1}{N-1}\}$. In this proof, we use $x^i$ and $n^i$ to denote the $i$th coordinate of $x$ and $n$ respectively. Then we construct a partition of unity (**PoU**) via

$$\phi_n(x) = \prod_{i=1}^d \psi\left(3N(x^i - \frac{n^i - 1}{N - 1})\right) \tag{18}$$

with the trapezoid function $\psi : \mathbb{R}^d \to \mathbb{R}$ as

$$\psi(x) = \begin{cases} 1 & |x| \leq 1 \\ 1 - |x| & 1 \leq |x| \leq 2 \\ 0 & |x| \geq 2 \end{cases}.$$

We can now write our approximation for $f$ as

$$\hat{f}(x) = \sum_n \phi_n(x) f_n$$

where $f_n = f(x_n)$ with $x_n$ being the center point of $U_n$. This yields the following approximation error:

$$
\begin{aligned}
\|f - \hat{f}\|_{L^\infty([0,1]^d)} &= \sup_{x \in [0,1]^d} \left| f(x) - \hat{f}(x) \right| \\
&= \sup_{x \in [0,1]^d} \left| f(x) - \sum_{n \in \{1,\dots,N\}^d} f_n \phi_n(x) \right| \\
&= \sup_{x \in [0,1]^d} \left| \sum_{n \in \{1,\dots,N\}^d} f(x) \phi_n(x) - \sum_{n \in \{1,\dots,N\}^d} f_n \phi_n(x) \right| \\
&= \sup_{x \in [0,1]^d} \sum_{n \in \{1,\dots,N\}^d} |f(x) - f_n| \phi_n(x) \\
&= \sup_{x \in [0,1]^d} \sum_{|x^i - \frac{n^i}{N-1}| \leq \frac{1}{N-1}} |f(x) - f_n| \\
&\leq \sup_{x \in [0,1]^d} \sum_{|x^i - \frac{n^i}{N-1}| \leq \frac{1}{N-1}} H_f \|x - x_n\|_2^\beta \\
&\leq \sup_{x \in [0,1]^d} 2^d \max_{|x^i - \frac{n^i}{N-1}| \leq \frac{1}{N-1}} H_f \|x - x_n\|_2^\beta \\
&\leq \sup_{x \in [0,1]^d} 2^d H_f \frac{d^\beta}{(N-1)^\beta} = \frac{2^d d^\beta H_f}{(N-1)^\beta}
\end{aligned}
$$

Thus we can control this error simply by increasing $N$. It suffices to pick $N = d(\frac{2^d H_f}{\epsilon})^{\frac{1}{\beta}} + 1$ to obtain an $\epsilon$ error bound.

We next construct an approximation to $\hat{f}$ with a transformer neural network T. In fact we can represent $\hat{f}$ **exactly** as a transformer neural network.

At a high level, we will proceed by building up each $\phi_n$ as an accumulated partial product $p_n$ in parallel over $d$ transformer blocks. Each block will use at most $d_{embd} N_{embd}^d$ attention heads. Each head will be responsible for multiplying two partial products. Conveniently $\phi_n$ can be exactly implemented with a two-layer FFN and applied column-wise to each term. The constant terms $f_n$ will then be multiplied in and summed at the final layer to compute the output.

**Step 1: Embed the input**  Now fix input $x \in \mathbb{R}^{1 \times d}$. First we augment the input, via concatenation, with a sequence of 0s: $\mathbf{0}_{Nd+N^d}$ forming input $x' \in \mathbb{R}^{1 \times d + Nd + N^d}$ (note this is a linear operation). We can then construct a linear embedding layer $[1, 0, 0, 0, 0]^T = E \in \mathbb{R}^{d_{embd} \times 1}$ resulting in the input embedding matrix $H \in \mathbb{R}^{d_{embd} \times (d + Nd + N^d d)}$ of the form

$$
\begin{bmatrix}
x^1 & \dots & x^d & \mathbf{0}_{Nd} & \dots & \mathbf{0}_{N^d d} \\
0 & \dots & 0 & \mathbf{0}_{Nd} & \dots & \mathbf{0}_{N^d d} \\
0 & \dots & 0 & \mathbf{0}_{Nd} & \dots & \mathbf{0}_{N^d} \\
0 & \dots & 0 & \mathbf{0}_{Nd} & \dots & \mathbf{0}_{N^d d} \\
0 & \dots & 0 & \mathbf{0}_{Nd} & \dots & \mathbf{0}_{N^d d}
\end{bmatrix}
$$

where $d_{embd} = 5$. To this we add a fixed positional encoding $PE \in \mathbb{R}^{d_{embd} \times (d+Nd+N^d d)}$, producing the first hidden embedding matrix

$$H_1 = \begin{bmatrix} x^1 & \dots & x^d & \mathbf{0}_{Nd} & \dots & \mathbf{0}_{N^d d} \\ 0 & \dots & 0 & \mathbf{0}_{Nd} & \dots & \mathbf{0}_{N^d d} \\ \mathcal{I}_1 & \dots & \dots & \dots & \dots & \mathcal{I}_{d+dN+N^d d} \\ 1 & \dots & \dots & \dots & \dots & 1 \end{bmatrix}$$

where $I_i \in \mathbb{R}^2$ represents an *interaction term* determining when each token embedding will interact with another in the attention mechanism. We can set $I_i = (\cos(\frac{i}{l}\frac{\pi}{2}), \sin(\frac{i}{l}\frac{\pi}{2}))$ where $l$ is the number of hidden tokens. Note, this type of positional embedding is commonly known as the sinusoidal positional encoding and can be visualized as rotations of the unit vector $e_1$ around the first quadrant of the unit circle.

**Step 2: Pre-compute $\psi(3N(x^i - \frac{j-1}{N-1}))$** Now we pre-compute the terms $\psi(3N(x^i - \frac{j-1}{N-1}))$ from which we can construct our partition of unity. We define $s_{i,j} = \psi(3(N-1)(x^i - \frac{j-1}{N-1}))$ for $1 \le i \le d, 1 \le j \le N$. We know $\psi(3(N-1)\cdot)$ can be computed exactly with a two-layer FFN. So all we must do is prepare the input $x^i - \frac{j-1}{N-1}$. We will do this with one transformer block $B_1$. The results will be stored in token columns $h_{d+1}, ..., h_{d+Nd}$.

Define $B_1 = \mathcal{B}(Nd, 7)$ i.e. $B_1$ has a multi-headed attention layer with $Nd$ attention heads and a feed-forward layer with depth 7. We will construct each of the $ij$th blocks using the Interaction Lemma 3. We may choose *data kernels*

$$Q_{ij}^{data} = \begin{bmatrix} 0 & 0 & 0 & 0 & 1 \\ 0 & 0 & 0 & 0 & 1 \end{bmatrix} \in \mathbb{R}^{2 \times d_{embd}}, \quad K_{ij}^{data} = \begin{bmatrix} 1 & 0 & 0 & 0 & 0 \\ 0 & 0 & 0 & 0 & -\frac{j}{N-1}+1 \end{bmatrix} \in \mathbb{R}^{2 \times d_{embd}}$$

so that for the token embedding $h_{ij} = h_{d+i*d+j}$, we have

$$A_{ij}(h_{ij}) = \sum_{k=1}^{l} \sigma(\langle Q_{ij}h_{ij}, K_{ij}h_k \rangle)V_{ij}h_k$$

$$= \sum_{k=1}^{l} \sigma(\langle Q_{ij}^{data}h_{ij}, K_{ij}^{data}h_k \rangle + \langle Q_{ij}^{\mathcal{I}}h_{ij}, K_{ij}^{\mathcal{I}}h_k \rangle - C)e_2$$

$$= \sum_{k=1}^{l} \sigma(h_k^1 - \frac{j-1}{N-1} + 1 + \langle Q_{ij}^{\mathcal{I}}h_{ij}, K_{ij}^{\mathcal{I}}h_k \rangle - C)e_2$$

$$= \sigma(x^i - \frac{j-1}{N-1} + 1)e_2 = (x^i - \frac{j-1}{N-1} + 1)e_2$$

where we can choose token $h_{ij}$ to only interact with token $h_i$. Otherwise we have $A_{ij}(h_t) = 0$ for $h_t \ne h_{ij}$. Further, the weights of $\theta_{A_{ij}}$ of $A_{ij}$ are bounded such that $\|\theta_{A_{ij}}\|_\infty = O(l^2)$ according to Lemma 3.

The added output of the multi-headed attention layer is then $H_{1.5} = \text{MHA}_1(H) + H$ and is given by

$$H_{1.5} = \begin{bmatrix} x^1 & \dots & x^d & \mathbf{0}_{Nd} & \dots & \dots & \mathbf{0}_{N^d d} \\ 0 & \dots & 0 & x^1+c & \dots & x^d-1+c & \mathbf{0}_{N^d d} \\ \mathcal{I}_1 & \dots & \dots & \dots & \dots & \dots & \mathcal{I}_{d+Nd+N^d d} \\ 1 & \dots & \dots & \dots & \dots & \dots & 1 \end{bmatrix}.$$

It remains to subtract the positive term $c = 1$ and then apply $\psi$ for the tokens $d+1, ..., d+dN$ while leaving the other tokens untouched. Define

$$
W_1 = \begin{bmatrix} 0 & 1 & 0 & 0 & -1+M \\ 0 & -1 & 0 & 0 & M \\ 0 & 0 & 1 & 0 & 0 \\ 0 & 0 & 0 & 1 & 0 \\ 0 & 0 & 0 & 0 & 1 \end{bmatrix}, \quad b_1 = 0
$$

$$
W_2 = \begin{bmatrix} 1 & 0 & 0 & 0 & -M \\ 0 & 1 & 0 & 0 & -M \\ 0 & 0 & 1 & 0 & 0 \\ 0 & 0 & 0 & 1 & 0 \\ 0 & 0 & 0 & 0 & 1 \end{bmatrix}, \quad b_2 = 0.
$$

Here $W_1$ shifts values in the second component of each token to the first and subtracts 1. The second component is negated to be subtracted from $H_{1.5}$. $I_t$ and the last component are preserved. A large positive number $M > 2\|x\|_\infty$ is added to prevent negative terms from getting erased by ReLU. $W_2$ removes $M$ after the activation is applied. We can construct a two-layer network $\text{FFN}_1$ to apply $\psi$ to the first component of each token $h_t$. In tokens $h_d, ..., h_{d+Nd}$ this produces the desired $s_{ij}$ terms.

Now we simply must ensure we zero-out changes to tokens outside the range $d+1, ..., d+Nd$. Via Lemma 4 we can construct a two-layer feed-forward *gating network* $\text{FFN}_2$ such that $\text{FFN}_2(h_t) = h_t$ for $t \leq d$ and $\text{FFN}_2(h_t)$ is zero except for the last three rows when $t > d + dN$. We can again invoke the same lemma to produce $\text{FFN}_3 \in \mathcal{FFN}(2)$ such that $\text{FFN}_3(h_t) = h_t$ when $t > d$ and zero except for the last three rows otherwise. Applying $\text{FFN}_2$ and $\text{FFN}_3$ zeroes out tokens outside $d+1, ..., d+dN$ while preserving the rest. Finally, we define the last layer

$$
W_7 = \begin{bmatrix} 1 & 0 & 0 & 0 & 0 \\ 0 & 1 & 0 & 0 & 0 \\ 0 & 0 & 0 & 0 & 0 \\ 0 & 0 & 0 & 0 & 0 \\ 0 & 0 & 0 & 0 & 0 \end{bmatrix}, \quad b_7 = 0
$$

which zeros out all except the first two components. This produces the next embedding matrix $H_2$:

$$
H_2 = \begin{bmatrix} x^1 & ... & x^d & s_{1,1} & ... & s_{d,N} & \mathbf{0}_{N^d d} \\ 0 & ... & 0 & \mathbf{0}_{Nd} & ... & ... & \mathbf{0}_{N^d d} \\ \mathcal{I}_1 & ... & ... & ... & ... & ... & \mathcal{I}_{d+Nd+N^d d} \\ 1 & ... & ... & ... & ... & ... & 1 \end{bmatrix}
$$

.

**Step 3: Building up partial products** Now we can start construction of the partial products for each patch $\phi_n$, $n \in \{1, ...., N\}^d$. For a fixed patch $n$, we write $p_{n,k,i}$ to indicate the $i$th partial product of $2^{k-1}$ terms. Formally, $p_{n,k,i} = p_{n,k-1,2i-1} \cdot p_{n,k-1,2i}$ with $p_{n,1,i} = s_{i,n^i}$ where $s_{i,n^i} = \psi(3(N-1)(x^i - \frac{n^i-1}{N-1}))$ and $n \in \{1, ..., N\}^d$, $1 \leq k \leq \log_2(d)$, $1 \leq i \leq \frac{d}{2^k}$. See Figure 7 for a diagram of how each partial product is assembled. Architecturally, this will be done in $\log(d) + 1$ transformer blocks. The first block will copy the product terms into the last $N^d d$ tokens. The remaining $\log(d)$ blocks compute the recursively assembled partial products by multiplying $p_{n,k-1,2i-1} \cdot p_{n,k-1,2i}$. Storage and assembly will be done in the last $d+Nd+1, ..., d+Nd+N^d d$ tokens.

For the first transformer block we define $\text{B}_2 = \mathcal{B}(N^d d, 0)$. We index each attention head as $A_{n,i}$ where $n \in \{1, ...., N\}^d$ and $1 \leq i \leq d$. Fix a patch $n$ and the corresponding product $\prod_{i=1}^d s_{i,n^i}$. These terms have been pre-computed and stored in tokens $h_{d+1}, ..., h_{d+Nd}$. We will then use each attention head $A_{n,i}$ to copy $s_{i,n^i}$ into its corresponding token $h_{n,i} = h_{d+Nd+d \sum_{p=1}^d (n^p-1)N^{d-p}+i}$. Concretely, for a fixed attention head $A_{n,i}$ define the data kernels

$$
Q_{n,i}^{data} = \begin{bmatrix} 0 & 0 & 0 & 0 & 1 \\ 0 & 0 & 0 & 0 & 0 \end{bmatrix} \quad K_{n,i}^{data} = \begin{bmatrix} 1 & 0 & 0 & 0 & 0 \\ 0 & 0 & 0 & 0 & 0 \end{bmatrix}
$$

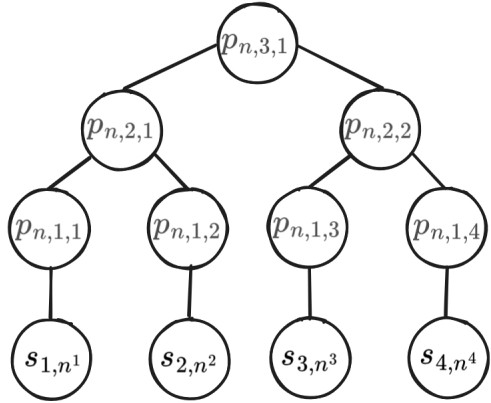

Figure 7: Recursive assembly of partial products from constituent terms. Formally, $p_{n,k,i} = p_{n,k-1,2i-1} \cdot p_{n,k-1,2i}$ with $p_{n,1,i} = s_{i,n^i}$ for $n \in \{1, ..., N\}^d$, $1 \leq k \leq \log_2(d)$, $1 \leq i \leq \frac{d}{2^k}$.

via the Interaction Lemma 3 we may pick $A_{n,i}$ so that

$$A_{n,i}h_{n,i} = \sum_{k=1}^{l} \sigma(\langle Q_{n,i}h_{n,i}, K_{n,i}h_k \rangle)V_{n,i}h_k$$

$$= \sum_{k=1}^{l} \sigma(h_{n,i}^5 h_k^1 + Q_{n,i}^I \mathcal{I}_{n,i} \cdot K_{n,i}^I I_k - C)e_1$$

$$= \sigma(1 \cdot s_{i,n^i})e_1 = s_{i,n^i}e_1$$

where the interaction terms zero-out all terms in the sum except when $k = d + id + n^i$ (which is the index of the token containing $s_{i,n^i}$). Similarly to as in B$_1$ we have $A_{n,i}h_k = 0$ when $k \neq d + Nd + \sum_{p=1}^{d}(n^p - 1)N^{d-p} + i$. Set the feed-forward layer FFN $= 0$. Then the next token embedding matrix can be written as

$$H_3 = \begin{bmatrix} x^1 & ... & x^d & s_{1,1} & ... & s_{d,N} & P_1 \\ 0 & ... & 0 & \mathbf{0}_{Nd} & ... & ... & \mathbf{0}_{N^d d} \\ \mathcal{I}_1 & ... & ... & ... & ... & ... & \mathcal{I}_{d+Nd+N^d d} \\ 1 & ... & ... & ... & ... & ... & 1 \end{bmatrix}$$

where $P_1 \in \mathbb{R}^{N^d d}$ is

$$P_1 = \begin{bmatrix} p_{\{1\}^d,1,1} & p_{\{1\}^d,2,2} & \cdots & p_{\{1\}^d,1,d} & p_{\{2\}\times\{1\}^{d-1},1,1} & \cdots & p_{\{N\}^d,1,d} \end{bmatrix}$$

$$= \begin{bmatrix} s_{1,1} & s_{2,1} & ... & s_{d,1} & s_{1,2} & ... & s_{d,N} \end{bmatrix}$$

where we recall $p_{\{1\}^d,1,1}$ denotes the first term in the base level product for for the patch $n = \{1, ..., 1\}$. Now define the next block B$_3 \in \mathcal{B}(N^d \frac{d}{2}, 1)$. Index the corresponding attention heads as $A_{n,i}$ for $n \in \{1, ..., N\}^d$, $1 \leq i \leq \frac{d}{2}$. Define data kernels

$$Q_{n,i}^{data} = \begin{bmatrix} 1 & 0 & 0 & 0 & 0 \\ 0 & 0 & 0 & 0 & 0 \end{bmatrix} \quad K_{n,i}^{data} = \begin{bmatrix} 1 & 0 & 0 & 0 & 0 \\ 0 & 0 & 0 & 0 & 0 \end{bmatrix}$$

where we choose the interaction terms so that the token containing $p_{n,1,2i-1}$ only interacts with the token containing $p_{n,1,2i}$ so that they multiply to form $p_{n,2,i}$ as shown in Figure 7. Note, in addition to computing the product, we must also subtract $p_{n,1,2i-1}$ from each corresponding token so that the output of the residual from B$_3$ cancels with the existing terms in the embedding matrix $H_3$. Write

$$A_{n,i}h_{n,i} = \sum_{k=1}^{l} \sigma(\langle Q_{n,i}h_{n,i}, K_{n,i}h_k \rangle)V_{n,i}h_k = \sigma(p_{n,1,2i-1}p_{n,1,2i})e_2 = \sigma(p_{n,2,i})e_2 = p_{n,2,i}e_2.$$

Adding the result onto $H_3$ gives the intermediate embedding matrix

$$H_{3.5} = \begin{bmatrix} x^1 & \cdots & x^d & s_{1,1} & \cdots & s_{d,N} & P_1 \\ 0 & \cdots & 0 & \mathbf{0}_{Nd} & \cdots & \cdots & P_{1.5} \\ \mathcal{I}_1 & \cdots & \cdots & \cdots & \cdots & \cdots & \mathcal{I}_{d+Nd+N^d d} \\ 1 & \cdots & \cdots & \cdots & \cdots & \cdots & 1 \end{bmatrix}$$

where $P_{1.5} \in \mathbb{R}^{N^d d}$ is

$$P_{1.5} = \begin{bmatrix} p_{\{1\}^d,2,1} & * & p_{\{1\}^d,2,2} & * & \cdots & p_{\{N\}^d,2,\frac{d}{2}} & * \end{bmatrix}.$$

Note: $*$ denotes a non-essential entry which can take any value. We then design our FFN network as $W_1 x$ where

$$W_1 = \begin{bmatrix} -1 & 1 & 0 & 0 & 0 \\ 0 & -1 & 0 & 0 & 0 \\ 0 & 0 & 0 & 0 & 0 \\ 0 & 0 & 0 & 0 & 0 \\ 0 & 0 & 0 & 0 & 0 \end{bmatrix}, \quad b_1 = 0$$

which when added to $H_{3.5}$ swaps the first component for the second component of each token and zeros out the second component. Then the next embedding matrix $H_4$ gives

$$H_4 = \begin{bmatrix} * & \cdots & P_2 \\ \mathbf{0}_{d+Nd+N^d d} & \cdots & 0 \\ \mathcal{I}_1 & \cdots & \mathcal{I}_{d+Nd+N^d d} \\ 1 & \cdots & 1 \end{bmatrix}$$

where $P_2 = P_{1.5} \in \mathbb{R}^{N^d d}$. More generally define the $k$th block implementing the $k$th level of the parallel product as $B_{k+2} = \mathcal{B}(N^d \frac{d}{2^k}, \mathcal{FFN}(1))$ with attention heads $A_{n,k,i}$ where $n \in \{1, ..., N\}^d$, $1 \le i \le \frac{d}{2^k}$. We can construct each attention head $A_{n,k,i}$ so that the token containing $p_{n,k-1,2i-1}$ only interacts with the token containing $p_{n,k-1,2i}$ to produce $p_{n,k,i}$:

$$A_{n,k,i} h_{n,i} = \sum_{t=1}^{l} \sigma(\langle Q_{n,i} h_{n,i}, K_{n,i} h_t \rangle) V_{n,i} h_t = \sigma(p_{n,k-1,2i-1} p_{n,k-1,2i}) e_2 = p_{n,k,i} e_2.$$

In the FFN set $W_1$ as before, yielding

$$H_{k+2} = \begin{bmatrix} * & \cdots & P_k \\ \mathbf{0}_{d+Nd+N^d d} & \cdots & 0 \\ \mathcal{I}_1 & \cdots & \mathcal{I}_{d+Nd+N^d d} \\ 1 & \cdots & 1 \end{bmatrix}$$

where

$$P_k = \begin{bmatrix} p_{\{1\}^d,k,1} & * & \cdots & p_{\{N\}^d,k,1} & * & \cdots \end{bmatrix}.$$

At $k = \log(d)$ we have

$$H_{\log(d)+2} = \begin{bmatrix} * & \cdots & P_{\log(d)} \\ \mathbf{0}_{d+Nd+N^d d} & \cdots & 0 \\ \mathcal{I}_1 & \cdots & \mathcal{I}_{d+Nd+N^d d} \\ 1 & \cdots & 1 \end{bmatrix}$$

where component $P_{\log(d)+2}^{d+Nd+\sum_{p=1}^{d}(n^p-1)N^p+i} = p_{n,\log(d),1} = \phi_n(x)$ as desired in (18). It simply now remains to multiply each $\phi_n(x)$ by the corresponding local average $f_n$ and sum the result. This can be implemented with a single attention block $B_{\log(d)+3} \in \mathcal{B}(N^d, 1)$. For $n \in \{1, ..., N\}^d$ define $A_n$ with the data kernels

$$Q_n = \begin{bmatrix} 1 & 0 & 0 & 0 & 0 \\ 0 & 0 & 0 & 0 & 1 \end{bmatrix} \quad K_n = \begin{bmatrix} 0 & 0 & 0 & 0 & f_n \\ 0 & 0 & 0 & 0 & R \end{bmatrix}$$

where $\|f\|_{L^\infty([0,1]^d)} \leq R$ and the token $h_{d+Nd+\sum_{p=1}^d (n^p-1)N^p+1} = h_{n,1}$ only interacts with itself so that

$$A_n h_{n,1} = \sum_{k=1}^l \sigma(h_{n,1}^1 \cdot f_n + R + Q_n^I \mathcal{I}_{n,1} \cdot K_n^I I_k - C)e_2$$

$$= \sigma(f_n \phi_n(x) + R)e_2 = (f_n \phi_n(x) + R)e_2$$

where the last equality comes from $|f_n \phi_n(x)| \leq |f_n| \leq R$. We implement FFN to subtract $R$ via

$$W_1 = \begin{bmatrix} 0 & 0 & 0 & 0 & 0 \\ 0 & 1 & 0 & 0 & -R \\ 0 & 0 & 0 & 0 & 0 \\ 0 & 0 & 0 & 0 & 0 \\ 0 & 0 & 0 & 0 & 0 \end{bmatrix}$$

so that

$$H_{\log(d)+4} = \begin{bmatrix} * & \cdots & \cdots \\ \mathbf{0}_{d+Nd} & \cdots & P_{\log(d)+4} \\ \mathcal{I}_1 & \cdots & \mathcal{I}_{d+Nd+N^d d} \\ 1 & \cdots & 1 \end{bmatrix}$$

where $P_{\log(d)+4}^{n,1} = f_n \phi_n$ for $n \in \{1, ..., N\}^d$ and is 0 elsewhere. To sum up the resulting terms we define the final transformer block $B_{\log(d)+4} \in \mathcal{B}(1,0)$ containing a single attention head $A_1$ as

$$Q = \begin{bmatrix} 0 & 0 & 0 & 0 & 1 \\ 0 & 0 & 0 & 0 & 0 \\ 0 & 0 & 0 & 0 & 0 \\ 0 & 0 & 0 & 0 & 0 \\ 0 & 0 & 0 & 0 & R \end{bmatrix} \quad K = \begin{bmatrix} 0 & 1 & 0 & 0 & 0 \\ 0 & 0 & 0 & 0 & 0 \\ 0 & 0 & 0 & 0 & 0 \\ 0 & 0 & 0 & 0 & 0 \\ 0 & 0 & 0 & 0 & 1 \end{bmatrix} \quad V = \begin{bmatrix} 0 & 0 & 0 & 0 & 1 \\ 0 & 0 & 0 & 0 & 0 \\ 0 & 0 & 0 & 0 & 0 \\ 0 & 0 & 0 & 0 & 0 \\ 0 & 0 & 0 & 0 & 0 \end{bmatrix}$$

which when applied to each token simply sums up the second components of all tokens. The result when applied to $h_t$ where $t = l$ gives

$$A_1 h_t = \sum_{k=1}^l \sigma(\langle Qh_t, Kh_k\rangle)Vh_k$$

$$= \sum_{k=1}^l \sigma(h_k^2 + R)e_1 = (lR + \sum_{n \in \{1,...,N\}^d} f_n \phi_n)e_1.$$

We can subtract $lR$ with the FFN as

$$W_1 = \begin{bmatrix} 1 & 0 & 0 & 0 & -lR \\ 0 & 0 & 0 & 0 & 0 \\ 0 & 0 & 0 & 0 & 0 \\ 0 & 0 & 0 & 0 & 0 \\ 0 & 0 & 0 & 0 & 0 \end{bmatrix}$$

yielding the final result $h_1^2 = \sum_n f_n \phi_n$. Applying the decoding head $D$ returns the desired result. This shows we can construct the approximation $\hat{f}$ exactly with an $O(\log(d))$ layer transformer.

$\square$

## E.2 Proof of Theorem 2

With Lemma 1, we can now move on to the proof of Theorem 2.

*Proof of Theorem 2.* Our goal is to approximate $f : \mathcal{M} \to \mathbb{R}$ in $L^\infty(\mathcal{M})$ with a transformer neural network. Our construction proceeds similarly to the construction in Chen et al. [2019] with feedforward neural networks.

First, we will decompose $f(x)$ as a sum of terms over local neighborhoods $U_1, ..., U_{C_{\mathcal{M}}} \subseteq \mathcal{M}$ covering $\mathcal{M}$. Approximations on overlapping neighborhoods containing $x$ will then be combined via a partition of unity (**PoU**) $\{\rho_n\}_{n=1}^{C_{\mathcal{M}}}$ which subordinates $\{U_n\}_{n=1}^{C_{\mathcal{M}}}$, while contributions from neighborhoods not containing $x$ will be zeroed-out via an indicator $\mathbf{1}_{U_n}$. This will give us the expression

$$f(x) = \sum_{n=1}^{C_{\mathcal{M}}} f_n(x)\mathbf{1}_{U_n}(x) \tag{19}$$

where $f_n : \mathcal{M} \to \mathbb{R}$ agrees with $f$ on $U_n$. Next, on each local neighborhood we will project the input $x \in \mathcal{M} \subseteq \mathbb{R}^D$ to $[0,1]^d$. This will give us the following *local decomposition* of the target function:

$$f(x) = \sum_{n=1}^{C_{\mathcal{M}}} \tilde{f}_n \circ \phi_n(x)\mathbf{1}_{U_n}(x)$$

where $\tilde{f}_n : \mathbb{R}^d \to \mathbb{R}$ and $\phi_n : \mathcal{M} \to \mathbb{R}^d$ is a projection. This step is critical: we now must only approximate a low-dimensional function $\tilde{f}_n$ and a simple projection $\phi_n$ instead of the high-dimensional function $f_n$. After this, we must simply approximate each $\tilde{f}_n, \phi_n, \mathbf{1}_{U_n}$ using transformer neural networks.

**Step 1: Local Decomposition**    Denote the open Euclidean ball with center $c$ and radius $r$ in $\mathbb{R}^D$ by $B(c,r)$. Given a manifold $\mathcal{M}$ satisfying Assumption 1, the collection of open balls $\{B(c,r)\}_{c \in \mathcal{M}}$ with some $r \leq \tau/4$ is an open cover of $\mathcal{M}$. Since $\mathcal{M}$ is compact, there are a finite number of points, denoted by $c_n$, for $n = 1, \ldots, C_{\mathcal{M}}$ such that $\mathcal{M} \subset \cup_{n=1}^{C_{\mathcal{M}}} B(c_n, r)$. We can then decompose $\mathcal{M}$ into $C_{\mathcal{M}}$ charts $(U_n, \phi_n)_{n=1}^{C_{\mathcal{M}}}$ where $U_n = \mathcal{M} \cap B(c_n, r)$ and each $\phi_n : U_n \to \mathbb{R}^d$ is an orthogonal projection of the local neighborhood $U_n$ onto the local tangent space $T_{c_n}(\mathcal{M})$ . Via Lemma 8, as long as $r \leq \tau/4$, each $\phi_n$ is a diffeomorphism between $U_n$ and a subset in $T_{c_n}(\mathcal{M})$. Here the number of charts $C_{\mathcal{M}}$ is a constant depending on the complexity of the manifold $\mathcal{M}$. We can bound it as $C_{\mathcal{M}} \lesssim \frac{SA(\mathcal{M})}{r^d} d \log(d)$ where $SA(\mathcal{M})$ is the surface area of $\mathcal{M}$ [Conway and Sloane, 1988].

For the open covering $\{U_n\}_{n=1}^{C_{\mathcal{M}}}$, there exists a smooth partition of unity $\{\rho_n\}_{n=1}^{C_{\mathcal{M}}}$ such that $f(x) = \sum_{n=1}^{C_{\mathcal{M}}} \rho_n(x)f(x)$ for $x \in \mathcal{M}$ [Tu, 2011]. Each $\rho_n$ is supported only in its corresponding chart $U_n$. We additionally include an indicator function $\mathbf{1}_{U_n}$ for each neighborhood $U_n$, resulting the decomposition

$$f(x) = \sum_{n=1}^{C_{\mathcal{M}}} \underbrace{f(x)\rho_n(x)}_{f_n(x)} \mathbf{1}_{U_n}(x).$$

We then aim to approximate each term in the sum on its corresponding local neighborhood $U_n$ and then sum the result. Because we are only locally approximating each $\rho_n$, the indicator is critical in the approximation as it will be used to suppress contributions from $U_n$ such that the input $x \notin U_n$.

Fix a local neighborhood $U_n$. We will now project the input $x \in \mathcal{M} \subseteq \mathbb{R}^D$ to the local tangent space $T_{c_n}(\mathcal{M}) \subseteq [0,1]^d$. We can write

$$f_n(x) = (f_n \circ \phi_n^{-1}) \circ \phi_n(x) = \tilde{f}_n \circ \phi_n(x)$$

where $\tilde{f}_n : \mathbb{R}^d \to \mathbb{R}$ and again $\phi_n : \mathcal{M} \subseteq \mathbb{R}^D \to \mathbb{R}^d$ is an orthogonal projection. Then we can write $f$ as

$$f(x) = \sum_{n=1}^{C_{\mathcal{M}}} \tilde{f}_n \circ \phi_n(x)\mathbf{1}_{U_n}(x). \tag{20}$$

**Step 2: Local Approximation**    In (20), we have rewritten $f$ as a sum of functions which can be approximated locally on each $U_n$ and indicators $\mathbf{1}_{U_n}$. We will approximate each component with transformer neural networks. Specifically, let $\mathrm{T}_{\tilde{f}_n}$ approximate $\tilde{f}_n$, $\mathrm{T}_{\phi_n}$ approximate $\phi_n$, and $\mathrm{T}_{\mathbf{1}_{U_n}}$ approximate $\mathbf{1}_{U_n}$, $1 \leq n \leq C_{\mathcal{M}}$. Then we will have the overall approximation

$$f(x) \approx \mathrm{T}_f(x) = \sum_{n=1}^{C_{\mathcal{M}}} \mathrm{T}_{\tilde{f}_n} \circ \mathrm{T}_{\phi_n}(x)\mathrm{T}_{\mathbf{1}_{U_n}}(x).$$

This yields the error decomposition

$$\|f - \mathbf{T}_f\|_{L^\infty(\mathcal{M})} = \sum_{n=1}^{C_\mathcal{M}} \|\tilde{f}_n \circ \phi_n \mathbf{1}_{U_n} - \mathbf{T}_{\tilde{f}_n} \circ \mathbf{T}_{\phi_n} \mathbf{T}_{\mathbf{1}_{U_n}}\|_{L^\infty(\mathcal{M})}$$

$$\leq \sum_{n=1}^{C_\mathcal{M}} \|\tilde{f}_n \circ \phi_n \mathbf{T}_{\mathbf{1}_{U_n}} - \mathbf{T}_{\tilde{f}_n} \circ \mathbf{T}_{\phi_n} \mathbf{T}_{\mathbf{1}_{U_n}}\|_{L^\infty(\mathcal{M})} + \|\tilde{f}_n \circ \phi_n \mathbf{1}_{U_n} - \tilde{f}_n \circ \phi_n \mathbf{T}_{\mathbf{1}_{U_n}}\|_{L^\infty(\mathcal{M})}$$

$$\leq \sum_{n=1}^{C_\mathcal{M}} \|\tilde{f}_n \circ \phi_n - \mathbf{T}_{\tilde{f}_n} \circ \mathbf{T}_{\phi_n}\|_{L^\infty(U_n)} + E_{n,3}$$

$$\leq \sum_{n=1}^{C_\mathcal{M}} \|\tilde{f}_n \circ \phi_n - \mathbf{T}_{\tilde{f}_n} \circ \phi_n\|_{L^\infty(U_n)} + \|\mathbf{T}_{\tilde{f}_n} \circ \phi_n - \mathbf{T}_{\tilde{f}_n} \circ \mathbf{T}_{\phi_n}\|_{L^\infty(U_n)} + E_{n,3}$$

$$= \sum_{n=1}^{C_\mathcal{M}} E_{n,1} + E_{n,2} + E_{n,3}$$

where the error terms $E_{n,1}, E_{n,2}, E_{n,3}$ denote

$$E_{n,1} := \|\tilde{f}_n \circ \phi_n - \mathbf{T}_{\tilde{f}_n} \circ \phi_n\|_{L^\infty(U_n)} \tag{21}$$

$$E_{n,2} := \|\mathbf{T}_{\tilde{f}_n} \circ \phi_n - \mathbf{T}_{\tilde{f}_n} \circ \mathbf{T}_{\phi_n}\|_{L^\infty(U_n)} \tag{22}$$

$$E_{n,3} := \|\tilde{f}_n \circ \phi_n \mathbf{1}_{U_n} - \tilde{f}_n \circ \phi_n \mathbf{T}_{\mathbf{1}_{U_n}}\|_{L^\infty(\mathcal{M})} \tag{23}$$

respectively. It remains to construct transformer approximations for each term.

**Step 2.1: Approximating $\tilde{f}_n$ and Bounding $E_{n,1}$**  We first handle the $E_{n,1}$ error in (21). Fix $1 \leq n \leq C_\mathcal{M}$. We can write

$$\|\tilde{f}_n \circ \phi_n - \mathbf{T}_{\tilde{f}_n} \circ \phi_n\|_{L^\infty(U_n)} \leq \|\tilde{f}_n - \mathbf{T}_{\tilde{f}_n}\|_{L^\infty([0,1]^d)}$$

so it suffices to approximate $\tilde{f}_n : [0,1]^d \to \mathbb{R}$. Now choose the desired accuracy $\delta_{n,1} > 0$. Via Lemma 1 we can construct an approximation such that

$$\|\tilde{f}_n - \mathbf{T}_{\tilde{f}_n}\|_{L^\infty([0,1]^d)} < \delta_{n,1}.$$

This construction $\mathbf{T}_{\tilde{f}_n}$ has $O(\log(d))$ transformer block layers, $O(1)$ feed-forward layers, $O(d\delta_{n,1}^{-\frac{d}{\beta}})$ tokens, $O(1)$ embedding dimension, $O(d\delta_{n,1}^{-\frac{d}{\beta}})$ attention heads per-layer, and weights with magnitude at most $O(d\delta_{n,1}^{-\frac{2d}{\beta}})$. Via the Parallelization Lemma 7 we can compute each approximation of $\tilde{f}_n$ in parallel using a transformer with $O(C_\mathcal{M} d\delta_1^{-\frac{d}{\beta}})$ attention heads and $O(C_\mathcal{M} d_{embd})$ FFN width, where we set $\delta_1 = \delta_{1,1} = ... = \delta_{C_\mathcal{M},1}$, and $O(\log(d))$ layers.

**Step 2.2: Approximating $\phi_n$ Exactly Such That $E_{n,2} = 0$**  Fix $1 \leq n \leq C_\mathcal{M}$. We must now control the $E_{n,2}$ error in (22):

$$E_{n,2} = \|\mathbf{T}_{\tilde{f}_n} \circ \phi_n - \mathbf{T}_{\tilde{f}_n} \circ \mathbf{T}_{\phi_n}\|_{L^\infty(U_n)}$$

We will implement $\phi_n$ using a transformer block $\mathbf{B} \in \mathcal{B}(dD, 5)$. We can identify the tangent space at the point $c_n$ as

$$T_{c_n}(\mathcal{M}) = \text{span}(v_{n,1}, ..., v_{n,d})$$

where the vectors $v_{n,i} \in \mathbb{R}^D$, $1 \leq i \leq d$, form an orthonormal basis of $T_{c_n}(\mathcal{M})$. Then we can write the projection map $\phi_n$ as

$$\phi_n(x) = s_n(V_n^T(x - c_n) + u_n)$$

where $s_n \in (0,1]$ is a scaling factor, $u_n \in \mathbb{R}^d$ is a translation, and $V_n = [v_{n,1} \quad ... \quad v_{n,d}] \in \mathbb{R}^{D \times d}$ so that $\phi_n(x) \in [0,1]^d$. Suppose we have an input hidden embedding matrix of the form

$$H = \begin{bmatrix} x^1 & ... & x^D & \mathbf{0}_d \\ 0 & ... & ... & 0 \\ \mathcal{I}_1 & ... & ... & \mathcal{I}_l \\ 1 & ... & ... & 1 \end{bmatrix} \in \mathbb{R}^{d_{embd} \times l}$$

We will implement the operation $v_{n,i}^j \cdot (x^j - c_n^j)$ via an attention head $A_{i,j}$, $1 \le i \le d$, $1 \le j \le D$. Define the data kernels

$$Q_i^{data} = \begin{bmatrix} 0 & 0 & 0 & 0 & 1 \\ 0 & 0 & 0 & 0 & 1 \end{bmatrix} \quad K_i^{data} = \begin{bmatrix} v_{n,i}^j & 0 & 0 & 0 & -v_{n,i}^j c_n^j \\ 0 & 0 & 0 & 0 & 2M \end{bmatrix}$$

via the Interaction Lemma 3 we can define $A_{i,j}$ so that token $h_{D+i}$ interacts with token $h_j$ so that

$$\begin{aligned} A_{i,j}(h_{D+i}) &= \sigma(\langle Q_i^{data} h_{D+i}, K_i^{data} h_j \rangle) \\ &= \sigma(v_{n,i}^j x^j - v_{n,i}^j c_n^j + 2M) \\ &= \sigma(v_{n,i}^j(x^j - c_n^j) + 2M) \\ &= (v_{n,i}^j(x^j - c_n^j) + 2M)e_1 \end{aligned}$$

and is 0 on other tokens $h_t \ne h_{D+i}$, where $M$ bounds $x \in \mathcal{M}$ is such that $v_{n,i}^j(x^j - c_n^j) + 2M \ge 0$ for all $n, i, j$. Then the output of the multi-headed attention MHA on $h_{D+i}$ is

$$\begin{aligned} \text{MHA}(h_{D+i}) &= \sum_{\substack{1 \le k \le d \\ 1 \le j \le D}} A_{k,j} h_{D+i} = \sum_{j=1}^D A_{i,j} h_{D+i} \\ &= \sum_{j=1}^D v_{n,i}^j(x^j - c_n^j)e_1 + 2Me_1 = \langle v_{n,i}, x - c_n \rangle e_1 + 2DMe_1 \end{aligned}$$

This yields the intermediate embedding matrix

$$H + \text{MHA}(H) = \begin{bmatrix} x^1 & \dots & x^D & \langle v_{n,1}, x - c_n \rangle + 2DM & \dots & \langle v_{n,d}, x - c_n \rangle + 2DM \\ 0 & \dots & \dots & \dots & \dots & 0 \\ \mathcal{I}_1 & \dots & \dots & \dots & \dots & \mathcal{I}_l \\ 1 & \dots & \dots & \dots & \dots & 1 \end{bmatrix}$$

We can then design a five layer FFN layer to subtract $2DM$ and then add $u_n$ and multiply $s_n$ only from the embedding tokens $D + 1, ..., D + d$. The first layer will simply subsract $2DM$ and then add $u_n$ and multiply $s_n$ from each token. This can be implemented by the matrix

$$W_1 = \begin{bmatrix} s_n & 0 & 0 & 0 & -2DMs_n + u_n s_n \\ 0 & 0 & 0 & 0 & 0 \\ 0 & 0 & 1 & 0 & 0 \\ 0 & 0 & 0 & 1 & 0 \\ 0 & 0 & 0 & 0 & 1 \end{bmatrix}$$

Then, via the gating Lemma 4 we can construct a two-layer feed-forward network $\text{FFN}_{gating}$ such that, for a chosen $1 \le t_u \le l$, we have $\text{FFN}_{gating}(h_k) = h_k$ when $k < t_u$ and $\text{FFN}_{gating}(h_k)$ is zero except for the last three rows for $k \ge t_u$. We can again invoke the lemma to construct another FFN layer so that for $k \ge t_l \implies \text{FFN}_{gating}(h_k) = h_k$ and $k < t_l \implies \text{FFN}_{gating}(h_k)$ is zero except for the last three rows. So, applying Lemma 4 twice, we can construct a four-layer feed-forward network $\text{FFN}_{gating}$ such that $\text{FFN}_{gating}(h_t) = h_t$ for $D + 1 \le D + d$ and is zero except for the last three rows otherwise. This yields the desired output.

$$H' = \text{B}(H) = \begin{bmatrix} x^1 & \dots & x^D & s_n(\langle v_{n,1}, x - c_n \rangle + u_n) & \dots & s_n(\langle v_{n,d}, x - c_n \rangle + u_n) \\ 0 & \dots & \dots & \dots & \dots & 0 \\ \mathcal{I}_1 & \dots & \dots & \dots & \dots & \mathcal{I}_l \\ 1 & \dots & \dots & \dots & \dots & 1 \end{bmatrix}$$

Further, this approximation is **exact**, so that $E_{n,2} = 0$. We can then compute each $\phi_n$ in parallel via the Parallelization Lemma 7 with a transformer having $O(C_{\mathcal{M}} dD)$ attention heads and $O(C_{\mathcal{M}} d_{embd})$ FFN width.

**Step 2.3: Approximating $\mathbf{1}_{U_n}(x)$ and Bounding $E_{n,3}$**     Again fix $1 \le n \le C_{\mathcal{M}}$. By construction we can write

$$\mathbf{1}_{U_n}(x) = \begin{cases} 1 & \|x - c_n\|_2^2 < r^2 \\ 0 & \|x - c_n\|_2^2 \ge r^2 \end{cases}$$

Suppose we are given an input embedding matrix of the form

$$H_1 = \begin{bmatrix} x^1 & \dots & x^D & \mathbf{0}_D \\ 0 & \dots & \dots & 0 \\ \mathcal{I}_1 & \dots & \dots & \mathcal{I}_l \\ 1 & \dots & \dots & 1 \end{bmatrix} \in \mathbb{R}^{d_{embd} \times 2D}$$

We must start by computing $\|x - c_n\|_2^2$. Via the Addition Lemma 5 we can construct a transformer block $B_1 \in \mathcal{B}(D, 3)$ such that

$$H_2 = B_1(H_1) = \begin{bmatrix} x^1 & \dots & x^D & x^1 - c_n^1 & \dots & x^D - c_n^D \\ 0 & \dots & \dots & \dots & \dots & 0 \\ \mathcal{I}_1 & \dots & \dots & \dots & \dots & I\mathcal{I}_l \\ 1 & \dots & \dots & \dots & \dots & 1 \end{bmatrix}$$

We now construct transformer block $B_2 \in \mathcal{B}(D, 2)$ squares each term in the sum. For $1 \le i \le D$ define the data kernels

$$Q_i^{data} = \begin{bmatrix} 1 & 0 & 0 & 0 & 0 \\ 0 & 0 & 0 & 0 & 0 \end{bmatrix} \quad K_i^{data} = \begin{bmatrix} 1 & 0 & 0 & 0 & 0 \\ 0 & 0 & 0 & 0 & 0 \end{bmatrix}$$

Then via the Interaction Lemma 3 we can construct $A_i$ so that the token $h_{D+i}$ interacts with itself such that

$$A_i(h_{D+i}) = \sigma((x^i - c_n^i)^2)e_2 = (x^i - c_n^i)^2 e_2$$

and 0 otherwise. The output of the resulting multi-headed attention gives

$$H_2' = \begin{bmatrix} x^1 & \dots & x^D & x^1 - c_n^1 & \dots & x^D - c_n^D \\ 0 & \dots & 0 & (x^1 - c_n^1)^2 & \dots & (x^D - c_n^D)^2 \\ \mathcal{I}_1 & \dots & \dots & \dots & \dots & I_l \\ 1 & \dots & \dots & \dots & \dots & 1 \end{bmatrix}$$

Via Lemmas 6 and 4 we can construct FFN $\in \mathcal{FFN}(5)$ which replaces the first row with the second row for $h_{D+1}, \dots, h_{2D}$. This gives the output

$$H_3 = B_2(H_2) = \begin{bmatrix} x^1 & \dots & x^D & (x^1 - c_n^1)^2 & \dots & (x^D - c_n^D)^2 \\ 0 & \dots & \dots & & \dots & 0 \\ \mathcal{I}_1 & \dots & \dots & \dots & \dots & I_l \\ 1 & \dots & \dots & \dots & \dots & 1 \end{bmatrix}$$

and it remains only to take the sum. This can be done with $B_3 \in \mathcal{B}(D - 1, 0)$. For $1 \le i \le D - 1$ define the data kernels

$$Q_i^{data} = \begin{bmatrix} 0 & 0 & 0 & 0 & 1 \\ 0 & 0 & 0 & 0 & 0 \end{bmatrix} \quad K_i^{data} = \begin{bmatrix} 1 & 0 & 0 & 0 & 0 \\ 0 & 0 & 0 & 0 & 0 \end{bmatrix}$$

via the Interaction Lemma 3 we construct $A_i$ so that $h_{D+1}$ interacts only with $h_{D+1+i}$ such that

$$A_i(h_{D+1}) = \sigma(\langle Q_i^{data} h_{D+1}, K_i^{data} h_{D+1+i} \rangle)e_1 = \sigma((x^{i+1} - c_n^{i+1})^2)e_1 = (x^{i+1} - c_n^{i+1})^2 e_1$$

The output of the multi-headed attention is then

$$\text{MHA}(h_{D+1}) = \sum_{i=1}^{D-1} A_i(h_{D+1}) = \sum_{i=1}^{D-1} (x^{i+1} - c_n^{i+1})e_1$$

and 0 otherwise. This gives the intermediate output

$$H_3' = \begin{bmatrix} x^1 & \dots & x^D & \sum_{i=1}^{D}(x^i - c_n^i)^2 & \dots & (x^D - c_n^D)^2 \\ 0 & \dots & \dots & \dots & \dots & 0 \\ \mathcal{I}_1 & \dots & \dots & \dots & \dots & I_l \\ 1 & \dots & \dots & \dots & \dots & 1 \end{bmatrix}$$

where in particular $h_{D+1}^1 = \|x - c_n\|_2^2$. It remains to approximate the indicator

$$\mathbf{1}_{r^2}(s) = \begin{cases} 1 & s < r^2 \\ 0 & s \ge r^2 \end{cases}$$

**2.3a: Approximating $\mathbf{1}_{r^2}(s)$**    Recall our goal is to approximate the function $f : \mathcal{M} \to \mathbb{R}$ in $L^\infty$. However, we cannot approximate $\mathbf{1}_{r^2}$ in $L^\infty$ as it is a discontinuous function. To address this issue, recall the error term corresponding to the approximation of $\mathbf{1}_{U_n}$ is $E_{n,3} = \|\tilde{f}_n \circ \phi_n \mathbf{1}_{U_n} - \tilde{f}_n \circ \phi_n \mathbf{T}_{\mathbf{1}_{U_n}}\|_{L^\infty(\mathcal{M})}$. We know on the boundary of $U_n$ that $\tilde{f}_n \circ \phi_n = f_n = f\rho_n$ must be 0 as $\rho_n$ is 0 and further $f\rho_n$ is $\beta$-Hölder continuous as $f$ is $\beta$-Hölder continuous and $\rho_n$ is smooth. This suggests the following approach. Define the approximating indicator

$$
\hat{\mathbf{1}}_{r^2,\Delta}(s) = \begin{cases} 1 & s \le r^2 - \Delta \\ 1 - \frac{1}{\Delta}s + \frac{r^2 - \Delta}{\Delta} & r^2 - \Delta < s < r^2 \\ 0 & s \ge r^2 \end{cases}
$$

for some $\Delta > 0$. Clearly for $s \le r^2 - \Delta$ and $s \ge r^2$ we have $\hat{\mathbf{1}}_{r^2,\Delta}(s) = \mathbf{1}_{r^2}(s)$. So we argue $\tilde{f}_n$ is small for $x \in \mathcal{M}$ in the range $\mathcal{K}_n = \{x \in \mathcal{M} : r^2 - \Delta < \|x - c_n\|_2^2 < r^2\}$. This is handled by Lemma 8 in Chen et al. [2022] which gives us the following bound on $E_{n,3}$:

$$
E_{n,3} \le \frac{c}{r}\Delta
$$

for some constant $c$. So it suffices to make $\Delta$ small. It then remains to implement $\hat{\mathbf{1}}_{r^2,\Delta}$. We will do so using a feed-forward layer $\text{FFN} \in \mathcal{FFN}(O(\log(\frac{1}{\Delta})))$. Note that we can write the target exactly as

$$
\hat{\mathbf{1}}_{r^2,\Delta}(s) = \sigma\left(1 - \sigma\left(\frac{1}{\Delta}(s - (r^2 - \Delta))\right)\right)
$$

while is realized by a two-layer ReLU network. However, the weight $\frac{1}{\Delta}$ will scale unboundedly with the desired accuracy $\epsilon$. So we must deepen the FFN to utilize a logarithmic number of layers $O(\log(\frac{1}{\Delta}))$ as

$$
\hat{\mathbf{1}}_{r^2,\Delta}(s) = \sigma\left(1 - \sigma\left(\left(\frac{1}{\Delta}\right)^{\frac{1}{\log(\frac{1}{\Delta})}} \circ \sigma ... \circ \sigma\left(\left(\frac{1}{\Delta}\right)^{\frac{1}{\log(\frac{1}{\Delta})}}(s - (r^2 - \Delta)))...\right)\right)\right)
$$

which prevents the weights $(\frac{1}{\Delta})^{\frac{1}{\log(\frac{1}{\Delta})}}$ from blowing up. We can then approximate each $\mathbf{1}_{U_n}$ in parallel using a transformer with $O(C_\mathcal{M}D)$ and three layers.

**Step 3: Bringing it all together**    Recall our goal is to approximate $f$ on $\mathcal{M}$ by writing $f$ as

$$
f(x) = \sum_{n=1}^{C_\mathcal{M}} \tilde{f}_n \circ \phi_n(x) \mathbf{1}_{U_n}(x).
$$

We argued we can successfully approximate

1. $\tilde{f}_n$ up to $\delta_{n,1}$ accuracy in $L^\infty$ via a transformer neural network with $O(d\delta_{n,1}^{-\frac{d}{\beta}})$ width and $O(\log(d))$ depth. All approximations can be computed in parallel via a transformer with $O(C_\mathcal{M}d\delta_{n,1}^{-\frac{d}{\beta}})$ attention heads and $O(\log(d))$ layers.

2. Each $\phi_n$ exactly via a transformer neural network with $O(dD)$ width and constant depth. All the $\phi_n$ can be computed in parallel via a transformer with $O(C_\mathcal{M}dD)$ heads and constant depth.

3. Each $\mathbf{1}_{U_n}$ up to $\delta_{n,3}$ accuracy in $L^\infty$ via a transformer with $O(D)$ width and a feed-forward layer with $O(\log(\frac{1}{\Delta}))$ depth. All $\mathbf{1}_{U_n}$ can be approximated in parallel via a transformer with $O(C_\mathcal{M}D)$ attention heads and a constant number of transformer blocks.

Via the Parallelization Lemma 7 we can compute the approximations of $\mathbf{1}_{U_n}$ and $\tilde{f}_n \circ \phi_n$ in parallel via a transformer with $O(C_\mathcal{M}(d\delta_{n,1}^{-d} + dD))$ attention heads, $O(C_\mathcal{M}d_{embd})$ FFN width, and $O(\log(d))$

transformer blocks. As a result will have the embedding matrix

$$
H = \begin{bmatrix}
x^1 & \dots & x^D & c_{f_1} & \dots & c_{f_{C_\mathcal{M}}} & c_{U_1} & \dots & c_{U_{C_\mathcal{M}}} & * \\
0 & \dots & \dots & \dots & \dots & \dots & \dots & \dots & \dots & 0 \\
\mathcal{I}_1 & \dots & \dots & \dots & \dots & \dots & \dots & \dots & \dots & \mathcal{I}_L \\
1 & \dots & \dots & \dots & \dots & \dots & \dots & \dots & \dots & 1
\end{bmatrix}
$$

where each $c_{f_n} = \tilde{f}_n \circ \phi_n(x)$ and $c_{U_n} = \mathbf{1}_{U_n}(x)$. We implement the final sum via two transformer blocks $B_1, B_2$. First define $B_1 \in \mathcal{B}(C_\mathcal{M}, 1)$ with attention heads $A_i, 1 \le i \le C_\mathcal{M}$ with data kernels

$$
Q_i^{data} = \begin{bmatrix} 1 & 0 & 0 & 0 & 0 \\ -1 & 0 & 0 & 0 & M \end{bmatrix} \quad K_i^{data} = \begin{bmatrix} 1 & 0 & 0 & 0 & 0 \\ 0 & 0 & 0 & 0 & 1 \end{bmatrix}
$$

with interaction kernels chosen so that $h_{D+i}$ only interacts with $h_{D+C_\mathcal{M}+i}$ under $A_i$. Then

$$
A_i(h_{D+i}) = \sigma(c_{f_i} c_{U_i} - c_{f_i} + M)e_1 = (c_{f_i} c_{U_i} - c_{f_i} + M)e_1
$$

We can then design a two-layer FFN to subtract $M$ from columns $D+1, ..., D+C_\mathcal{M}$ yielding the output embedding matrix

$$
H = \begin{bmatrix}
x^1 & \dots & x^D & c_{f_1} c_{U_1} & \dots & c_{f_{C_\mathcal{M}}} c_{U_{C_\mathcal{M}}} & * \\
0 & \dots & \dots & \dots & \dots & \dots & 0 \\
\mathcal{I}_1 & \dots & \dots & \dots & \dots & \dots & \mathcal{I}_L \\
1 & \dots & \dots & \dots & \dots & \dots & 1
\end{bmatrix}
$$

We can sum up the result with another transformer block $B_2 \in \mathcal{B}(C_\mathcal{M}, 2)$ and storing the desired result in the first component of the last token $t = l$. Applying the decoding layer $D$ yields the desired result.

It simply remains to control the errors. Recall we have the error bound

$$
\|f - T_f\|_{L^\infty(\mathcal{M})} \le \sum_{n=1}^{C_\mathcal{M}} E_{n,1} + E_{n,2} + E_{n,3}
$$

$$
\le \sum_{n=1}^{C_\mathcal{M}} \delta_{n,1} + \delta_{n,3} = C_\mathcal{M}\delta_{n,1} + C_\mathcal{M}\delta_{n,3}
$$

Then for a target accuracy $\epsilon$ we choose $\delta_{n,1} = \frac{\epsilon}{2C_\mathcal{M}}$ and $\delta_{n,3} = \frac{\epsilon}{2C_\mathcal{M}}$ so that

$$
\|f - T_f\|_{L^\infty(\mathcal{M})} \le C_\mathcal{M}\delta_{n,1} + C_\mathcal{M}\delta_{n,3} = C_\mathcal{M}\frac{\epsilon}{2C_\mathcal{M}} + C_\mathcal{M}\frac{\epsilon}{2C_\mathcal{M}} = \epsilon
$$

This completes the proof of Theorem 2.

$\square$

The above result concludes our approximation theory results. To summarize, we first showed a (sufficiently smooth) function $f : [0,1]^d \to \mathbb{R}$ can be approximated up to $\epsilon$ accuracy in $L^\infty$ with a transformer neural network with $O(d\epsilon^{-d})$ width and $O(\log(d))$ depth in Lemma 1. We then used this construction to show a function $f : \mathcal{M} \to \mathbb{R}$ on a compact manifold $\mathcal{M}$ with intrinsic dimension $d$ can be approximated up to $\epsilon$ accuracy in $L_1$ with a transformer neural network with width $O(\max(dD, d\epsilon^{-d}))$ and depth $O(\log(d))$.

# F  Proof of Transformer Generalization Theory

We next prove Theorem 1 in Appendix F.1. Much of this follows the same argument as in Chen et al. [2022] which focused on feedforward neural networks. The main novelty of this paper is a bound on the covering number of $\mathcal{T}(L_T, L_{\text{FFN}}, w_{\text{FFN}}, l, d_{embd}, m, R, \kappa)$ given in Lemma 2, which is proved in Appendix F.2.

## F.1 Proof of Theorem 1

*Proof of Theorem 1.* Suppose we have $x_1, ..., x_n \sim Q$ as i.i.d. training samples drawn from $Q$ on the manifold $\mathcal{M}$ and their evaluations $f(x_1), ..., f(x_n)$. Set the approximating function class as $\mathcal{T}(L_T, L_{\text{FFN}}, w_{\text{FFN}}, l, d_{embd}, m, R, \kappa)$. We define the transformer empirical risk minimizer $\hat{\text{T}}_n$ in (1) as the transformer network which minimizes the empirical $L^2$ training loss. Our goal is to bound the squared generalization error in (6).

We first rewrite the squared generalization error by adding and subtracting (twice) the training objective where we note a factor of two is used to accelerate convergence of the statistical error.

$$\mathbb{E} \int_{\mathcal{M}} (\hat{\text{T}}_n(x) - f(x))^2 dQ$$

$$= 2\mathbb{E}\frac{1}{n}\sum_{i=1}^{n}(\hat{\text{T}}_n(x_i) - f(x_i))^2 + \mathbb{E}\int_{\mathcal{M}}(\hat{\text{T}}_n(x) - f(x))^2 dQ - 2\mathbb{E}\frac{1}{n}\sum_{i=1}^{n}(\hat{\text{T}}_n(x_i) - f(x_i))^2. \quad (24)$$

The first (bias) term in (24) can be controlled via Theorem 2:

$$\mathbb{E}\frac{1}{n}\sum_{i=1}^{n}(\hat{\text{T}}_n(x_i) - f(x_i))^2 = \mathbb{E}\inf_{\text{T}\in\mathcal{T}}\frac{1}{n}\sum_{i=1}^{n}(\text{T}(x_i) - f(x_i))^2$$

$$\leq \inf_{\text{T}\in\mathcal{T}}\mathbb{E}\frac{1}{n}\sum_{i=1}^{n}(\text{T}(x_i) - f(x_i))^2$$

$$= \inf_{\text{T}\in\mathcal{T}}\int_{\mathcal{M}}\left(\text{T}(x) - f(x)\right)^2 dQ$$

$$\leq \inf_{\text{T}\in\mathcal{T}}\int_{\mathcal{M}}\|\text{T} - f\|_{L^\infty(\mathcal{M})}^2 dQ < \epsilon^2$$

where the last line follows from Theorem 2. Note, we can pass the expectation inside the infimum via Jensen's inequality. Set $d_T^2(x) = (\text{T}(x) - f(x))^2$.

It remains to control the last two (variance) terms in (24). We will do so by controlling the covering number of the approximating function class $\mathcal{T}(L_T, L_{\text{FFN}}, w_{\text{FFN}}, l, d_{embd}, m, R, \kappa)$. Via Lemma 6 from Chen et al. [2022] we have the bound

$$\mathbb{E}\int_{\mathcal{M}}(\hat{\text{T}}_n(x) - f(x))^2 dQ - 2\mathbb{E}\frac{1}{n}\sum_{i=1}^{n}(\hat{\text{T}}_n(x_i) - f(x_i))^2$$

$$\leq \inf_{\delta > 0}\left[\frac{104R^2}{3n}\log\mathcal{N}\left(\frac{\delta}{4R}, \mathcal{T}, \|\cdot\|_\infty\right) + (4 + \frac{1}{2R})\delta\right] \quad (25)$$

where $\mathcal{N}(\frac{\delta}{4R}, \mathcal{T}, \|\cdot\|_\infty)$ denotes of the covering number of the network class $\mathcal{T}$ under the $L^\infty$ norm. Our goal is now to bound this covering number of $\mathcal{T}(L_T, L_{\text{FFN}}, w_{\text{FFN}}, l, d_{embd}, m, R, \kappa)$. We do so with the following lemma:

**Lemma 2.** *Consider a transformer neural network class $\mathcal{T} = \mathcal{T}(L_T, L_{\text{FFN}}, w_{\text{FFN}}, l, d_{embd}, m, R, \kappa)$ with input $x \in \mathbb{R}^D$ satisfying $\|x\|_\infty \leq M$.. Let $\delta > 0$. Then*

$$\mathcal{N}(\delta, \mathcal{T}, \|\cdot\|_\infty) \leq \left(\frac{2^{L_T^2+1}L_{\text{FFN}}M^{3L_T}d_{embd}^{18L_T^2}w_{\text{FFN}}^{18L_T^2 L_{\text{FFN}}}\kappa^{6L_T^2 L_{\text{FFN}}}m^{L_T^2}l^{L_T^2}}{\delta}\right)^{4d_{embd}^2 w_{\text{FFN}}^2 D(m+L_{\text{FFN}})L_T}.$$

Lemma 2 is proved in Appendix F.2.

With the covering number of $\mathcal{T}(L_T, L_{\text{FFN}}, w_{\text{FFN}}, l, d_{embd}, m, R, \kappa)$ in hand we may now apply (25) to obtain

$$\mathbb{E}\int_{\mathcal{M}}(\hat{\text{T}}_n(x) - f(x))^2 dQ - 2\mathbb{E}\frac{1}{n}\sum_{i=1}^{n}(\hat{\text{T}}_n(x_i) - f(x_i))^2$$

$$\leq \inf_{\delta > 0}\left[\frac{104R^2}{3n}\log\mathcal{N}\left(\frac{\delta}{4R}, \mathcal{T}, \|\cdot\|_\infty\right) + \left(4 + \frac{1}{2R}\right)\delta\right]$$

$$\leq \frac{104R^2}{3n}\log\mathcal{N}\left(\frac{1}{4Rn}, \mathcal{T}, \|\cdot\|_\infty\right) + \left(4 + \frac{1}{2R}\right)\frac{1}{n}$$

where we set $\delta = \frac{1}{n}$. This yields

$$\mathcal{N}\left(\frac{\delta}{4Rn}, \mathcal{T}, \|\cdot\|_\infty\right) \le \log\left(\left(2^{L_T^2+1}L_{\text{FFN}}M^{3L_T}d_{embd}^{18L_T^2}w_{\text{FFN}}^{18L_T^2 L_{\text{FFN}}}\kappa^{6L_T^2 L_{\text{FFN}}}m^{L_T^2}l^{L_T^2}n\right)^{4d_{embd}^2 w_{\text{FFN}}^2 D(m+L_{\text{FFN}})L_T}\right)$$

$$\le 4d_{embd}^2 w_{\text{FFN}}^2 D(m+L_{\text{FFN}})L_T\left(5L_T^2 L_{\text{FFN}}\log(2ML_{\text{FFN}}d_{embd}\kappa m l n)\right)$$

$$\le 20\log(2ML_{\text{FFN}}d_{embd}w_{\text{FFN}}\kappa m l n)Dd_{embd}^2 w_{\text{FFN}}^2 m L_T^3 L_{\text{FFN}}^2.$$

Recall for target accuracy $\epsilon > 0$ we can choose $L_T = \log(d)$, $L_{\text{FFN}} = \log(\epsilon^{-1})$, $w_{\text{FFN}} \le 4C_\mathcal{M}$, $d_{embd} = 5$, $l \le C_\mathcal{M}d\epsilon^{-\frac{d}{\beta}}$, $m \le C_\mathcal{M}d\epsilon^{-\frac{d}{\beta}}$, $\kappa \le (MC_\mathcal{M}d\epsilon^{-\frac{d}{\beta}})^2$, $l \le C_\mathcal{M}d\epsilon^{-\frac{d}{\beta}}$. This simplifies the above to

$$\mathcal{N}(\frac{\delta}{4R}, \mathcal{T}, \|\cdot\|_\infty) \le 900\left(\log(d)^2\log(60d^3\epsilon^{-4\frac{d}{\beta}}n)\right)C_\mathcal{M}^3 Dd\epsilon^{-\frac{d}{\beta}} \le \tilde{O}\left(Dd^2\epsilon^{-\frac{d}{\beta}}\right)$$

where $\tilde{O}$ hides $\log$ terms and constants. Then we can bound the variance term as

$$\mathbb{E}\int_\mathcal{M}(\hat{T}_n(x) - f(x))^2 dQ - 2\mathbb{E}\frac{1}{n}\sum_{i=1}^n(\hat{T}_n(x_i) - f(x_i))^2 \le \tilde{O}\left(\frac{Dd^2\epsilon^{-\frac{d}{\beta}}}{n}\right)$$

Putting together the bounds on the bias and variance yields a bound on the empirical risk:

$$\mathbb{E}\int_\mathcal{M}(\hat{T}_n(x) - f(x))^2 dQ \le \tilde{O}\left(\epsilon^2 + \frac{Dd^2\epsilon^{-\frac{d}{\beta}}}{n}\right)$$

and it simply remains to pick $\epsilon$ to balance the error terms. We do this by choosing $\epsilon$ such that $\epsilon^2 = \frac{\epsilon^{-\frac{d}{\beta}}}{n}$ yielding $\epsilon = n^{-\frac{\beta}{2\beta+d}}$. This gives our final bound on the expected empirical risk:

$$\mathbb{E}\int_\mathcal{M}(\hat{T}_n(x) - f(x))^2 dQ - 2\mathbb{E}\frac{1}{n}\sum_{i=1}^n(\hat{T}_n(x_i) - f(x_i))^2 \le \tilde{O}\left(Dd^2 n^{-\frac{2\beta}{2\beta+d}}\right)$$

as desired.

$\square$

### F.2 Proof of Lemma 2 about the Transformer Covering Number

*Proof of Lemma 2.* The key is to bound the difference in $\|\cdot\|_\infty$ between two transformers $T, T' \in \mathcal{T}(L_T, L_{\text{FFN}}, w_{\text{FFN}}, l, d_{embd}, m, R, \kappa)$. Set $\eta > 0$ and choose $T, T'$ so that $\|\theta - \theta'\|_\infty < \eta$. In other words, the weight parameters in $T$ and $T'$ differ at most by $\eta$. Fix $x \in [0,1]^D$. Recall the decoder $D : \mathbb{R}^{d_{embd} \times l} \to \mathbb{R}$ is fixed to output the first element of the first row. We compute

$$|T(x) - T'(x)| = |D \circ B_L \circ ... \circ B_1 \circ (PE + E(x)) - D' \circ B'_L \circ ... \circ B'_1 \circ (PE + E'(x))|$$

$$\le \|B_L \circ ... \circ B_1 \circ (PE + E(x)) - B'_L \circ ... \circ B'_1 \circ (PE + E'(x))\|_\infty.$$

To handle this term, we write

$$\|B_L \circ ... \circ B_1 \circ (PE + E(x)) - B'_L \circ ... \circ B'_1 \circ (PE + E'(x))\|_\infty$$

$$\le \|B_L \circ ... \circ B_1 \circ (PE + E(x)) - B_L \circ ... \circ B_1 \circ (PE + E'(x))\|_\infty$$

$$+ \|B_L \circ ... \circ B_1 \circ (PE + E'(x)) - B'_L \circ ... \circ B'_1 \circ (PE + E'(x))\|_\infty.$$

We will first bound the second term.

Consider two multi-headed attention layers $MHA_1, MHA_2 \in \mathcal{MHA}(m)$ with attention heads $A_j^i$, $i \in \{1, 2\}$, $1 \le j \le m$. For $H \in [0, M]^{d_{embd} \times l}$ we compute

$$\|MHA_1(H) - MHA_2(H)\|_\infty = \|\sum_{j=1}^m A_j^1(H) - \sum_{j=1}^m A_j^2(H)\|_\infty = \|\sum_{j=1}^m \left[A_j^1(H) - A_j^2(H)\right]\|_\infty$$

$$\le \sum_{j=1}^m \|A_j^1(H) - A_j^2(H)\|_\infty$$

Consider two attention heads $A_1, A_2$ with weight parameters $Q_i, K_i, V_i$, $i \in \{1, 2\}$. For $H \in [-M, M]^{d_{embd} \times l}$, we compute

$$\|A_1(H) - A_2(H)\|_\infty \leq \max_{1 \leq j \leq l} \|A_1(h_j) - A_2(h_j)\|_\infty$$

$$= \max_{1 \leq j \leq l} \|\sum_{k=1}^{l} \sigma(\langle Q_1 h_j, K_1 h_k\rangle)V_1 h_k - \sum_{k=1}^{l} \sigma(\langle Q_2 h_j, K_2 h_k\rangle)V_2 h_k\|_\infty$$

$$\leq \max_{1 \leq j \leq l} \sum_{k=1}^{l} \|\sigma(\langle Q_1 h_j, K_1 h_k\rangle)V_1 h_k - \sigma(\langle Q_2 h_j, K_2 h_k\rangle)V_2 h_k\|_\infty$$

$$\leq \max_{1 \leq j \leq l} \sum_{k=1}^{l} \|\sigma(\langle Q_1 h_j, K_1 h_k\rangle)V_1 h_k - \sigma(\langle Q_2 h_j, K_2 h_k\rangle)V_1 h_k\|_\infty$$

$$+ \|\sigma(\langle Q_2 h_j, K_2 h_k\rangle)V_1 h_k - \sigma(\langle Q_2 h_j, K_2 h_k\rangle)V_2 h_k\|_\infty$$

where the last inequality comes from adding and subtracting the term $\langle Q_2 h_j, K_2 h_k\rangle V_1 h_k$. We bound the first term via

$$\|\sigma(\langle Q_1 h_j, K_1 h_k\rangle)V_1 h_k - \sigma(\langle Q_2 h_j, K_2 h_k\rangle)V_1 h_k\|_\infty$$
$$= |\langle Q_1 h_j, K_1 h_k\rangle - \langle Q_2 h_j, K_2 h_k\rangle| \|V_1 h_k\|_\infty$$
$$\leq [|\langle Q_1 h_j, K_1 h_k\rangle - \langle Q_1 h_j, K_2 h_k\rangle| + |\langle Q_1 h_j, K_2 h_k\rangle - \langle Q_2 h_j, K_2 h_k\rangle|] \|V_1 h_k\|_\infty$$
$$\leq [|\langle Q_1 h_j, K_1 h_k - K_2 h_k\rangle| + |\langle Q_1 h_j - Q_2 h_j, K_2 h_k\rangle|] \|V_1 h_k\|_\infty$$
$$\leq [\|Q_1 h_j\|_2 \|K_1 h_k - K_2 h_k\|_2 + \|Q_1 h_j - Q_2 h_j\|_2 \|K_2 h_k\|_2] \|V_1 h_k\|_\infty$$
$$\leq [\|Q_1\|_1 \|h_j\|_\infty \|K_1 - K_2\|_1 \|h_k\|_\infty + \|Q_1 - Q_2\|_1 \|h_j\|_\infty \|K_2\|_1 \|h_k\|_\infty] \|V_1 h_k\|_\infty$$
$$\leq [2\kappa^2 d_{embd}^4 M^2 \eta] \|V_1 h_k\|_\infty$$
$$\leq 2\kappa^3 d_{embd}^6 M^3 \eta$$

where we note $\|Ax\|_2 \leq \|Ax\|_1 \leq \|A\|_1 \|x\|_\infty$ for $A \in \mathbb{R}^{d_{embd} \times d_{embd}}, x \in \mathbb{R}^{d_{embd}}$ and $\|A\|_1 \leq d_{embd}^2 \|A\|_\infty$. This bounds the first term.

The second term can be bounded as

$$\|\sigma(\langle Q_2 h_j, K_2 h_k\rangle)V_1 h_k - \sigma(\langle Q_2 h_j, K_2 h_k\rangle)V_2 h_k\|_\infty$$
$$= \|V_1 - V_2\|_1 |\langle Q_2 h_j, K_2 h_k\rangle h_k\|_\infty$$
$$\leq d_{embd}^2 \kappa \eta \|\langle Q_2 h_j, K_2 h_k\rangle h_k\|_\infty$$
$$\leq d_{embd}^2 \kappa \eta \|Q_2 h_j\|_2 \|K_2 h_k\|_2 \|h_k\|_\infty$$
$$\leq d_{embd}^6 \kappa^3 M^3 \eta.$$

Thus we can bound the attention difference $\|A_1(H) - A_2(H)\|_\infty$ as

$$\|A_1(H) - A_2(H)\|_\infty \leq \max_{1 \leq j \leq l} \sum_{k=1}^{l} \|\langle Q_1 h_j, K_1 h_k\rangle V_1 h_k - \langle Q_2 h_j, K_2 h_k\rangle V_1 h_k\|_\infty$$

$$+ \|\langle Q_2 h_j, K_2 h_k\rangle V_1 h_k - \langle Q_2 h_j, K_2 h_k\rangle V_2 h_k\|_\infty$$

$$\leq \max_{1 \leq j \leq l} \sum_{k=1}^{l} 3\kappa^3 d_{embd}^6 M^3 \eta = 3\kappa^3 d_{embd}^6 M^3 l \eta.$$

We can also now bound the multi-headed difference $\|\mathrm{MHA}_1(H) - \mathrm{MHA}_2(H)\|_\infty$ as

$$\|\mathrm{MHA}_1(H) - \mathrm{MHA}_2(H)\|_\infty \leq \sum_{j=1}^{m} \|A_j^1(H) - A_j^2(H)\|_\infty \leq 3\kappa^3 d_{embd}^6 M^3 \eta = 3\kappa^3 d_{embd}^6 M^3 m l \eta.$$

A bound on the first residual layer in the transformer block immediately follows via

$$\|(H + \mathrm{MHA}_1(H)) - (H + \mathrm{MHA}_2(H))\|_\infty = \|\mathrm{MHA}_1(H) - \mathrm{MHA}_2(H)\|_\infty.$$

To bound the difference in the output of the FFN layer in the transformer block we have for $f_1, f_2 \in$ FFN($L_{\text{FFN}}$) and input $H \in [0, M']^{d_{embd} \times l}$ we have

$$\|f_1(H) - f_2(H)\|_\infty \leq L_{\text{FFN}}(d_{embd}M' + 2)(\kappa d_{embd})^{L_{\text{FFN}}-1}\eta$$

which also bounds the second residual layer of the transformer block. Then to bound the difference of two transformer blocks $\text{B}_1, \text{B}_2 \in \mathcal{B}(m, L_{\text{FFN}})$ write, for $H \in [0, M]^{d_{embd} \times l}$,

$$
\begin{aligned}
\|\text{B}_1(H) - \text{B}_2(H)\|_\infty &= \|\big(H + \text{MHA}_1(H) + \text{FFN}_1(H + \text{MHA}_1(H))\big) \\
&\quad - \big(H + \text{MHA}_2(H) + \text{FFN}_2(H + \text{MHA}_2(H))\big)\|_\infty \\
&= \|\text{MHA}_1(H) + \text{FFN}_1(H + \text{MHA}_1(H)) - \text{MHA}_2(H) + \text{FFN}_2(H + \text{MHA}_2(H))\|_\infty \\
&\leq \|\text{MHA}_1(H) - \text{MHA}_2(H)\|_\infty \\
&\quad + \|\text{FFN}_1(H + \text{MHA}_1(H)) - \text{FFN}_2(H + \text{MHA}_2(H))\|_\infty \\
&\leq 3\kappa^3 d_{embd}^6 M^3 ml\eta + \|\text{FFN}_1(H + \text{MHA}_1(H)) - \text{FFN}_2(H + \text{MHA}_2(H))\|_\infty
\end{aligned}
$$

where we can bound the first term via analysis above. For the second term write

$$
\begin{aligned}
&\|\text{FFN}_1(H + \text{MHA}_1(H)) - \text{FFN}_2(H + \text{MHA}_2(H))\|_\infty \\
&\leq \|\text{FFN}_1(H + \text{MHA}_1(H)) - \text{FFN}_1(H + \text{MHA}_2(H))\|_\infty + \|\text{FFN}_2(H + \text{MHA}_2(H)) - \text{FFN}_1(H + \text{MHA}_2(H))\|_\infty.
\end{aligned}
$$

To handle the first term we must bound he Lipschitz constant of $\text{FFN}_1$. Let $H, H' \in \mathbb{R}^{d_{embd} \times l}$. Write

$$
\begin{aligned}
\|\text{FFN}_1(H) - \text{FFN}_1(H')\|_\infty &= \max_{1 \leq j \leq l} \|\text{FFN}_1(h_j) - \text{FFN}_1(h'_j)\|_\infty \\
&= \max_{1 \leq j \leq l} \|W_{L_{\text{FFN}}}\sigma(W_{L_{\text{FFN}}-1}...\sigma(W_1 h_j + b_1)... + b_{L_{\text{FFN}}-1}) + b_{L_{\text{FFN}}} \\
&\quad - W_{L_{\text{FFN}}}\sigma(W_{L_{\text{FFN}}-1}...\sigma(W_1 h'_j + b_1)... + b_{L_{\text{FFN}}-1}) - b_{L_{\text{FFN}}}\|_\infty \\
&\leq \max_{1 \leq j \leq l} \|W_{L_{\text{FFN}}}\|_1 \|W_{L_{\text{FFN}}-1}...\sigma(W_1 h_j + b_1)... + b_{L_{\text{FFN}}-1} \\
&\quad - W_{L_{\text{FFN}}-1}...\sigma(W_1 h'_j + b_1)... + b_{L_{\text{FFN}}-1}\|_\infty \\
&\leq \max_{1 \leq j \leq l} \left[\prod_{i=1}^{L_{\text{FFN}}} \|W_i\|_1\right]\|h_j - h'_j\|_\infty \leq w_{\text{FFN}}^{2L_{\text{FFN}}}\kappa^{L_{\text{FFN}}}\|H - H'\|_\infty
\end{aligned}
$$

where we again use that $\|Wx\|_\infty \leq \|W\|_1\|x\|_\infty$ and that $\sigma(\cdot)$ is 1-Lipschitz. Applying this to the first term from above we get

$$
\begin{aligned}
\|\text{FFN}_1(H + \text{MHA}_1(H)) - \text{FFN}_1(H + \text{MHA}_2(H))\|_\infty &\leq w_{\text{FFN}}^{2L_{\text{FFN}}}\kappa^{L_{\text{FFN}}}\|\text{MHA}_1(H) - \text{MHA}_2(H)\|_\infty \\
&\leq w_{\text{FFN}}^{2L_{\text{FFN}}}\kappa^{L_{\text{FFN}}}3\kappa^3 d_{embd}^6 M^3 ml\eta \\
&= 3\kappa^{3+L_{\text{FFN}}}w_{\text{FFN}}^{2L_{\text{FFN}}}d_{embd}^6 M^3 ml\eta.
\end{aligned}
$$

We can bound $\|\text{FFN}_2(H + \text{MHA}_2(H)) - \text{FFN}_1(H + \text{MHA}_2(H))\|_\infty$ as

$$
\begin{aligned}
&\|\text{FFN}_2(H + \text{MHA}_2(H)) - \text{FFN}_1(H + \text{MHA}_2(H))\|_\infty \\
&\leq \|\text{FFN}_2(H + \text{MHA}_2(H)) - \text{FFN}_1(H + \text{MHA}_2(H))\|_\infty \leq L_{\text{FFN}}(w_{\text{FFN}}M' + 2)(\kappa w_{\text{FFN}})^{L_{\text{FFN}}-1}\eta
\end{aligned}
$$

where $\|H + \text{MHA}_2(H)\|_\infty \leq M'$. Putting all the estimates together, we have

$$
\begin{aligned}
&\|\text{B}_1(H) - \text{B}_2(H)\|_\infty \\
&\leq 3\kappa^3 d_{embd}^6 M^3 ml\eta + \|\text{FFN}_1(H + \text{MHA}_1(H)) - \text{FFN}_2(H + \text{MHA}_2(H))\|_\infty \\
&\leq 3\kappa^3 d_{embd}^6 M^3 ml\eta + 3\kappa^{3+L_{\text{FFN}}}w_{\text{FFN}}^{2L_{\text{FFN}}}d_{embd}^6 M^3 ml\eta + L_{\text{FFN}}(w_{\text{FFN}}M' + 2)(\kappa w_{\text{FFN}})^{L_{\text{FFN}}-1}\eta \\
&\leq \big(4\kappa^{3+L_{\text{FFN}}}w_{\text{FFN}}^{2L_{\text{FFN}}}d_{embd}^6 M^3 ml + L_{\text{FFN}}(w_{\text{FFN}}M' + 2)(\kappa w_{\text{FFN}})^{L_{\text{FFN}}-1}\big)\eta
\end{aligned}
$$

and it remains to control $M'$ in terms of $M$. We know $\|H + \text{MHA}(H)\|_\infty \leq M'$ and so we compute

$$
\begin{aligned}
\|H + \text{MHA}(H)\|_\infty &\leq M + \|\text{MHA}(H)\|_\infty \\
&\leq M + \sum_{j=1}^{m} \max_{1 \leq i \leq l} \sum_{k=1}^{l} \|\langle Q_j h_i, K_j h_k\rangle V_j h_k\|_\infty \\
&\leq M + mld_{embd}^6\kappa^3 M \leq 2d_{embd}^6\kappa^3 mlM.
\end{aligned}
$$

Thus, we can complete our estimate on $\|B_1(H) - B_2(H)\|_\infty$ as

$$\|B_1(H) - B_2(H)\|_\infty \leq \left(4\kappa^{3+L_{\text{FFN}}}w_{\text{FFN}}^{2L_{\text{FFN}}}d_{embd}^6 M^3 ml + L_{\text{FFN}}(w_{\text{FFN}}(2d_{embd}^6\kappa^3 mlM) + 2)(\kappa w_{\text{FFN}})^{L_{\text{FFN}}-1}\right)\eta.$$

Now consider the multi-block difference

$$\|B_{L_T} \circ ... \circ B_1(H) - B'_{L_T} \circ ... \circ B'_1(H)\|_\infty \leq \|B_{L_T} \circ ... \circ B_1(H) - B_{L_T} \circ ... \circ B'_1(H)\|_\infty$$
$$+ \|B_{L_T} \circ ... \circ B'_1(H) - B'_{L_T} \circ ... \circ B'_1(H)\|_\infty.$$

To bound the second difference we must bound the output $\|B(H)\|_\infty$, $B \in \mathcal{B}(m, L_{\text{FFN}})$. Suppose $\|H\|_\infty \leq M$. Write

$$\|B(H)\|_\infty = \|\text{FFN}(H + \text{MHA}(H)) + \text{MHA}(H) + H\|_\infty$$
$$\leq \|H\|_\infty + \|\text{MHA}(H)\|_\infty + \|\text{FFN}(H + \text{MHA}(H))\|_\infty$$
$$\leq 2d_{embd}^6\kappa^3 mlM + \|\text{FFN}(H + \text{MHA}(H))\|_\infty$$

so we must bound the output of the FFN layer. We have for $H' \in \mathbb{R}^{d_{embd} \times l}$ with $\|H'\|_\infty \leq M$, we have

$$\|\text{FFN}(H')\|_\infty \leq (\kappa d_{embd}^2)^{L_{\text{FFN}}-1}M + \kappa(\kappa d_{embd}^2)^{L_{\text{FFN}}-2}.$$

Since $\|H + MHA(H)\|_\infty \leq 2d_{embd}^6\kappa^3 mlM$, we have

$$\|B(H)\|_\infty \leq 2d_{embd}^6\kappa^3 mlM + (\kappa w_{\text{FFN}}^2)^{L_{\text{FFN}}-1}2d_{embd}^6\kappa^3 mlM + \kappa(\kappa w_{\text{FFN}}^2)^{L_{\text{FFN}}-2}$$
$$\leq 4d_{embd}^4 w_{\text{FFN}}^{2L_{\text{FFN}}}\kappa^{L_{\text{FFN}}+1}mlM.$$

Iterating this gives

$$\|B_{L_T-1} \circ ... \circ B_1(H)\|_\infty \leq (4d_{embd}^4 w_{\text{FFN}}^{2L_{\text{FFN}}}\kappa^{L_{\text{FFN}}+1}ml)^{L_T-1}M.$$

Then we can bound the above second term as

$$\|B_{L_T} \circ ... \circ B'_1(H) - B'_{L_T} \circ ... \circ B'_1(H)\|_\infty$$
$$\leq \left(4\kappa^{3+L_{\text{FFN}}}d_{embd}^{6+2L_{\text{FFN}}}((4d_{embd}^{2L_{\text{FFN}}+4}\kappa^{L_{\text{FFN}}+1}ml)^{L_T-1}M)^3 ml\right.$$
$$\left. + L_{\text{FFN}}(d_{embd}(2d_{embd}^6\kappa^3 ml((4d_{embd}^{2L_{\text{FFN}}+4}\kappa^{L_{\text{FFN}}+1}ml)^{L_T-1}M)) + 2)(\kappa d_{embd})^{L_{\text{FFN}}-1}\right)\eta$$
$$\leq 4^{3L_T-2}L_{\text{FFN}}d_{embd}^{6L_T L_{\text{FFN}}+12L_T-10L_{\text{FFN}}-6}\kappa^{3L_T L_{\text{FFN}}+3L_T-2L_{\text{FFN}}}M^3 m^{L_T}l^{L_T}\eta$$
$$\leq 4^{3L_T}L_{\text{FFN}}M^3 d_{embd}^{18L_T L_{\text{FFN}}}w_{\text{FFN}}^{18L_T L_{\text{FFN}}}\kappa^{6L_T L_{\text{FFN}}}m^{L_T}l^{L_T}\eta.$$

To bound the first term $\|B_{L_T} \circ ... \circ B_1(H) - B_{L_T} \circ ... \circ B'_1(H)\|_\infty$ we must control the Lipschitz constant of of $B \in \mathcal{B}(m, L_{\text{FFN}})$. Let $H, H' \in \mathbb{R}^{d_{embd} \times l}$ with $\|H\|_\infty, \|H'\|_\infty \leq M$. Compute

$$\|B(H) - B(H')\|_\infty = \|\text{FFN}(H + \text{MHA}(H)) - \text{FFN}(H' + \text{MHA}(H')) + \text{MHA}(H) - \text{MHA}(H')\|_\infty$$
$$\leq \|\text{FFN}(H + \text{MHA}(H)) - \text{FFN}(H' + \text{MHA}(H'))\|_\infty + \|\text{MHA}(H) - \text{MHA}(H')\|_\infty$$
$$\leq \|\text{FFN}(H + \text{MHA}(H)) - \text{FFN}(H' + \text{MHA}(H'))\|_\infty + \|\text{MHA}(H) - \text{MHA}(H')\|_\infty$$
$$\leq w_{\text{FFN}}^{2L_{\text{FFN}}}\kappa^{L_{\text{FFN}}}\|\text{MHA}(H) - \text{MHA}(H') + H - H'\|_\infty + \|\text{MHA}(H) - \text{MHA}(H')\|_\infty$$
$$\leq w_{\text{FFN}}^{2L_{\text{FFN}}}\kappa^{L_{\text{FFN}}}\|H - H'\|_\infty + (w_{\text{FFN}}^{2L_{\text{FFN}}}\kappa^{L_{\text{FFN}}} + 1)\|\text{MHA}(H) - \text{MHA}(H')\|_\infty$$

and it suffices to control the Lipschitz constant of $\text{MHA} \in \mathcal{MHA}(m)$. For the same $H, H'$ compute

$$\|\text{MHA}(H) - \text{MHA}(H')\|_\infty \leq \max_{1 \leq i \leq l}\sum_{j=1}^m \sum_{k=1}^l \|\langle Q_j h_i, K_j h_k\rangle V_j h_k - \langle Q_j h'_i, K_j h'_k\rangle V_j h'_k\|_\infty.$$

We can bound $\left|\langle Q_j h_i, K_j h_k\rangle V_j h_k - \langle Q_j h'_i, K_j h'_k\rangle V_j h'_k\right|$ as

$$\|\langle Q_j h_i, K_j h_k\rangle V_j h_k - \langle Q_j h'_i, K_j h'_k\rangle V_j h'_k\|_\infty$$
$$\leq \|\langle Q_j h_i, K_j h_k\rangle V_j h_k - \langle Q_j h'_i, K_j h'_k\rangle V_j h'_k\|_\infty + \|\langle Q_j h'_i, K_j h'_k\rangle V_j h'_k - \langle Q_j h'_i, K_j h'_k\rangle V_j h_k\|_\infty$$
$$\leq \left|\langle Q_j h_i, K_j h_k\rangle - \langle Q_j h'_i, K_j h'_k\rangle\right|\|V_j h_k\|_\infty + \left|\langle Q_j h'_i, K_j h'_k\rangle\right|\|V_j h'_k - V_j h_k\|_\infty$$
$$\leq \left|\langle Q_j h_i, K_j h_k\rangle - \langle Q_j h'_i, K_j h'_k\rangle\right|d_{embd}^2\kappa M + \left|\langle Q_j h'_i, K_j h'_k\rangle\right|d_{embd}^2\kappa\|h_k - h'_k\|_\infty$$
$$\leq \left|\langle Q_j h_i, K_j h_k\rangle - \langle Q_j h'_i, K_j h'_k\rangle\right|d_{embd}^2\kappa M + 2d_{embd}^4\kappa^2 M^2 d_{embd}^2\kappa\|h_k - h'_k\|_\infty$$
$$\leq 4d_{embd}^6\kappa^3 M^2\|h_k - h'_k\|_\infty.$$

Applying this gives

$$\|\text{MHA}(H) - \text{MHA}(H')\|_\infty \leq 4d_{embd}^6\kappa^3 M^2 ml\|H - H'\|_\infty.$$

Then

$$\|\text{B}(H) - \text{B}(H')\|_\infty \leq w_{\text{FFN}}^{2L_{\text{FFN}}}\kappa^{L_{\text{FFN}}}\|H - H'\|_\infty + (w_{\text{FFN}}^{2L_{\text{FFN}}}\kappa^{L_{\text{FFN}}} + 1)\|\text{MHA}(H) - \text{MHA}(H')\|_\infty$$

$$\leq w_{\text{FFN}}^{2L_{\text{FFN}}}\kappa^{L_{\text{FFN}}}\|H - H'\|_\infty + (w_{\text{FFN}}^{2L_{\text{FFN}}}\kappa^{L_{\text{FFN}}} + 1)4d_{embd}^6\kappa^3 M^2 ml\|H - H'\|_\infty$$

$$\leq 8w_{\text{FFN}}^{2L_{\text{FFN}}}\kappa^{L_{\text{FFN}}}d_{embd}^6\kappa^3 M^2 ml\|H - H'\|_\infty.$$

With this we can bound the first term of the multi-block difference

$$\|\text{B}_{L_T} \circ ... \circ \text{B}_1(H) - \text{B}_{L_T} \circ ... \circ \text{B}_1'(H)\|_\infty \leq 8d_{embd}^{2L_{\text{FFN}}+6}\kappa^{L_{\text{FFN}}}d_{embd}^6\kappa^3\|B_{L_T-1} \circ ... \circ B_1(H)\|_\infty^2 ml$$

$$* \|B_{L_T-1} \circ ... \circ B_1(H) - B_{L_T-1}' \circ ... \circ B_1'(H)\|_\infty$$

$$\leq 8d_{embd}^{2L_{\text{FFN}}+6}\kappa^{L_{\text{FFN}}}d_{embd}^6\kappa^3((4d_{embd}^{2L_{\text{FFN}}+4}\kappa^{L_{\text{FFN}}+1}ml)^{L_T-1}M)^2 ml$$

$$* \|B_{L_T-1} \circ ... \circ B_1(H) - B_{L_T-1}' \circ ... \circ B_1'(H)\|_\infty$$

$$\leq 2^7 d_{embd}^{5L_T L_{\text{FFN}}}\kappa^{2L_T L_{\text{FFN}}}d_{embd}^6\kappa^3 M^2 m^{L_T}l^{L_T}$$

$$* \|B_{L_T-1} \circ ... \circ B_1(H) - B_{L_T-1}' \circ ... \circ B_1'(H)\|_\infty$$

. Putting together the estimates on both terms gives

$$\|\text{B}_{L_T} \circ ... \circ \text{B}_1(H) - \text{B}_{L_T}' \circ ... \circ \text{B}_1'(H)\|_\infty \leq 2^7 d_{embd}^{5L_T L_{\text{FFN}}}\kappa^{2L_T L_{\text{FFN}}}d_{embd}^6\kappa^3 M^2 m^{L_T}l^{L_T}$$

$$* \|B_{L_T-1} \circ ... \circ B_1(H) - B_{L_T-1}' \circ ... \circ B_1'(H)\|_\infty$$

$$+ 4^{3L_T}L_{\text{FFN}}M^3 d_{embd}^{18L_T L_{\text{FFN}}}\kappa^{6L_T L_{\text{FFN}}}m^{L_T}l^{L_T}\eta$$

$$\leq 2^{7L_T}L_{\text{FFN}}M^3 d_{embd}^{18L_T L_{\text{FFN}}}\kappa^{6L_T L_{\text{FFN}}}m^{L_T}l^{L_T}$$

$$* \|B_{L_T-1} \circ ... \circ B_1(H) - B_{L_T-1}' \circ ... \circ B_1'(H)\|_\infty$$

which reduces the bound on the $L_T$ layer transformer class to the bound on the bound on the $L_T - 1$ layer transformer class. Finally, via induction, we get

$$\|\text{B}_{L_T} \circ ... \circ \text{B}_1(H) - \text{B}_{L_T}' \circ ... \circ \text{B}_1'(H)\|_\infty \leq \left(2^{7L_T}L_{\text{FFN}}M^3 d_{embd}^{18L_T}w_{\text{FFN}}^{18L_T L_{\text{FFN}}}\kappa^{6L_T L_{\text{FFN}}}m^{L_T}l^{L_T}\right)^{L_T}\eta$$

$$\leq 2^{7L_T^2}L_{\text{FFN}}M^{3L_T}d_{embd}^{18L_T^2}w_{\text{FFN}}^{18L_T^2 L_{\text{FFN}}}\kappa^{6L_T^2 L_{\text{FFN}}}m^{L_T^2}l^{L_T^2}\eta.$$

It simply remains to deal with the encoding layer $H = \text{PE} + \text{E}(x)$. Both $E$ and $PE$ are fixed among architectures and further $\|\text{PE} + \text{E}(x)\|_\infty = \|x\|_\infty + 1 \leq M + 1$. Thus

$$\|\text{B}_{L_T}' \circ ... \circ \text{B}_1' \circ (\text{PE} + \text{E}'(x))\|_\infty \leq (4d_{embd}^{2L_{\text{FFN}}+4}\kappa^{L_{\text{FFN}}+1}ml)^{L_T}M.$$

This allows us to complete the total bound on the transformer difference of $\text{T}, \text{T}' \in \mathcal{T}(L_T, L_{\text{FFN}}, w_{\text{FFN}}, l, d_{embd}, m, R, \kappa)$ such that $\|\theta - \theta'\|_\infty < \eta$ as

$$\|\text{T}(x) - \text{T}'(x)\|_\infty = \|\text{D} \circ \text{B}_{L_T} \circ ... \circ \text{B}_1 \circ (\text{PE} + \text{E}(x)) - \text{D}' \circ \text{B}_{L_T}' \circ ... \circ \text{B}_1' \circ (\text{PE} + \text{E}'(x))\|_\infty$$

$$\leq 2^{L_T^2+1}L_{\text{FFN}}M^{3L_T}d_{embd}^{18L_T^2 L_{\text{FFN}}}\kappa^{6L_T^2 L_{\text{FFN}}}m^{L_T^2}l^{L_T^2}\eta = C_T\eta.$$

Now, we may compute $\mathcal{N}(\mathcal{T}, \delta, \|\cdot\|_\infty)$. We must cover $\mathcal{T}(L_T, L_{\text{FFN}}, w_{\text{FFN}}, l, d_{embd}, m, R, \kappa)$ with a $\delta$-net such that, for all $\text{T} \in \mathcal{T}(L_T, L_{\text{FFN}}, w_{\text{FFN}}, l, d_{embd}, m, R, \kappa)$, we can find $T'$ in the net such that $\|\text{T} - \text{T}'\|_\infty < \delta$. Set $\eta = \frac{\delta}{C_T}$. We construct the $\delta$-net by uniformly discretizing the set of weights $\theta$ corresponding to $\text{T}_\theta \in \mathcal{T}$ with step size given by $\eta$. Then for any $\text{T} \in \mathcal{T}$, we can find a $T'$ in the grid such that $\|\theta_{\text{T}} - \theta_{\text{T}'}\|_\infty < \eta$. Then we have $\|\text{T} - \text{T}'\|_\infty < C_T\eta = C_T\frac{\delta}{C_T} = \delta$. We can compute the number of parameters in $\theta$ as

$$|\theta| = |\theta_E| + |\theta_D| + \sum_{i=1}^{L_T}|\theta_{\text{B}_i}| = d_{embd} + d_{embd}D + L_T|\theta_{\text{B}}|$$

$$= d_{embd} + d_{embd}D + L_T(|\theta_{\text{MHA}}| + |\theta_{\text{FFN}}|)$$

$$= d_{embd} + d_{embd}D + L_T(3d_{embd}^2 m + L_{\text{FFN}}w_{\text{FFN}}^2)$$

$$\leq 4d_{embd}^2 w_{\text{FFN}}^2 D(m + L_{\text{FFN}})L_T.$$

We can compute the number of steps per parameter as $\frac{2\kappa}{\eta}$. Thus the covering number of $\mathcal{T}(L_T, L_{\text{FFN}}, w_{\text{FFN}}, l, d_{embd}, m, R, \kappa)$ is bounded by

$$\mathcal{N}(\mathcal{T}, \delta, \|\cdot\|_\infty) \leq \left(\frac{\kappa}{\eta}\right)^{4d_{embd}^2 w_{\text{FFN}}^2 D(m+L_{\text{FFN}})L_T}$$

$$= \left(\frac{C_T \kappa}{\delta}\right)^{4d_{embd}^2 w_{\text{FFN}}^2 D(m+L_{\text{FFN}})L_T}$$

$$= \left(\frac{2^{L_T^2+1} L_{\text{FFN}} M^{3L_T} d_{embd}^{18L_T^2} w_{\text{FFN}}^{18L_T^2 L_{\text{FFN}}} \kappa^{6L_T^2 L_{\text{FFN}}} m^{L_T^2} l^{L_T^2}}{\delta}\right)^{4d_{embd}^2 w_{\text{FFN}}^2 D(m+L_{\text{FFN}})L_T}.$$

$\square$

## G  Building Blocks of Transformer Neural Networks

**Lemma 3** (Interaction Lemma). *Set $d_{embd} = 5$ and let $\kappa, M > 0$. Fix $l \in \mathbb{N}$, $1 \leq t_1, t_2 \leq l$, and $1 \leq i \leq d_{embd}$. Let $H \in \mathbb{R}^{d_{embd} \times l}$ be a structured transformer embedding matrix such that $h_t^{d_{embd}-2:d_{embd}-1} = \mathcal{I}_t$ and $h_t^{d_{embd}} = 1$. Further suppose $\|H\|_\infty \leq M$ for some $M > 0$. Let $B : \mathbb{R}^{d_{embd}} \times \mathbb{R}^{d_{embd}} \to \mathbb{R}$ be a kernel function on tokens which can be written in the form $B(h, h') = \sigma(\langle Q^B h, K^B h'\rangle)$ for some $h, h' \in \mathbb{R}^{d_{embd}}, Q^B, K^B \in \mathbb{R}^{(d_{embd}-3) \times d_{embd}}$ with $\|Q^B\|_{\infty,\infty}, \|K^B\|_{\infty,\infty} \leq \kappa$. Then we can construct an attention head $A$ acting on $H$ such that $A(h_{t_1}) = B(h_{t_1}, h_{t_2})e_i$ and otherwise $A(h_t) = 0$ for $t \neq t_1$. Further we have $\|\theta_A\|_\infty = O(d_{embd}^4 \kappa^2 M^2 l^2)$. Note: we call the matrices $Q^B, K^B$ "data kernels".*

*Proof of Lemma 3.* Define the query, key, and value matrices as

$$Q = \begin{bmatrix} & Q^B & \\ 0 & 0 & & & 0 \\ 0 & 0 & Q^{\mathcal{I}} & & 0 \\ 0 & 0 & 0 & 0 & 1 \end{bmatrix}, \quad K = \begin{bmatrix} & K^B & \\ 0 & 0 & & & 0 \\ 0 & 0 & K^{\mathcal{I}} & & 0 \\ 0 & 0 & 0 & 0 & -C \end{bmatrix}, \quad V = e_i e_{d_{embd}}^T$$

where we call $Q^{\mathcal{I}}, K^{\mathcal{I}} \in \mathbb{R}^{2\times 2}$ the *interaction kernels*, $Q^B, K^B \in \mathbb{R}^{d_{embd}-3 \times d_{embd}}$ the *data kernels*, and $C > 0$ simply a large positive number. We can choose $Q^{\mathcal{I}}, K^{\mathcal{I}}$ so $K^{\mathcal{I}} = P_{\mathcal{I}_{t_2}}$ is a projection onto the unit interaction vector $\mathcal{I}_{t_2}$ (the interaction term for $h_{t_2}$) and $Q^{\mathcal{I}}$ is a dilation and rotation of $\mathcal{I}_{t_1}$ onto $\mathcal{I}_{t_2}$ i.e. $Q^{\mathcal{I}} \mathcal{I}_{t_1} = C\mathcal{I}_{t_2}$. We must now compute $A(H)$. For an arbitrary $1 \leq t \leq l$ we can compute the action of $A$ on $h_t$ as

$$A(h_t) = \sum_{k=1}^l \sigma(\langle Qh_t, Kh_k\rangle)Vh_k$$

$$= \sum_{k=1}^l \sigma(\langle Q^B h_t, K^B h_k\rangle + \langle Q^{\mathcal{I}}\mathcal{I}_t, K^{\mathcal{I}}\mathcal{I}_k\rangle - C)e_i$$

Now we must case on whether the input token $t = t_1$.

**Case 1:** $t = t_1$  Write

$$A(h_{t_1}) = \sum_{k=1}^l \sigma(\langle Q^B h_{t_1}, K^B h_k\rangle + \langle Q^{\mathcal{I}}\mathcal{I}_{t_1}, K^{\mathcal{I}}\mathcal{I}_k\rangle - C)e_i$$

To handle the same we must again go by casework, this by casing on whether $k = t_2$.

**Case 1a):** $k = t_2$  When $k = t_2$ we have $\langle Q^{\mathcal{I}}\mathcal{I}_{t_1}, K^{\mathcal{I}}\mathcal{I}_{t_2}\rangle = \langle C\mathcal{I}_{t_2}, \mathcal{I}_{t_2}\rangle$ since by construction $Q^{\mathcal{I}}\mathcal{I}_{t_1} = \mathcal{I}_{t_2}$ and $K^{\mathcal{I}}$ is a projection onto $\mathcal{I}_{t_2}$ so that $K^{\mathcal{I}}\mathcal{I}_{t_2} = \mathcal{I}_{t_2}$. This further simplifies as $\langle C\mathcal{I}_{t_2}, \mathcal{I}_{t_2}\rangle = C$ since $\mathcal{I}_{t_2}$ is unit length. Thus

$$\sigma(\langle Q^B h_{t_1}, K^B h_k\rangle + \langle Q^{\mathcal{I}}\mathcal{I}_{t_1}, K^{\mathcal{I}}\mathcal{I}_k\rangle - C) = \sigma(\langle Q^B h_{t_1}, K^B h_k\rangle + C - C)$$

$$= \sigma(\langle Q^B h_{t_1}, K^B h_k\rangle)$$

Now we must consider the other terms in the sum when $k \neq t_2$.

**Case 1b):** $k \neq t_2$ When $k \neq t_2$ we compute

$$\langle Q^{\mathcal{I}}\mathcal{I}_{t_1}, K^{\mathcal{I}}\mathcal{I}_k \rangle \leq \|Q^{\mathcal{I}}\mathcal{I}_{t_1}\|_2 \|K^{\mathcal{I}}\mathcal{I}_k\|_2$$
$$= C\|P_{\mathcal{I}_{t_2}}\mathcal{I}_k\|_2 < C$$

where we use Cauchy-Schwarz at the first inequality and note that $\|P_{\mathcal{I}_{t_2}}\mathcal{I}_k\|_2 < 1$ since $k \neq t_2$ for the second inequality. Then, for large enough $C$, we have

$$\sigma(\langle Q^B h_{t_1}, K^B h_k \rangle + \langle Q^{\mathcal{I}}\mathcal{I}_{t_1}, K^{\mathcal{I}}\mathcal{I}_k \rangle - C) \leq \sigma(\langle Q^B h_{t_1}, K^B h_k \rangle + C\|P_{\mathcal{I}_{t_2}}\mathcal{I}_k\|_2 - C) = 0$$

so we simply must choose $C$ so that $\langle Q^B h_{t_1}, K^B h_k \rangle + C\|P_{\mathcal{I}_{t_2}}\mathcal{I}_k\|_2 - C < 0$. Compute

$$\langle Q^B h_{t_1}, K^B h_k \rangle + C\|P_{\mathcal{I}_{t_2}}\mathcal{I}_k\|_2 - C < 0 \iff C > \frac{\langle Q^B h_{t_1}, K^B h_k \rangle}{1 - \|P_{\mathcal{I}_{t_2}}\mathcal{I}_k\|_2}$$

We can bound

$$\left|\langle Q^B h_{t_1}, K^B h_k \rangle\right| \leq \|Q^B h_{t_1}\|_2 \|K^B h_k\|_2$$
$$\leq \|Q^B\|_{1,1}\|h_{t_1}\|_\infty \|K^B\|_{1,1}\|h_{t_1}\|_\infty$$
$$\leq \|Q^B\|_{\infty,\infty}d_{embd}^2\|K^B\|_{\infty,\infty}d_{embd}^2 M^2 \leq d_{embd}^4 \kappa^2 M^2$$

so it remains to control $\frac{1}{1 - \|P_{\mathcal{I}_{t_2}}\mathcal{I}_k\|_2}$ when $k \neq t_2$ by upper bounding $\|P_{\mathcal{I}_{t_2}}\mathcal{I}_k\|_2$. Compute

$$\|P_{\mathcal{I}_{t_2}}\mathcal{I}_k\|_2 \leq \max_{1 \leq t_1 \neq t_2 \leq l} \|P_{\mathcal{I}_{t_1}}\mathcal{I}_{t_2}\|_2$$
$$\leq \|P_{\mathcal{I}_0}\mathcal{I}_1\|_2$$
$$= \langle \mathcal{I}_0, \mathcal{I}_1 \rangle$$
$$= \cos(\frac{\pi}{2l})$$
$$\leq 1 - \frac{1}{2}\frac{\pi^2}{2! \cdot 2^2 l^2}$$

where we note the projection is maximized by any adjacent interaction terms $\mathcal{I}_t, \mathcal{I}_{t+1}$. Then

$$\frac{\langle Q^B h_{t_1}, K^B h_k \rangle}{1 - \|P_{\mathcal{I}_{t_2}}\mathcal{I}_k\|_2} \leq \frac{d_{embd}^4 \kappa^2 M^2}{\frac{\pi^2}{2^2! \cdot 2^2 l^2}} = O(d_{embd}^4 \kappa^2 M^2 l^2)$$

which gives a bound on $C$. So for large enough $C$ we can conclude

$$\mathrm{A}(h_{t_1}) = \sigma(\langle Q^B h_{t_1}, K^B h_k \rangle)$$

It remains to consider the case $t \neq t_1$.

**Case 2:** $t \neq t_1$  Now we consider A applied to tokens other than $h_{t_1}$. We have

$$\mathrm{A}(h_t) = \sum_{k=1}^{l} \sigma(\langle Q^B h_t, K^B h_k \rangle + \langle Q^{\mathcal{I}}\mathcal{I}_t, K^{\mathcal{I}}\mathcal{I}_k \rangle - C)e_i$$

so we must argue each term in the sum is 0 by showing $\langle Q^B h_t, K^B h_k \rangle + \langle Q^{\mathcal{I}}\mathcal{I}_t, K^{\mathcal{I}}\mathcal{I}_k \rangle - C < 0$. We again split into two cases.

**Case 2a):** $k \neq t_2$  When $k \neq t_2$ we have $\langle Q^B h_t, K^B h_k \rangle + \langle Q^{\mathcal{I}}\mathcal{I}_t, K^{\mathcal{I}}\mathcal{I}_k \rangle - C < 0$ via the same argument as in case **1b**.

**Case 2b):** $k = t_2$  When $k = t_2$ we must instead more carefully bound the inner product. We can simplify the interaction terms as

$$\langle Q^{\mathcal{I}}\mathcal{I}_t, K^{\mathcal{I}}\mathcal{I}_{t_2} \rangle = \|Q^{\mathcal{I}}\mathcal{I}_t\|_2 \|K^{\mathcal{I}}\mathcal{I}_{t_2}\|_2 \cos(\theta_{t,t_2}) = C\cos(\theta_{t,t_2})$$

where $\theta_{t,t_2}$ is the angle between $Q^{\mathcal{I}}\mathcal{I}_t$ and $K^{\mathcal{I}}\mathcal{I}_k$. Crucially, because $t \neq t_1$, $Q^{\mathcal{I}}\mathcal{I}_t \neq C\mathcal{I}_{t_2}$ and so $\theta_{t,t_2} > 0$ and $\cos(\theta_{t,t_2}) < 1$. Then we can say

$$\langle Q^B h_t, K^B h_{t_2}\rangle + \langle Q^{\mathcal{I}}\mathcal{I}_t, K^{\mathcal{I}}\mathcal{I}_{t_2}\rangle - C < 0 \iff \langle Q^B h_t, K^B h_{t_2}\rangle + C\cos(\theta_{t,t_2}) - C < 0$$
$$\iff C > \frac{\langle Q^B h_t, K^B h_{t_2}\rangle}{1 - \cos(\theta_{t,t_2})}$$

We upper bound the numerator by $O(d_{embd}^4 \kappa^2 M^2 l^2)$. Note that the $Q^{\mathcal{I}}$ rotates by clockwise $\frac{t_1 - t_2}{l}\frac{\pi}{2}$ radians. So $Q^{\mathcal{I}}\mathcal{I}_t = C(\cos(\frac{(t+t_1-t_2)\pi}{2l}), \sin(\frac{(t+t_1-t_2)\pi}{2l}))$. Thus we can upper bound $\cos(\theta_{t,t_2})$ as

$$\cos(\theta_{t,t_2}) = \langle \mathcal{I}_{t+t_1-t_2}, \mathcal{I}_{t_2}\rangle$$
$$\leq \langle \mathcal{I}_0, \mathcal{I}_1\rangle$$
$$= \cos(\frac{\pi}{2l})$$
$$\leq 1 - \frac{1}{2}\frac{\pi^2}{2! \cdot 2^2 l^2}$$

So as in case **1b** we have $C = O(d_{embd}^4 \kappa^2 M^2 l^2)$. For sufficiently large $C$ we have $\mathrm{A}(h_t) = 0$ as desired. This completes the proof of Lemma 3.

$\square$

**Lemma 4** (Gating FFNs). *Given an embedding matrix $H \in \mathbb{R}^{d_{embd}\times l}$ which has all ones in the last row and whose tokens contain interaction terms and given a half-space of contiguous tokens $h_k, ..., h_l$ or $h_1, ..., h_k$, we can design* $\mathrm{FFN} \in \mathcal{FFN}(2)$ *such that*

$$\mathrm{FFN}(h_t) = \begin{cases} h_t & t \in \{1, ..., k\} \\ \begin{bmatrix} \mathbf{0} \\ \mathcal{I}_t^1 \\ \mathcal{I}_t^2 \\ 1 \end{bmatrix} & \text{otherwise} \end{cases}$$

*where* $\begin{bmatrix} \mathbf{0} \\ \mathcal{I}_t^1 \\ \mathcal{I}_t^2 \\ 1 \end{bmatrix}$ *is zero except for the last three coordinates. We call this* $\mathrm{FFN}$ *a **gating feed-forward network**. We additionally have* $\|\theta_{\mathrm{FFN}}\|_\infty \leq O(l\|H\|_\infty)$.

*Proof of Lemma 4.* Without loss of generality assume we are given a contiguous prefix of tokens $h_1, ..., h_k$, $k \leq l$. We can choose a vector $v \in \mathbb{S}^1$ such that $v \cdot \mathcal{I}_t > 0$ for $t \in \{1..., k\}$ and $v \cdot \mathcal{I}_t < 0$ for $t > k$. We call such $v$ the *pivot vector*. In particular we choose $\mathbf{R}_{\frac{\pi}{2}}(\frac{\mathcal{I}_k + \mathcal{I}_{k+1}}{\|\mathcal{I}_k + \mathcal{I}_{k+1}\|_2})$ where $\mathbf{R}_{\frac{\pi}{2}}$ is a quarter-circle rotation clockwise. For a large $C > 0$, construct

$$W_1 = \mathcal{I}_{d_{embd}} + \begin{bmatrix} 0 & 0 & Cv^1 & Cv^2 & 0 \\ 0 & 0 & Cv^1 & Cv^2 & 0 \\ 0 & 0 & 0 & 0 & 0 \\ 0 & 0 & 0 & 0 & 0 \\ 0 & 0 & 0 & 0 & 0 \end{bmatrix}, \quad b_1 = 0$$

$$W_2 = \mathcal{I}_{d_{embd}} + \begin{bmatrix} 0 & 0 & -Cv^1 & -Cv^2 & 0 \\ 0 & 0 & -Cv^1 & -Cv^2 & 0 \\ 0 & 0 & 0 & 0 & 0 \\ 0 & 0 & 0 & 0 & 0 \\ 0 & 0 & 0 & 0 & 0 \end{bmatrix}, \quad b_2 = 0.$$

where $\mathcal{I}_{d_{embd}}$ is the identity matrix of size $d_{embd} \times d_{embd}$. Then

$$z = \sigma(W_1 h_t + b_1) = \begin{bmatrix} \sigma(h_t^1 + CI_t \cdot v) \\ \sigma(h_t^2 + CI_t \cdot v) \\ \sigma(h_t^{d_{embd}-2}) \\ \sigma(h_t^{d_{embd}-1}) \\ \sigma(h_t^{d_{embd}}) \end{bmatrix} = \begin{bmatrix} \sigma(h_t^1 + CI_t \cdot v) \\ \sigma(h_t^2 + CI_t \cdot v) \\ \mathcal{I}_t^1 \\ \mathcal{I}_t^2 \\ 1 \end{bmatrix}$$

which is $\begin{bmatrix} \mathbf{0} \\ \mathcal{I}_t^1 \\ \mathcal{I}_t^2 \\ 1 \end{bmatrix}$ if $\mathcal{I}_t \cdot v < 0$. Applying the second layer to $\begin{bmatrix} \mathbf{0} \\ \mathcal{I}_t^1 \\ \mathcal{I}_t^2 \\ 1 \end{bmatrix}$ yields $\begin{bmatrix} \mathbf{0} \\ \mathcal{I}_t^1 \\ \mathcal{I}_t^2 \\ 1 \end{bmatrix}$. Otherwise assume

$\mathcal{I}_t \cdot v > 0$. Then $\sigma(h_t^i + CI_t \cdot v) = h_t^i + CI_t \cdot v$. Applying the second layer yields

$$W_2 z + b_2 = h_t$$

as desired. Further we can compute a bound on $C > 0$ as $|CI_t \cdot v| > \|H\|_\infty \iff C > \frac{\|H\|_\infty}{|I_t \cdot v|}$. We compute

$$|\mathcal{I}_t \cdot v| = |\langle \mathcal{I}_t, \mathbf{R}_{\frac{\pi}{2}} \frac{\mathcal{I}_k + \mathcal{I}_{k+1}}{\|\mathcal{I}_k + \mathcal{I}_{k+1}\|_2} \rangle| \geq \frac{1}{2} |\langle \mathcal{I}_k, \mathbf{R}_{\frac{\pi}{2}} \mathcal{I}_k \rangle + \langle \mathcal{I}_k, \mathbf{R}_{\frac{\pi}{2}} \mathcal{I}_{k+1} \rangle|$$

$$= \frac{1}{2} |\langle \mathcal{I}_k, \mathbf{R}_{\frac{\pi}{2}} \mathcal{I}_{k+1} \rangle| = |\frac{1}{2} \langle \mathcal{I}_0, \mathbf{R}_{\frac{\pi}{2}} \mathcal{I}_1 \rangle|$$

$$= \frac{1}{2} \sin\left(\frac{\pi}{2l}\right) \geq \frac{1}{4} \frac{\pi}{2l}$$

where the last inequality holds for $l \geq 2$. Thus $\frac{\|H\|_\infty}{|I_t \cdot v|} \leq \frac{2^3 l \|H\|_\infty}{\pi} < C$. $\qquad\square$

**Lemma 5** (Constant Addition). *Let $M > 0$ and $c$ be a vector in $\mathbb{R}^D$ such that $\|c\|_\infty \leq \frac{M}{2}$. Let $H$ be an embedding matrix of the form*

$$H = \begin{bmatrix} x^1 & \dots & x^D & \mathbf{0}_D \\ 0 & \dots & \dots & 0 \\ \mathcal{I}_1 & \dots & \dots & \mathcal{I}_l \\ 1 & \dots & \dots & 1 \end{bmatrix} \in \mathbb{R}^{d_{embd} \times l}$$

*where $l = 2D$ such that $\|H\|_{\infty,\infty} \leq \frac{M}{2}$. Then there exists a transformer block $\mathrm{B} \in \mathcal{B}(D, 3)$ such*

$$\mathrm{B}(H) = \begin{bmatrix} x^1 & \dots & x^D & x^1 - c^1 & \dots & x^D - c^D \\ 0 & \dots & \dots & \dots & \dots & 0 \\ \mathcal{I}_1 & \dots & \dots & \dots & \dots & \mathcal{I}_l \\ 1 & \dots & \dots & \dots & \dots & 1 \end{bmatrix}$$

*with $\|\theta_{\mathrm{B}}\|_\infty = O(lM)$. In this case we say $\mathrm{B}$ implements the addition of $c$ to $x$.*

*Proof of Lemma 5.* We begin by defining the attention heads $\mathrm{A}_i$, $1 \leq i \leq D$. For each head $\mathrm{A}_i$ we define the data kernels

$$Q_i^{data} = \begin{bmatrix} 0 & 0 & 0 & 0 & 1 \\ 0 & 0 & 0 & 0 & 1 \end{bmatrix} \quad K_i^{data} = \begin{bmatrix} 1 & 0 & 0 & 0 & 0 \\ 0 & 0 & 0 & 0 & -c^i + M \end{bmatrix}.$$

Then, via the Interaction Lemma 3, we can construct $\mathrm{A}_i$ so that token $h_{D+i}$ interacts with $h_i$ such that

$$\mathrm{A}_i(h_{D+i}) = \sigma(\langle Q_i^{data} h_{D+i}, K_i^{data} h_i \rangle) e_1$$
$$= \sigma(x^i - c^i + M) e_1 = (x^i - c^i + M) e_1$$

and $A_i(h_t) = 0$ when $t \neq D + i$. Then the residual multi-headed attention yields

$$\text{MHA}(H) + H = \begin{bmatrix} x^1 & ... & x^D & x^1 - c^1 + M & ... & x^D - c^D + M \\ 0 & ... & ... & ... & ... & ... \\ \mathcal{I}_1 & ... & ... & ... & ... & \mathcal{I}_L \\ 1 & ... & ... & ... & ... & 1 \end{bmatrix}$$

Via the gating lemma 4 we can design a three layer FFN to subtract $M$ from only $h_{D+1}, ..., h_{2D}$. Thus

$$B(H) = \begin{bmatrix} x^1 & ... & x^D & x^1 - c^1 & ... & x^D - c^D \\ 0 & ... & ... & ... & ... & ... \\ \mathcal{I}_1 & ... & ... & ... & ... & \mathcal{I}_L \\ 1 & ... & ... & ... & ... & 1 \end{bmatrix}$$

as desired.

$\square$

**Lemma 6** (Replacing FFN). *Set $d_{embd} = 5$ and $l \in \mathbb{N}$. Let $H \in \mathbb{R}^{d_{embd} \times l}$ be a structured intermediate embedding matrix. We can construct a feed-forward network $\text{FFN}_{replace} \in \mathcal{FFN}(1)$ with the following output:*

$$\text{FFN}_{replace}(H) = \begin{bmatrix} -h_1^1 + h_1^2 & ... & -h_l^1 + h_l^2 \\ 0 & ... & 0 \\ 0 & ... & 0 \\ 0 & ... & 0 \\ 0 & ... & 0 \end{bmatrix}$$

*In this case we say $\text{FFN}_{replace}$ **replaces** the the first row of $H$ with the second row.*

*Proof of Lemma 6.* Set

$$W = \begin{bmatrix} -1 & 1 & 0 & 0 & 0 \\ 0 & 0 & 0 & 0 & 0 \\ 0 & 0 & 0 & 0 & 0 \\ 0 & 0 & 0 & 0 & 0 \\ 0 & 0 & 0 & 0 & 0 \end{bmatrix}$$

Applying $W$ token-wise gives the desired result. $\square$

**Lemma 7** (Transformer Parallelization). *Let $m_1, m_2, l_1, l_2, L_{\text{FFN1}}, L_{\text{FFN2}}, w_{\text{FFN1}}, w_{\text{FFN2}}, d_{embd} \in \mathbb{N}$. Fix transformer blocks $B \in \mathcal{B}(m_1, L_{\text{FFN1}}, w_{\text{FFN1}})$ with input $H_1 \in \mathbb{R}^{d_{embd} \times l_1}$ and $B_2 \in \mathcal{B}(m_2, L_{\text{FFN2}}, w_{\text{FFN2}})$ with input $H_2 \in \mathbb{R}^{d_{embd} \times l_2}$. Further suppose both inputs are structured such that each token $h_{i,t}$ has an interaction term $\mathcal{I}_t$ and constant term $1$. Further suppose all attention heads in both $B_1$ and $B_2$ are implemented using the Interaction Lemma and all FFN layers are either Gating FFN layers or are independent of the interaction terms. Then there exists $B_3 \in \mathcal{B}(m_1 + m_2, \max(L_{\text{FFN1}}, L_{\text{FFN2}}) + 2, w_{\text{FFN1}} + w_{\text{FFN2}})$ which takes as input $H_3 \in \mathbb{R}^{d_{embd} \times (l_1+l_2)}$ such that*

$$B_3(H_3) = B_3\left([H_1' \quad H_2']\right) = [B_1(H_1) \quad B_2(H_2)] \in \mathbb{R}^{d_{embd} \times (l_1+l_2)}$$

*where $H_i' \in \mathbb{R}^{d_{embd} \times l_i}$ is the same as $H_i$ except for the interaction terms where we have $\mathcal{I}_{3,t}' = (cos(\frac{t}{l_1+l_2} \frac{\pi}{2}), sin(\frac{t}{l_1+l_2} \frac{\pi}{2}))$ for $1 \leq t \leq l_1+l_2$. If $B_3$ satisfies this relationship we say $B_3$ **parallelizes** $B_1$ and $B_2$.*

*Proof of Lemma 7.* Let $B_1(H_1) = \text{FFN}_1(\text{MHA}_1(H_1)) + \text{MHA}_1(H_1) + H_1$ and $B_2(H_2) = \text{FFN}_2(\text{MHA}_1(H_2)) + \text{MHA}_2(H_2) + H_2$. Let $A_{1,i_1}$, $1 \leq i_1 \leq m_1$, denote the attention heads of $\text{MHA}_2$ and $A_{2,i_2}$, $1 \leq i_2 \leq m_2$, denote the attention heads of $\text{MHA}_2$. We construct $B_3$ as follows.

First we construct the new input embedding matrix $H_3$ from $H_1$ and $H_2$. For $1 \leq t \leq l_1$ set $h_{3,t} = h_{1,t}$ except we define a new interaction term $\mathcal{I}_{3,t} = (cos(\frac{t}{l_1+l_2} \frac{\pi}{2}), sin(\frac{t}{l_1+l_2} \frac{\pi}{2}))$. For

$l_1 + 1 \leq t \leq l_1 + l_2$ define $h_{3,t} = h_{3,t-l_1}$ except where we again must change only the interaction term. The main difficulty in defining $\text{MHA}_3$ will then simply be updating the interaction kernels of the Interaction Lemma structured attention heads to respect the new interaction terms $\mathcal{I}_t$, $1 \leq t \leq l_1 + l_2$.

We now define the attention heads $\text{A}_{3,j}$ in $\text{MHA}_3$. When $1 \leq j \leq m_1$ write $Q_{3,j} = Q'_{1,j}$ where $Q'_{1,j}$ is the same $Q_{1,j}$ except for a new interaction kernel $Q''^I_{1,j}$. Let $h_{1,s_j}$, $s_j \in \{1, ..., l_1\}$, be the source token in $H_1$ interacting with $Q'_{1,j}$ and $h_{1,t_j}, t_j \in \{1, ..., l_1\}$, be the target token of the old interaction kernel $Q^I_{1,j}$ (recall $Q^I_{1,j}$ is defined as the rotation of $\mathcal{I}_{1,s_j}$ onto $\mathcal{I}_{1,t_j}$). Then define $Q''^I_{1,j}$ as a rotation onto the new target interaction kernel $\mathcal{I}_{3,t_j}$. Similarly define $K_{3,j} = K'_{1,j}$ where $K'_{1,j} = K_{1,j}$ up to a difference in interaction kernels. Let $K^I_{1,j}$ be the old interaction kernel which is a projection on to the target interaction term $\mathcal{I}_{1,t_j}$. Then define the new interaction kernel as a projection onto the new target interaction term $\mathcal{I}_{3,t_j}$. Simply set $V_{3,j} = V_{1,j}$. Now we can check $\text{A}_{3,j}(H_3)^{1,...,l_1} = \text{A}_{1,j}(H_1)$ up to the interaction terms. Compute

$$
\begin{aligned}
\text{A}_{3,j}(h_{3,s_j}) &= \sum_{k=1}^{l_1+l_2} \sigma(\langle Q_{3,j}h_{3,s_j}, K_{3,j}h_{3,k}\rangle)V_{3,j}h_{3,k} \\
&= \sigma(\langle Q_{3,j}h_{3,s_j}, K_{3,j}h_{3,t_j}\rangle)V_{3,j}h_{3,t_j} \\
&= \sigma(\langle Q_{1,j}h_{1,s_j}, K_{1,j}h_{1,t_j}\rangle)V_{1,j}h_{1,t_j} = A_{1,j}(h_{1,s_j})
\end{aligned}
$$

where the second equality comes from the sparse construction of $\text{A}_{3,j}$, the third equality comes from the construction of $Q_{3,j}, K_{3,j}$, and the last equality comes from the definition of the sparsely interacting $\text{A}_{1,j}$. Otherwise $\text{A}_{3,j}(h_{3,t}) = 0$ for $t \neq s_j$. Thus we can conclude $\text{A}_{3,j}(H_3) = \text{A}_{1,j}(H_1)$ up to interaction terms. The case $l_1 + 1 \leq j \leq l_1 + l_2$ proceeds analogously. Thus we conclude

$$\text{MHA}_3(H_3) = [\text{B}_1(H_1) \quad \text{B}_2(H_2)]$$

up to the interaction terms. Now we must construct the parallelized feed-forward layer $\text{FFN}_3$ from $\text{FFN}_1$ and $\text{FFN}_2$. Let $W_{1,k}$ be the $k$th weight matrix of $\text{FFN}_1$. By assumption we know $W_{1,k}$ is either a gating layer or is independent of token interaction terms. If $W_{1,k}$ is independent of interaction terms we can simply set $W'_{1,k} = W_{1,k}$. Otherwise if $W_{1,k}$ is a gating layer on some half-space $\{1, ..., p\}, 1 \leq p \leq l_1$, we must pick a new pivot vector $v' = \mathbf{R}_{\frac{\pi}{2}}\left(\frac{\mathcal{I}_{3,p}+\mathcal{I}_{3,p+1}}{\|\mathcal{I}_{3,p}+\mathcal{I}_{3,p+1}\|_2}\right)$. Set $W'_{1,k}$ accordingly to the new $v'$. We may similarly choose $W'_{2,k}$ as we did for $W_{1,k}$. Then we set the new layer $W_{3,k}$ as

$$W_{3,k} = \begin{bmatrix} W'_{1,k} & \mathbf{0} \\ \mathbf{0} & W'_{2,k} \end{bmatrix} \in \mathbb{R}^{(w_{\text{FFN}1}+w_{\text{FFN}2})\times(w_{\text{FFN}1}+w_{\text{FFN}2})}, \quad b_{3,k} = \begin{bmatrix} b_{1,k} \\ b_{2,k} \end{bmatrix} \in \mathbb{R}^{w_{\text{FFN}1}+w_{\text{FFN}2}}$$

We must also construct an embedding network $W_{3,0}$

$$W_{3,0} = \begin{bmatrix} I_{d_{embd}} \\ I_{d_{embd}} \end{bmatrix} \in \mathbb{R}^{2d_{embd}t \times d_{embd}}$$

which duplicates the input token $h_{3,t} \in \mathbb{R}^{d_{embd}}$ for parallel processing. Finally, via lemma 4, we may construct a two-layer network $\text{FFN}_{1,gating}$ such that $\text{FFN}_{1,gating}(h_{3,t}) = h_{3,t}$ if $1 \leq t \leq l_1$ and has zeroed out data rows otherwise. We similarly define $\text{FFN}_{2,gating}$ for $l_1 + 1 \leq t \leq l_1 + l_2$. Finally we define the output layer $W_{3,\max(L_{\text{FFN}1},L_{\text{FFN}2})+2}$ as

$$[I_{d_{embd}} \quad I_{d_{embd}}] \in \mathbb{R}^{d_{embd} \times 2d_{embd}}$$

Thus, for input token $h_{3,t}$, $1 \leq t \leq l_1$, we have

$$
\begin{aligned}
\text{FFN}_3(h_{3,t}) &= W_{3,\max(L_{\text{FFN}1},L_{\text{FFN}2})+2}\left[\text{FFN}_{1,gating}\text{FFN}_1 \quad \text{FFN}_{2,gating}\text{FFN}_2\right]W_{3,0}h_{3,t} \\
&= W_{3,\max(L_{\text{FFN}1},L_{\text{FFN}2})+2}\left[\text{FFN}_{1,gating}\text{FFN}_1 \quad \text{FFN}_{2,gating}\text{FFN}_2\right]\begin{bmatrix} h_{3,t} \\ h_{3,t} \end{bmatrix} \\
&= W_{3,\max(L_{\text{FFN}1},L_{\text{FFN}2})+2}\begin{bmatrix} \text{FFN}_1(h_{3,t}) \\ 0 \end{bmatrix} \\
&= \text{FFN}_1(h_{3,t})
\end{aligned}
$$

as desired. The case $l_1 \leq t \leq l_1 + l_2$ proceeds analagously.

$\square$

# H   Other Lemmas

The following lemma [Chen et al., 2022, Lemma 2] is used for our construction of charts in the proof of Theorem 2.

**Lemma 8** (Local Diffeomorphism)**.** *Suppose Assumption 1 holds for manifold $\mathcal{M}$ and $r \leq \tau/4$. Then the local neighborhood $U_n = B(c_n, r) \cap \mathcal{M}$ is diffeomorphic to a subset of $\mathbb{R}^d$. In particular, the orthogonal projection $P_n$ onto the tangent space $T_{c_n}(\mathcal{M})$ is a diffeomorphism.*

