# OpenReview forum: "Understanding Scaling Laws with Statistical and Approximation Theory for Transformer Neural Networks on Intrinsically Low-dimensional Data"
_NeurIPS.cc/2024/Conference — NeurIPS 2024 poster_

### Official Review · Reviewer_e2w2 · 2024-07-07

**Soundness:** 3
**Presentation:** 3
**Contribution:** 3
**Rating:** 6
**Confidence:** 3

**Summary:**

This work derived a generalization error bound of using transformer architecture to estimate beta-Holder continuous functions. The bounds depend on the intrinsic dimension of the data. The generalization error can be decomposed into an approximation error and a variance error. They further showed that TFs with finite number of blocks can approximate arbitrary beta-Holder continuous functions up to any precision.  Both their generalization and error bound and approximation error bound exhibit power-law decay, with exponents depending on the smoothness and intrinsic dimension. Empirical results on real datasets also validated their theoretical findings.

**Strengths:**

The paper present novel approximation theory for transformers in terms of the intrinsic dimension.  Empirical observations are well-aligned with the theory.

**Weaknesses:**

1. The statement that TFs can approximate $f$ to any precision $\epsilon$ with finite number of blocks is a bit inaccurate and misleading. Namely,  Theorem 2 assumes $L_{FFN}$ is of order $O(\log(1/\epsilon))$. This means in the standard TF architecture, we would have $O(\log(1/\epsilon))$ number of FFN-attention blocks instead of only a finite number.  It would be great if the author can clarify this point.

**Questions:**

1. What is the number of hidden neurons in FFNs?
2. Does the result in (3) matches the minimax lower bound for estimating beta-holder class? What is the gain of using TFs instead of simple nonparametric methods to estimate $f$?

**Limitations:**

1. As the author discussed, this work doesn't consider the training dynamics of the learning problem.
2. The exponents in (3), (4) highly depend on  the smoothness of $f$ (i.e., $\beta$) and the intrinsic dimension of the data. However, the exact value of $\beta$ is hard to predict in practice. Is there any practical way to estimate $\beta$?

---

> ### Author Rebuttal · Authors · 2024-08-06
>
> We thank the reviewer for their thorough repsonse! We address some of your comments below.
>
> **Strengths:**
>
> 1. The paper present novel approximation theory for transformers in terms of the intrinsic dimension. Empirical observations are well-aligned with the theory.
>
> We are glad the reviewer found our approximation theory novel and our empirical findings well-aligned with our theory!
>
> **Weaknesses:**
>
> 1. The statement that TFs can approximate to any precision with finite number of blocks is a bit inaccurate and misleading. Namely, Theorem 2 assumes is FFN depth is of order $O(\log(\epsilon^{-1}))$. This means in the standard TF architecture, we would have $O(\log(\epsilon^{-1}))$ number of FFN-attention blocks instead of only a finite number. It would be great if the author can clarify this point.
>
> This is a good point, the number FFN layers per transformer block is $\log(\epsilon^{-1})$. However, we recall that the weight-matrices in each FFN are of constant size (5x5 since $d_{embd} = 5$) and thus the number of parameters in each FFN layers is negligible. The vast majority of parameters are used in highly-parallel attention layers of which we only need $\log(d)$ layers. Note: a transformer *block* consists of an attention layer and then a FFN network.
>
> **Questions:**
>
> 1. What is the number of hidden neurons in FFNs?
>
> The number of parameters in each FFN is  $d_{embd}^2\cdot \log(\epsilon^{-1}) = O(\log(\epsilon^{-1}))$.
>
> Note $d_{embd} = 5$.
>
> 2. Does the result in (3) matches the minimax lower bound for estimating beta-holder class? What is the gain of using TFs instead of simple nonparametric methods to estimate $f$?
>
> Yes, our bound matches the min-max rate up to logarithmic factors [1]. For Holder functions, both transformer and many nonparametric methods (kernel regression and piecewise polynomial regression) can achieve the min-max rate up to logarithmic factors. However, transformer has achived remarkable success in large language models, which makes a theoretical understanding of transformer significant and meaningful. It is well known that simple nonparametric methods can not achive the same performance as transformers in real-world applications with large scale complex data.
>
> We believe (but have not proved) that transformer networks are more *adaptive* to the regularity of the target function at different areas of the domain. For simple nonparametric methods, such as kernel methods, often one must choose a kernel width, making adaptivity more difficult.
>
>
>
> **Limitations:**
>
> 1. As the author discussed, this work doesn't consider the training dynamics of the learning problem.
>
> To our best knowledge, the training dynamics of multi-layer transformers is an open-question. We agree studying transformer dynamics is of great interest, but this is not the goal of the current paper.
>
> 2.. The exponents in (3), (4) highly depend on the smoothness of $f$ (i.e., $\beta$) and the intrinsic dimension of the data. However, the exact value of $\beta$ is hard to predict in practice. Is there any practical way to estimate $\beta$.
>
> In general we expect the target function to have some regularity as otherwise it would be impossible to estimate [1]. Assuming the output varies according to the variation of the input, Holder continuity is the proper and mostly widely considered assumption in nonparametric estimation. A Lipschitz assumption with $\beta = 1$ implies the variation of the output for the target function is proportional to the variation of the input which is widely used in literature. In general we are unaware of any methods for measuring the Holder index of the target function from samples.
>
>
> **References:**
>
> [1] A Distribution-Free Theory of Nonparametric Regression, https://link.springer.com/book/10.1007/b97848

---

> > ### Comment · Reviewer_e2w2 · 2024-08-13
> >
> > Thanks the authors for their response. My questions are partially addressed and I will maintain my score.

---

### Official Review · Reviewer_DzYy · 2024-07-11

**Soundness:** 4
**Presentation:** 3
**Contribution:** 3
**Rating:** 6
**Confidence:** 3

**Summary:**

This document appears to be a research paper on predicting scaling laws for transformer neural networks, particularly when applied to data with low intrinsic dimensionality. Here are some key points:
1. The paper aims to establish mathematical theories to explain and predict scaling laws observed in transformer models like large language models (LLMs).
2. It presents both statistical estimation and approximation theories for transformers when the input data lies on a low-dimensional manifold.
3. The main results predict power law relationships between generalization error and both training data size and network size, where the power depends on the intrinsic dimension d of the training data.
4. A key finding is that the transformer architecture constructed in their theory only requires logarithmic depth in d, which is an advantage over feedforward networks.
5. The authors test their theoretical predictions empirically by training LLMs on natural language datasets and find close agreement with observed scaling laws.
6. The paper argues that the intrinsic dimension of data is crucial in determining transformer scaling laws both theoretically and empirically. It provides formal definitions and theorems around transformer networks, generalization bounds, and approximation capabilities. The work aims to bridge gaps between theory and practice in understanding neural network scaling, leveraging the low-dimensional structure often present in real-world data

**Strengths:**

1. The paper provides a rigorous mathematical framework for understanding transformer scaling laws, which has been a significant open question in the field.
2. The work attempts to reconcile theoretical predictions with empirical observations, particularly in the context of large language models.
3. By incorporating the intrinsic dimension of data, the theory provides a more nuanced understanding of scaling laws that aligns better with real-world observations.
4. The authors test their theoretical predictions on actual language models, providing evidence for the practical relevance of their work.

**Weaknesses:**

1. Simplified assumptions: The theory relies on assumptions about data lying on a low-dimensional manifold, which may not always hold in practice, especially for complex, high-dimensional data like natural language.
2. Limited scope: The paper focuses primarily on regression tasks, while many practical applications of transformers (like language modeling) involve more complex objectives.
3. Generalizability: It's unclear how well the theory generalizes to other types of transformer architectures or variations that differ from the specific formulation used in the paper.
4. Computational complexity: The paper doesn't deeply address the computational aspects of their proposed methods, which could be a practical limitation for very large models.
5. Data dimension estimation: The reliability and stability of estimating the intrinsic dimension of complex datasets (like text) remains a challenge and could impact the practical application of the theory.

**Questions:**

1. Can you state the process of estimating the intrinsic dimension of textual datasets which is important for your work? If the dimension depends on the pre-trained model, it seems the posterior estimate is not supering to be good and its practicability seems weak.
2. In Theorem 1 and 2, the parameters of the network should satisfy a specific magnitude assumption, is this assumption significant and practical in applications?
3. In figure2, only 5 points are presented in each subfigure and the x-axis’s scope is different, how does the empirical result appear in other regions?
4. Why did you assume the Lipschitz regularity of language modeling objective equals 1? Is there any explanation for it?
5. The results shown in figure3 seem not good so what’s the significant reason?
6. The order of subscripts seems incorrect in Table 1.

**Limitations:**

1. The assumptions in the theorem seem not practical.
2. The construction in the proof of estimation theory is hard and not practical in empirical inference.
3. The estimation of the intrinsic dimension is not clear.

---

> ### Author Rebuttal · Authors · 2024-08-06
>
> **Strengths:**
>
> 1. The paper provides a rigorous mathematical framework for understanding transformer scaling laws, which has been a significant open question in the field. [...] The authors test their theoretical predictions on actual language models, providing evidence for the practical relevance of their work.
>
> We are glad the reviewer recognizes our work as making progress on a significant open question in the field and that the reviewer recognizes the importance of our theory being validated with experiments on actual language models.
>
> **Weaknesses:**
>
> 1. Simplified assumptions: The theory relies on assumptions about data lying on a low-dimensional manifold, which may not always hold in practice, especially for complex, high-dimensional data like natural language.
>
> The widely supposed *manifold hypothesis* suggests that most natural data, such that images and text, lie on low-dimensional manifolds embedded in a high-dimensional ambient space. Many papers [2,3] find evidence suggesting this is true for image data and show this helps explain the success of deep-learning in learning high-dimensional data with low-dimensional structures. In particular, we and others [4,5] find evidence that low-dimensional structures exist in textual data once it is embedded by a model. (15 ID vs. 768 ambient dimension).
>
> 2. Limited scope: The paper focuses primarily on regression tasks, while many practical applications of transformers (like language modeling) involve more complex objectives.
>
> Despite being proved for regression, our result also sheds light on real world applications using next-token prediction (classification). Theoretically it is known that results on regression problems can be used to estalish results for classification problems [1, Theorem 1.1] since classification essentially depends on estimation of the probability function for classification. This allows our results on regression for transformers to applied directly to next-token prediction which translates to practical applictaions. Additionally, we remark the regression objective itself is commonly used in many real-world scenarios e.g. training a transformer diffusion backbone.
>
> 3. Generalizability: It's unclear how well the theory generalizes to other types of transformer architectures or variations that differ from the specific formulation used in the paper.
>
> Our theory crucially relies upon a number of key lemmas (e.g. the Interaction lemma facilitating pairwise interaction between tokens in the attention mechanism) to develop our novel approximation theory and a novel covering number result to develop the statistical theory. Both of these are general and can be extended to different transformer architectures. Further, our experimental results suggest that, despite the architectures being used in practice differing from our theory, we are still able to make good predictions about the scaling behavior of these models both in terms of the model size and the number of training samples.
>
> 4. Computational complexity: The paper doesn't deeply address the computational aspects of their proposed methods, which could be a practical limitation for very large models.
>
>
> Our paper aims to predict scaling laws of transformer neural network which crucuially relies on the intrinsic dimension of training data. Typically, to estimate a scaling exponent empirically, one must pre-train a family of models across different data/model sizes. In contrast, with our theory, one only needs to pre-train a single model which can then be used to estimate intrinsic dimension from which all scaling can be theoretically predicted. In comparison with pre-training, which can take multiple weeks for the largest models, estimating the intrinsic dimension takes only a couple minutes.
>
> 5. Data dimension estimation: The reliability and stability of estimating the intrinsic dimension of complex datasets (like text) remains a challenge and could impact the practical application of the theory.
>
> Estimating the intrinsic dimension via model embeddings is a relatively established practice [1,4,5]. Further, in the case of text we have no other choice as existing ID estimation algorithms are designed for continuous data. However this should not be taken to mean that textual data does not have low-dimensional structure. While using embeddins to measure this low-dimensional structure may introduce some noise, we expect a model (such as an LLM) to have learned good enough representations to preserve the majority of the low-dimensional structure. We do ablations at the end of the paper to examine how sensitive this estimation is to various hyperparameters and find it is relatively stable. While embeddings from any layer could be chosen, we choose the final layer to stay consistent with prior work.
>
> **Please refer to our comment for the remainder of our response**

---

> ### Author Response · Authors · 2024-08-06
> **Rebuttal cont.**
>
> **Questions:**
>
> 1. Can you state the process of estimating the intrinsic dimension of textual datasets which is important for your work? If the dimension depends on the pre-trained model, it seems the posterior estimate is not supering to be good and its practicability seems weak.
>
> We refer the reviewer our response to the above bullet point. Maximum Likelihood Estimatio (MLE) gives rise to a non-linear measure of intrinsic dimension by applying the principle of maximum likelihood to the distances between close neighbors [6]. It has been commonly used in prior works [2,3] to estimate the intrinsic dimension of natural data and assess impact on downstream model performance. Our theoretical predictions of scaling laws for transformers using these estimates of ID are closer to empirical predictions in comparison with existing work [2].
>
>
>
> 2. In Theorem 1 and 2, the parameters of the network should satisfy a specific magnitude assumption, is this assumption significant and practical in applications?
>
> In practice neural networks are often trained with implicit regularization such as weight-normalization to prevent the weight parameters from blowing up. This makes it easy for such networks to satisfy our assumption on the magnitude of their weights. Note: Our magnitude upper bound is fairly large ($O(dn^{\frac{2}{2\beta +d}}M)$), allowing for plenty of flexibility during model training.
>
> 3. In figure2, only 5 points are presented in each subfigure and the x-axis’s scope is different, how does the empirical result appear in other regions?
>
> We find that transformer scaling laws are stable in certain ranges of data and model size as shown in Figures 2 and 3. It is well known that, empirically, scaling laws can break down when the data and model size further increases [7], probably due to optimization error or computational limit, which is not the focus of our paper.
>
> 4. Why did you assume the Lipschitz regularity of language modeling objective equals 1? Is there any explanation for it?
>
> In general we expect the target function to have some regularity as otherwise it would be impossible to estimate [1]. Assuming the output varies according to the variation of the input, Holder continuity is the proper and mostly widely considered assumption in nonparametric estimation. A Lipschitz assumption with $\beta = 1$ implies the variation of the output for the target function is proportional to the variation of the input which is widely used in literature. In general we are unaware of any methods for measuring the Holder exponent of the target function from samples.
>
> 5. The results shown in figure3 seem not good so what’s the significant reason?
>
> The results in Figure 3 have $\pm 0.03$ i.e. nearly as accurate as Figure 2 with $\pm 0.02$ error. We regard both as fairly accurate predictions of the scaling exponents.
>
> 6. The order of subscripts seems incorrect in Table 1.
>
> Thank you for flagging this! The values for $\alpha_N$ and $\alpha_D$ should be flipped.
>
> **Limitations:**
>
> 1. The construction in the proof of estimation theory is hard and not practical in empirical inference.
>
> Our construction in the approximation theory shows the universal approximation ability of transformers for Holder functions. These results are significant because **they allow us to quantitatively and precisely control the approximation error $\epsilon$ as a function of the transformer architecture size**. Importantly, this construction to control the approximation does not say anything about the learned parameters of the empirical risk minimizer or the parameters learned during the optimizaion process. We view this as an advantage of our theory, as we do not need to know anything about the empirical risk minimizer's learned parameters to control its generalization error.
>
>
> 2. The estimation of the intrinsic dimension is not clear.
>
> Re-iterating, MLE gives rise to a non-linear measure of intrinsic dimension by applying the principle of maximum likelihood to the distances between close neighbors [6]. It has been commonly used in prior work [2,3] to estimate the intrinsic dimension of natural data and assess impact on downstream model performance.
>
> **References:**
>
> [1]  A Distribution-Free Theory of Nonparametric Regression, https://link.springer.com/book/10.1007/b97848
>
> [2]  Scaling Laws from the Data Manifold Dimension, https://jmlr.org/papers/v23/20-1111.html
>
> [3] The Intrinsic Dimension of Images and Its Impact on Learning, https://arxiv.org/abs/2104.08894
>
> [4] The Shape of Learning: Anisotropy and Intrinsic Dimensions in Transformer-Based Models, https://arxiv.org/abs/2311.05928
>
> [5] An Intrinsic Dimension Perspective of Transformers for Sequential Modeling, https://openreview.net/forum?id=0UzYWLzPBjA
>
> [6] Maximum Likelihood Estimation of Intrinsic Dimension, https://www.stat.berkeley.edu/~bickel/mldim.pdf
>
> [7] Scaling Laws for Neural Language Models, https://arxiv.org/abs/2001.08361

---

### Official Review · Reviewer_1VDq · 2024-07-12

**Soundness:** 3
**Presentation:** 4
**Contribution:** 3
**Rating:** 6
**Confidence:** 1

**Summary:**

This paper makes a series of contributions:

- Transformer Generalization Error: Loosely speaking, assuming a transformer is trained to approximate a Holder function in a regression setting, and assuming the data lives on the low dimensional manifold, then the generalization error of the transformer is upper bounded in a particular manner (Theorem 1)
- New Covering Number for Transformers (although I believe this is not presented in the main text)
- Predicting Transformer Empirical Scaling Laws
- Some investigations into how transformer hyperparameters affect the estimated intrinsic dimensionality of text data


I am not familiar with these more theoretical methods, and I have never taken a course on differential geometry. Consequently, my review will be quite limited. I focus more on the empirical methods since I am more familiar with these.

**Strengths:**

Note: I am not familiar with these more theoretical methods, and I have never taken a course on differential geometry. Consequently, my review will be quite limited. I focus more on the empirical methods since I am more familiar with these.

- Having minor familiarity with Sharma and Kaplan, I think this work is a great extension towards language modeling.
- The paper is clearly very well written and thorough
- Figure 1 is visually nice (although I don’t have the background required to understand it)

**Weaknesses:**

Note: I am not familiar with these more theoretical methods, and I have never taken a course on differential geometry. Consequently, my review will be quite limited. I focus more on the empirical methods since I am more familiar with these.

- “Overall, we find the estimated ID is fairly stable across each factor.” -> Looking at Figure 4, this feels very wrong to me. In 3 out of 4 subplots, the intrinsic dimensionality clearly seems to be changing with no asymptotic value in sight.
- [nit] Figures 2 and 3: Please use a log scale. Don’t log transform the number of samples and then plot the log-transformed variable linearly.

I have many questions (below) that may become weaknesses, but I don't wish to penalize the authors if I've misunderstood their work.

**Questions:**

- Looking at equation (3) and (4), the scaling exponents $\alpha_D, \alpha_N$ are partially determined by $\beta$. Later, $\beta$ is set to 1. How is this justified? Why would this be true empirically?
- The paper critically concerns estimating the intrinsic dimensionality of the pretraining dataset D, but then claims this cannot be done directly for textual datasets. Why? No justification is given.
- Following the above bullet, the paper then estimates the intrinsic dimension of the data by estimating the intrinsic dimension of the output token embedding matrix. Why is using this alternative quantity reasonable?
- I’m not familiar with the “Maximum Likelihood Estimation ID algorithm”. What is this algorithm? Why was it chosen? How do the results depend on this choice? The ID estimation methods I'm more familiar with are quantities like participation ratio - how do other method
- Line 71: Is the word “layers” missing? i.e. “requiring only O(log(d)) layers independent of the desired accuracy…”
- Figure 3: Why are there 5 GPT2 model sizes? HuggingFace offers only 4.
- Figure 3: Why are only 4 Pythia models used? I believe that there are 8.
- What in this paper is a prediction? To me, the hallmark of a scientific prediction is a statement regarding what one should expect beyond already-existing data, followed by new experiments to confirm the new behavior. With the exception of Figure 2, I don't see much in the way of new predictions or new experiments to confirm predictions, and even Figure 2 is relatively weak in the sense of external verifiability.

---

> ### Author Rebuttal · Authors · 2024-08-06
>
> **Strengths:**
>
> - Having minor familiarity with Sharma and Kaplan, I think this work is a great extension towards language modeling.
>
> We thank the reviewer for recognizing this work as a great extension towards lanugage modeling! However, we want to emphasize the main contribution of our work is **theoretical** (as oppsed to [1] which is empirical). Via our novel approximation and statistical theory, we are able to establish theoretical bounds on Transformer scaling laws. We then do experimentation, extending similar experiments in [1] to transformers, demonstrating this theory holds well in practice.
>
> **Weaknesses:**
>
> 1. I am not familiar with these more theoretical methods, and I have never taken a course on differential geometry. Consequently, my review will be quite limited. I focus more on the empirical methods since I am more familiar with these.
>
> Again, we would like to emphasize that the main contributions of this paper are **theoretical**. This includes our statistical estimation theory in Theorem 1 which crucially relies upon a novel and non-trivial covering number result for transformers (Lemma 2) and a novel approximation theory in Theorem 2 which is constructed from a number of key sub-lemmas (e.g. see Lemma 3 - Interaction Lemma) allowing for sophisticated interaction of multiple tokens which is also of independent interest. We believe these results constitute a signficant step forward in the theoretical understanding of transformers.
>
>
> 2. “Overall, we find the estimated ID is fairly stable across each factor.” -> Looking at Figure 4, this feels very wrong to me. In 3 out of 4 subplots, the intrinsic dimensionality clearly seems to be changing with no asymptotic value in sight.
>
> Since the measurement of intrinsic dimension (**ID**) is essential to our theoretical predictions in practice, our ablations in Figure 4 are meant to demonstrate the stability of our measurements of intrinsic dimension under various **reaslistic** model hyperparameters - **not to suggest any asymptotic behavior**. In general the estimation of intrinsic dimension can be fairly noisy, depending variably even on the hyperparameters chosen in the ID estimation algorithm [1,2]. For example, [2] finds the ID of imagenet varies anywhere between 20-40 depending on the algorithmic hyperparameters used. Our estimates for the ID vary only by $\pm 5$ units across all model parameters we considered, motivating our remark that the estimation is fairly stable.
>
> 3. [nit] Figures 2 and 3: Please use a log scale. Don’t log transform the number of samples and then plot the log-transformed variable linearly.
>
> Our results show that the (squared) generalization error (**SGE**) can be bounded by power-laws in the number of samples $n$ as $SGE \lesssim n^{-\alpha_D}$. Then taking log of both sides yields $\log(SGE) \lesssim -\alpha_D \log(n)$ which exactly bounds the (log) SGE by the (log)-linear line in the number of samples with slope given by the scaling exponent $-\alpha_D$. This should hopefully clarify why we plot in log-log scale.
>
> **Questions:**
>
> 1. Looking at equation (3) and (4), the scaling exponents are partially determined by $\beta$. Later, $\beta$ is set to 1. How is this justified? Why would this be true empirically?
>
> Recall $\beta$ represents the Holder regularity index of the target function $f$. In general we expect the target function to have some regularity as otherwise it would be impossible to estimate [3]. Assuming the output varies according to the variation of the input, Holder regularity is the proper and mostly widely considered assumption in nonparametric estimation. A Lipschitz assumption with $\beta = 1$ implies the variation of the output for the target function is proportional to the variation of the input which is also widely used in literature. In general we are unaware of numerical methods for measuring the Holder index of the target function from samples.
>
> 2. The paper critically concerns estimating the intrinsic dimensionality of the pretraining dataset D, but then claims this cannot be done directly for textual datasets. Why? No justification is given.
> 3. The paper then estimates the intrinsic dimension of the data by estimating the intrinsic dimension of the output token embedding matrix. Why is using this alternative quantity reasonable?
>
> Estimating the intrinsic dimension via model embeddings is a relatively established practice [1]. Further, in the case of text we have no other choice as existing ID estimation algorithms are designed for continuous data. However this should not be taken to mean that textual data does not have low-dimensional structure. While using embeddins to measure this low-dimensional structure may introduce some noise, we expect a model (such as an LLM) to have learned good enough representations to preserve the majority of the low-dimensional structure. We do ablations at the end of the paper to examine how sensitive this estimation is to various hyperparameters and find it is relatively stable. While embeddings from any layer could be chosen, we choose the final layer to stay consistent with prior work.
>
> **Please refer to our comment for the remainder of our response**

---

> ### Author Response · Authors · 2024-08-06
> **Rebuttal cont.**
>
> 4. I’m not familiar with the “Maximum Likelihood Estimation ID algorithm”. What is this algorithm? Why was it chosen? How do the results depend on this choice? The ID estimation methods I'm more familiar with are quantities like participation ratio - how do other method
>
> Participation ratio is a linear measure of intrinsic dimension. Maximum Likelihood Estimation (MLE) [4] measures the intrisic dimension for low-dimensional nonlinear models by applying the principle of maximum likelihood to the distances between close neighbors. MLE has been commonly used in prior works [1,2] to estimate the intrinsic dimension of natural data and assess impact on downstream model performance.
>
> 4. Figure 3: Why are there 5 GPT2 model sizes? HuggingFace offers only 4.
>
> For GPT2 we reported the results published in [5].
>
> 5. Figure 3: Why are only 4 Pythia models used? I believe that there are 8.
>
> We used only four models because we were already able to extract an empirical scaling law over two  orders of model size. However, if the reviewer feels including the full suite would strengthen our experimental results we would be happy to plot more intermediate model sizes.
>
> 6. What in this paper is a prediction? To me, the hallmark of a scientific prediction is a statement regarding what one should expect beyond already-existing data, followed by new experiments to confirm the new behavior. With the exception of Figure 2, I don't see much in the way of new predictions or new experiments to confirm predictions, and even Figure 2 is relatively weak in the sense of external verifiability.
>
> We believe that, in science, the role of theory is not just to predict unobserved events but to provide **rigorous explanation** for observed phenomena. We would describe the latter situation as the theory **correctly predicting** the observed phenomena. Concretely, in our case, we refer to the exponents $\alpha_D, \alpha_N$ as being *predicted* by our theory as a function of input quantities $(\beta, d)$. As you point out, we make three previously unreported predictions in Figure 2. These are **new** in the sense that scaling laws on these datasets have not been previously reported in the literature. Additionally, to our knowledge, scaling behavior for the Pythia suite has also not been previously analyzed. Practically speaking, it would be very difficult to produce a new dataset/model substantially different from what is currently used in industry as, from the scaling perspective, this would require the collection of a tens of trillions token dataset or the training of a hundred-billion parameter model. Our goal, as a theory paper, is instead to rigorously explain existing phenomena and validate with novel, modestly-sized experiments on existing data. Further, these experiments are fully reproducible, as all code and data is open-source.
>
> **References:**
>
> [1]  Scaling Laws from the Data Manifold Dimension, https://jmlr.org/papers/v23/20-1111.html
>
> [2] The Intrinsic Dimension of Images and Its Impact on Learning, https://arxiv.org/abs/2104.08894
>
> [3] A Distribution-Free Theory of Nonparametric Regression, https://link.springer.com/book/10.1007/b97848
>
> [4] Maximum Likelihood Estimation of Intrinsic Dimension, https://www.stat.berkeley.edu/~bickel/mldim.pdf
>
> [5] Scaling Laws for Neural Language Models, https://arxiv.org/abs/2001.08361

---

> ### Comment · Reviewer_1VDq · 2024-08-10
> **Response to Authors' Rebuttal (Part 1)**
>
> I thank the authors for their rebuttal. Responding sequentially:
>
> > Again, we would like to emphasize that the main contributions of this paper are **theoretical**.
>
> I understand. Sadly, I'm not well equipped to evaluate the maths, which is why my confidence is so low. I did not bid on this paper, but I will try my best regardless.
>
> > A Lipschitz assumption with $\beta = 1$ implies the variation of the output for the target function is proportional to the variation of the input which is also widely used in literature. In general we are unaware of numerical methods for measuring the Holder index of the target function from samples.
>
> Without knowledge of any problem with this choice or any alternatives, this seems reasonable.
>
> > Estimating the intrinsic dimension via model embeddings is a relatively established practice [1]. Further, in the case of text we have no other choice as existing ID estimation algorithms are designed for continuous data.
>
> After reading your response, I wonder if there might be some miscommunication here.
>
> Sharma and Kaplan state that to estimate ID: "we use the activations from the last token in each
> sequence to measure the ID, though the ID does not vary significantly across token positions
> (see figure 10)", whereas your lines 219-220 state " we will estimate the intrinsic dimension of the input data by estimating the intrinsic dimension of token embeddings." I interpreted your sentence to mean that you took either the embedding matrix or the unembedding matrix (which may be the same, if you are tying/sharing the matrices).  After rereading the paragraph on lines 219 to 230, I think you are using the word "embedding" when Sharma and Kaplan (as well as I) use "activations".
>
> Could you please clarify what exactly you are doing? Are you using activations or the embedding vectors that comprise the (un)embedding matrix?
>
> If you are using activations, this seems far more sensible to me.
>
> > [nit] Figures 2 and 3: Please use a log scale. Don’t log transform the number of samples and then plot the log-transformed variable linearly.
>
> I fear we might have miscommunicated here. I wasn't questioning plotting in log-log scale. Rather, as I understand, you appear to have log transformed your data and then plotted the log-transformed data linearly, rather than plotting your data and transforming the axes logarithmically. In matplotlib pseudo-code, you appear to have plotted:
>
> ```
> log_n = np.log10(n)
> log_sge = np.log10(sge)
> plt.plot(log_n, log_sge)
> ```
>
> rather than plotting
>
> ```
> plt.plot(n, sgd)
> plt.xscale("log")
> plt.yscale("log")
> ```
>
> I feel like the latter is far more common, hence why I suggested it. I did label this point as a nit, so if you feel strongly that the former is preferable and tell me so, that's fine.

---

> ### Comment · Reviewer_1VDq · 2024-08-10
> **Response to Authors' Rebuttal (Part 2)**
>
> > Participation ratio is a linear measure of intrinsic dimension. Maximum Likelihood Estimation (MLE) [4] measures the intrisic dimension for low-dimensional nonlinear models by applying the principle of maximum likelihood to the distances between close neighbors.
>
> This doesn't actually tell me what MLE ID is or why PR is inappropriate here (certainly it's linear, but there's missing next step). However...
>
> > MLE has been commonly used in prior works [1,2] to estimate the intrinsic dimension of natural data and assess impact on downstream model performance.
>
> ... based on this information, MLE ID seems like a more reasonable choice (since I lack knowledge that would favor or disfavor this choice).
>
> > We used only four models because we were already able to extract an empirical scaling law over two orders of model size. However, if the reviewer feels including the full suite would strengthen our experimental results we would be happy to plot more intermediate model sizes.
>
> I strongly suspect that others will ask you this question, so I recommend doing it, but I now better understand that it probably won't matter as much.
>
> > our ablations in Figure 4 are meant to demonstrate the stability of our measurements of intrinsic dimension under various reaslistic model hyperparameters - not to suggest any asymptotic behavior
>
> Thank you for clarifying this point. I suggest perhaps lightly rephrasing "Overall, we find the estimated ID is fairly stable across each factor." to integrate the clarification you shared with me here.
>
> Based on the authors' rebuttal, I will increase my score to a 6 and decrease my confidence to a 1. To justify why:
>
> - I don't feel competent to assess the novelty or significance of the theoretical contributions (results or proof techniques), which is really the heart of this paper
> - the authors addressed my more empirical concerns
>
> **Ask**: If I can make one last request of the authors, I think a "Future Directions" section would be a nice addition to help readers decide which next research problems are worth pursuing.

---

> ### Author Response · Authors · 2024-08-11
> **Rebuttal cont.**
>
> Thank you for reading our rebuttal and taking the time to respond. We address some of your points below.
>
> - After reading your response, I wonder if there might be some miscommunication here. Sharma and Kaplan state that to estimate ID: "we use the activations from the last token in each sequence to measure the ID, though the ID does not vary significantly across token positions (see figure 10)", whereas your lines 219-220 state " we will estimate the intrinsic dimension of the input data by estimating the intrinsic dimension of token embeddings." I interpreted your sentence to mean that you took either the embedding matrix or the unembedding matrix (which may be the same, if you are tying/sharing the matrices). After rereading the paragraph on lines 219 to 230, I think you are using the word "embedding" when Sharma and Kaplan (as well as I) use "activations".
>
>     Could you please clarify what exactly you are doing? Are you using activations or the embedding vectors that comprise the (un)embedding matrix? If you are using activations, this seems far more sensible to me.
>
> Ah sorry about the confusion. We are using activations from the final transformer layer (pre logit transformation). Figure 9 measures the ID resulting from instead using activations from earlier transformer layers.
>
> - I fear we might have miscommunicated here. I wasn't questioning plotting in log-log scale. Rather, as I understand, you appear to have log transformed your data and then plotted the log-transformed data linearly, rather than plotting your data and transforming the axes logarithmically. In matplotlib pseudo-code, you appear to have plotted:
>
>     log_n = np.log10(n)
>     log_sge = np.log10(sge)
>     plt.plot(log_n, log_sge)
>
>     rather than plotting
>
>     plt.plot(n, sgd)
>     plt.xscale("log")
>     plt.yscale("log")
>
>     I feel like the latter is far more common, hence why I suggested it. I did label this point as a nit, so if you feel strongly that the former is preferable and tell me so, that's fine.
>
> Thanks for clarifying! We agree adjusting the plot scale (as you suggest) is cleaner and will update our paper accordingly.
>
> - This doesn't actually tell me what MLE ID is or why PR is inappropriate here (certainly it's linear, but there's missing next step). However [...] based on this information, MLE ID seems like a more reasonable choice (since I lack knowledge that would favor or disfavor this choice).
>
> A non-linear measure of ID is generally preferred since it can capture non-linear structure that a linear measure of ID will miss. See [1,2] for a demonstration why real-world data has a low non-linear ID but high linear ID. As a simple example, consider a three-dimensional (unit) helix. Inspecting the singular values of the helix would suggest ID = 3 when in reality the helix has ID = 1. MLE is simply a standard choice of non-linear ID measure.
>
> - Ask: If I can make one last request of the authors, I think a "Future Directions" section would be a nice addition to help readers decide which next research problems are worth pursuing.
>
> Thank you for the suggestion. We would be happy to include this in a longer version of the paper.
>
> References:
>
> [1] Nonlinear Dimensionality Reduction by Locally Linear Embedding, https://www.science.org/doi/full/10.1126/science.290.5500.2323?casa_token=yfncfe5drp0AAAAA%3AeCd9RAaQNuvRbWlvXsOnGOlpX0BtboQ4U3k-eQYu0oztePn9TZOXPGoPktxvph4GfvA-bmIKMBSO
>
> [2] A Global Geometric Framework for Nonlinear Dimensionality Reduction, https://www.science.org/doi/full/10.1126/science.290.5500.2319?casa_token=Lo9xDQgvfxAAAAAA%3Ap2bdmZCL2lej6ljrkru8QxldbkbpEjDfSWW3w3ix0D8-kjSw-rI68bZWBb_PczxfiifAZT8vdqc2

---

### Official Review · Reviewer_TAmh · 2024-07-19

**Soundness:** 2
**Presentation:** 3
**Contribution:** 2
**Rating:** 6
**Confidence:** 3

**Summary:**

This paper investigates the representational capabilities of transformers in regression tasks and their correlation with scaling laws. The authors present a novel analysis of the transformer's sample complexity on datasets with low intrinsic dimension $d$, or those residing on a $d$-dimensional manifold $\mathcal{M}$. Their findings yield a bound reminiscent of standard non-parametric regression results, with the key distinction that it depends on the intrinsic dimension rather than the actual data dimension.

The analysis hinges on a specific transformer construction that performs approximation on subregions of $\mathcal{M}$. Notably, when the model is sufficiently wide and deep, this construction can approximate the target function to $\epsilon$ precision with a depth independent of $\epsilon$. This characteristic contrasts favorably with feedforward models using ReLU activation, which typically require $O(\epsilon^{-1})$ layers.

Finally, the paper shows that if we estimate the intrinsic dimension of the data, the resulting estimate of the exponent can be informative of that predicted by the sample complexity from the theoretical result.

**Strengths:**

As far as I know, the theoretical result of the paper is novel and highly non-trivial. To a certain degree, it sheds light on the fundamental difference between a transformer and simpler models like the feed-forward model. I foresee these results to be useful for future research into the statistical properties of transformers.

**Weaknesses:**

The theoretical result is novel but I am not sure how insightful the bound of the construction is about what the model actually is doing and how the result is related to scaling law. There are several weaknesses in the paper which I will discuss below:

Despite its novel theoretical contributions, this paper has several significant weaknesses that warrant discussion:

1. **Limited applicability to real-world scenarios**: The theoretical model presented in the paper assumes a regression or supervised learning task, which diverges significantly from how transformer models are typically used in practice. The input representation ($x$ as a $D$-dimensional vector with the sequence being a linear transformation of $x$) also appears overly simplistic and may not adequately capture the complexity of real-world data and tasks. This disconnect raises questions about how well the theoretical results translate to practical applications.

2. **Relevance of the proposed construction**: While the construction presented in the paper is mathematically interesting at a technical level, it's unclear whether it accurately represents the actual learning process of transformer models. The authors do not provide empirical evidence to support that their theoretical construction aligns with the internal representations or mechanisms developed by transformers during training. This lack of validation leaves a significant gap between theory and practice. I would be more than happy to be proven wrong with empirical evidence.

3. **Tenuous connection to scaling laws**: The paper's attempt to link its theoretical results to empirical scaling laws appears weak. The predicted line for $\alpha_D$ in Figure 2 shows a substantial divergence from actual scaling laws for most datasets, with OpenWebText being a notable exception. The claimed error margin of $\pm 0.2$ is exceptionally large on a log-log scale, potentially overstating the accuracy of the predictions. Moreover, the fit for $\alpha_N$ is even less convincing, further undermining the paper's claims about its relevance to scaling laws.

These weaknesses suggest that while the theoretical work presented is novel, its practical implications and connections to real-world transformer behavior may be limited. The paper would benefit from stronger empirical validation and a more thorough exploration of how its theoretical insights relate to the actual functioning of transformer models in practice.

**Questions:**

1. Why is the intrinsic dimension dependent on the model since they are supposed to be intrinsic to the data? How do I know the approximation of it is reasonable? For example, why is "sub-sample final-layer tokens from the embedded subsequence and shuffle together all the embedding" a reasonable thing to do?

2.  How does the computational complexity of the proposed transformer construction compare to that of the feedforward model? If the transformer requires significantly more parameters to achieve the $\epsilon$-independent depth, does this truly represent an advantage, given that other parameters in the bound still depend on $\epsilon$? Can you provide a more comprehensive comparison that takes into account total computational resources, including the number of parameters and operations required? Without such a comparison, it's challenging to conclude whether this result definitively demonstrates the benefit of transformers over MLPs.

**Limitations:**

Some limitations I discussed above are already in the paper but many are not.

---

> ### Author Rebuttal · Authors · 2024-08-06
>
> We thank the reviewer for their thorough response! We address some of your concerns below.
>
> **Strengths**:
> - As far as I know, the theoretical result of the paper is novel and highly non-trivial. To a certain degree, it sheds light on the fundamental difference between a transformer and simpler models like the feed-forward model
>
> We appreciate the reviewer recognizes our theoretical result as novel and higly non-trivial and that it shed light on the fundamental difference between a transformer and simpler models like the feed-forward network. We would like to emphasize that the **main contribution of this paper is theoretical**. This includes our statistical estimation theory in Theorem 1 which  relies upon a novel and non-trivial covering number result for transformers (Lemma 2) and a novel approximation theory in Theorem 2 which is constructed from a number of key sub-lemmas (e.g. see Lemma 3 - Interaction Lemma) allowing for complex interaction of multiple tokens which is also of independent interest. We believe these results constitute a signficant step forward in the theoretical understanding of transformers
>
> **Weaknesses:**
>
> 1a. Limited applicability to real-world scenarios: The theoretical model presented in the paper assumes a regression or supervised learning task, which diverges significantly from how transformer models are typically used in practice... This disconnect raises questions about how well the theoretical results translate to practical applications.
>
> Despite being proved for regression, our result also sheds light on applications using next-token prediction (classification). Theoretically it is known that results on regression problems can be used to estalish results for classification problems [2, Theorem 1.1] since classification essentially depends on estimation of the probability function for classification. This allows our results on regression for transformers to be applied to next-token prediction which translates to practical applictaions. Additionally, we remark the regression objective itself is commonly used in many real-world scenarios e.g. training a transformer diffusion backbone [1]
>
> 1b. The input representation ($x$ as a - $D$ dimensional vector with the sequence being a linear transformation of $x$) also appears overly simplistic and may not adequately capture the complexity of real-world data and tasks
>
> We note that most common real-world tasks including language modeling, which embeds an input $x \in R^{vocab\_size}$ via a linear token embedding matrix, and ViT, which takes an image $x \in R^D$ and embeds it as token *patches* using a (linear) convolution, utilize this embedding strategy.
>
> 2. Relevance of the proposed construction: While the construction presented in the paper is mathematically interesting at a technical level, it's unclear whether it accurately represents the actual learning process of transformer models. The authors do not provide empirical evidence to support that their theoretical construction aligns with the internal representations or mechanisms developed by transformers during training. This lack of validation leaves a significant gap between theory and practice. I would be more than happy to be proven wrong with empirical evidence... This disconnect raises questions about how well the theoretical results translate to practical applications.
>
>
> In machine learning theory, to obtain an empirical minimizer $\hat{T}_n$, we start with an empirical risk function (in our case mean squared error (mse) on training data) and a hypothesis function class $\mathcal{T}$. The empirical risk minimizer (ERM) $\hat{T}_n$ is then the candidate in $\mathcal{T}$ which minimizes the empirical risk. Our goal is to understand how the generalization error of $\hat{T}_n$ depends on the number of samples and the intrinsic dimesion of the data. Note that our statistical learning theory does not make any claim about how $\hat{T}_n$ is achieved (i.e. the optimization process).
> The generalization error can be decomposed into two parts: bias (approximation error) and variance (stochastic error). The approximation error represents the expressivity of the transformer architecture space $\mathcal{T}$ e.g. how well can a function in $\mathcal{T}$ approximate the target $T$. Our approximation theory results are significant because **they allow us to quantitatively and precisely control the approximation error $\epsilon$ as a function of the transformer architecture size**. Importantly, this construction to control the approximation does not say anything about the learned parameters of the empirical risk minimizer or the parameters learned during the optimizaion process. Roughly speaking the discrepancy between the ERM and our construction is controlled by the stochastic error which we bound using the covering number of the transformer hypothesis space. We view this flexiblity as an advantage of our theory, as **we do not require the learned parametrs of the ERM to align with our construction in the approximation theory to control the ERM's generalization error**. We would further remark that this theory **does** translate to practical applications for understanding transformer scaling laws as evidenced by our experimental results on datasets similar to those used in [3].
>
> 3. The predicted line in Figure 2 shows a substantial divergence from actual scaling laws for most datasets, with OpenWebText being a notable exception. The claimed error margin of $\pm 0.2$ is exceptionally large on a log-log scale, potentially overstating the accuracy of the predictions. Moreover, the fit for $\alpha_D$ is even less convincing.
>
> Our margin of error in Figure 2 when predicting $\alpha_D$ is $\pm 0.02$ (an order of magnitude lower than $\pm 0.2$). We believe this is fairly accurate. The prediction of $\alpha_N$ is similarly accurate (up to $\pm 0.03$). Further this is an improvement over other empirically driven works [4] which estimate scaling laws using intrinsic dimension.

---

> ### Author Response · Authors · 2024-08-06
> **Rebuttal cont.**
>
> **Questions:**
>
> 1. Why is the intrinsic dimension dependent on the model since they are supposed to be intrinsic to the data? How do I know the approximation of it is reasonable? For example, why is "sub-sample final-layer tokens from the embedded subsequence and shuffle together all the embedding" a reasonable thing to do?
>
> Approximation of the intrinsic dimension of data via model embeddings is a relatively established practice [4,5,6]. In the case of text we have no other choice as existing ID estimation algorithms are designed for continuous data. However, this inapplicability should not be taken to mean that textual data does not have low-dimensional structure. While using embeddings to measure this low-dimensional structure may introduce some noise, we expect a good model (such as an LLM) to have learned good enough representations to preserve the majority of this structure. We do ablations at the end of the paper to examine how sensitive this estimation is to various hyperparameters and find it is relatively stable. While embeddings from any layer could be chosen, we choose the final layer to stay consistent with prior work [4,5,6].
>
> 2. How does the computational complexity of the proposed transformer construction compare to that of the feedforward model? If the transformer requires significantly more parameters to achieve the $\epsilon$-independent depth, does this truly represent an advantage, given that other parameters in the bound still depend on $\epsilon$? Can you provide a more comprehensive comparison that takes into account total computational resources, including the number of parameters and operations required? Without such a comparison, it's challenging to conclude whether this result definitively demonstrates the benefit of transformers over MLPs.
>
> Via our approximation theory, the number of parameters in a transformer is
>
> \begin{align*}
>     O(L_T\cdot(m\cdot d_{embd}+L_{FFN}\cdot d_{embd})) &= O(\log(d)(\log(\epsilon^{-1})+d\epsilon^{-\frac{d}{\beta}})) \\
>     &= O(\log(d)d\epsilon^{-\frac{d}{\beta}})
> \end{align*}
>
> The number of parameters in a FFN with the same target $\epsilon$ is
> \begin{align*}
>     O(Depth \cdot Width) = O(\log(\epsilon^{-1})
>     \log(d)d\epsilon^{-\frac{d}{\beta}})
> \end{align*} where the depth depends logarithmically on $\epsilon^{-1}$ [7]. So the transformer does indeed only require a factor of $\log(\epsilon^{-1})$ fewer parameters than the FFN [7]. Additionally, most of the parameters in our proposed transformer architecture class come from the attention heads. Computationally speaking, this is desirable as the attention heads can be more easily parallelized than sequential feed-forward layers.
>
> **References:**
>
> [1] Scalable Diffusion Models with Transformers, https://arxiv.org/abs/2212.09748
>
> [2] A Distribution-Free Theory of Nonparametric Regression, https://link.springer.com/book/10.1007/b97848
>
> [3] Scaling Laws for Neural Language Models, https://arxiv.org/abs/2001.08361
>
> [4]  Scaling Laws from the Data Manifold Dimension, https://jmlr.org/papers/v23/20-1111.html
>
> [5] The Shape of Learning: Anisotropy and Intrinsic Dimensions in Transformer-Based Models, https://arxiv.org/abs/2311.05928
>
> [6] An Intrinsic Dimension Perspective of Transformers for Sequential Modeling, https://openreview.net/forum?id=0UzYWLzPBjA
>
> [7] Error bounds for approximations with deep ReLU networks, https://arxiv.org/abs/1610.01145

---

> > ### Comment · Reviewer_TAmh · 2024-08-10
> >
> > Thank you for the detailed response. Some of my questions have been addressed, but many are still unresolved.
> >
> > > We note that most common real-world tasks including language modeling [...]
> >
> > Indeed we use embedding in practice too but in practice, we have a sequence of tokens/inputs and here a *single* input is being embedded into a sequence of latent tokens. In other words, the modeling assumption is that the whole sequence is a linear projection of a low-dimensional input. I don't see an immediate connection between this setting and real data. Could you explain this more?
> >
> > >  Note that our statistical learning theory does not make any claim about how T is achieved [...]
> >
> > Respectfully, I disagree that a good explanation of generalization in deep learning can be independent of the optimization procedure, see [1] for a more detailed discussion. Nonetheless, I think the jury is still out on this question so I will not include this in my final decision. Regarding scaling law, I will elaborate on this in the next point.
> >
> > > Our margin of error in Figure 2 when predicting [...]
> >
> > I apologize for the typo in my review where I meant to type 0.02 instead of 0.2, but the point I believe still stands. Let's take the Bigcode-SQL plot in Figure 2. By eyeballing, I'd say the final prediction is roughly -0.1 (empirical fit) and -0.05 (predicted). This translates to roughly $\exp(-0.05) - \exp(-0.1) = 0.046$ difference in validation loss. This is a huge difference. For a comparison, let's look at llama2 [2]. In figure 5, going from 34B to 70B, the *perplexity* decreased by about less than 0.1, roughly, that's a 0.028 difference in validation loss but represents a huge change in the model performance. As such, I still think the prediction made by the proposed framework is pretty far away from reality. If I misinterpreted the plot in any way, please let me know.
> >
> > > Approximation of the intrinsic dimension of data via model embeddings is a relatively established practice [...]
> >
> > I understand that it is a standard practice but it doesn't make it a reasonable one. You are making a claim about an intrinsic property of the data-generating process yet the actual computation relies on an arbitrarily chosen model. Real data is most likely not generated by a transformer so your measure of intrinsic dimension cannot be correct, so it is not clear to me what I can take away from empirical validation that is based on an erroneous foundation. Note that this is independent of the theoretical framework.
> >
> >
> > **Reference**
> >
> > [1] Fantastic Generalization Measures are Nowhere to be Found. Gastpar et al.
> >
> > [2] llama 2: Open Foundation and Fine-Tuned Chat Model.

---

> > > ### Author Response · Authors · 2024-08-11
> > > **Rebuttal cont.**
> > >
> > > Thank you for taking the time to read our rebuttal and respond. We address some of your points below:
> > >
> > > - Indeed we use embedding in practice too but in practice, we have a sequence of tokens/inputs and here a single input is being embedded into a sequence of latent tokens. In other words, the modeling assumption is that the whole sequence is a linear projection of a low-dimensional input. I don't see an immediate connection between this setting and real data. Could you explain this more?
> > >
> > > The pre-embeded input $x \in \mathbb{R}^D$ can be viewed as a sequence components. This is similar to what vision transformers do: take a vector $x \in \mathbb{R}^D$ as input and embed it as a sequence of pixel tokens. This is the same as our formulation. We would like to clarify that we do **not** assume that the whole sequence is a linear projection of a low-dimensional input. Rather, we apply a linear transformation to **each component** of the input i.e. tokenwise (as is usually done with transformers).
> > >
> > >
> > > - Respectfully, I disagree that a good explanation of generalization in deep learning can be independent of the optimization procedure, see [1] for a more detailed discussion. Nonetheless, I think the jury is still out on this question so I will not include this in my final decision. Regarding scaling law, I will elaborate on this in the next point.
> > >
> > > We agree that an understanding of optimization is essential for a complete understanding of generalization. However, other components (e.g. approximation and statistics) are equally necessary and useful for explaining certain **components** of the generalization error: namely those coming from the bias and variance. This is exactly what our work (and more generally the entre field of non-parametric statistics) does. These error components **directly result** in the model and data scaling laws observed empirically when other sources of error are small. Further, our bound is mini-max optimal (up to logarithmic factors). Using these results, which importantly depend on the intrinsic dimension of the data, we are able to theoretically predict empirical scaling laws more accurately than any other (theoretical works) we are aware of. We would again like to note that this theory is the **main contribution** of our work and therefore should be considered in its evaluation.
> > >
> > > In regards to [1], we agree that it is important to consider the algorithm used in generalization error analysis. As far we as we know, the convergence landscape of a multi-layer transformer network is a widely open-question in of itself. However, it does not negate the importance of the theory we contribute.

---

> > > > ### Author Response · Authors · 2024-08-11
> > > > **Rebuttal cont.**
> > > >
> > > > - I apologize for the typo in my review where I meant to type 0.02 instead of 0.2, but the point I believe still stands. Let's take the Bigcode-SQL plot in Figure 2. By eyeballing, I'd say the final prediction is roughly -0.1 (empirical fit) and -0.05 (predicted). This translates to roughly .046 difference in validation loss. This is a huge difference. For a comparison, let's look at llama2 [2]. In figure 5, going from 34B to 70B, the perplexity decreased by about less than 0.1, roughly, that's a 0.028 difference in validation loss but represents a huge change in the model performance. As such, I still think the prediction made by the proposed framework is pretty far away from reality. If I misinterpreted the plot in any way, please let me know.
> > > >
> > > >
> > > >
> > > > In Figure 2 we are plotting the data scaling exponent $\alpha_D$ (i.e. how does loss decrease as samples increase) for relatively small models (~125M parameters). However, in your example you are comparing these **data scaling** results to Llama's **model scaling** results going from 34B to 70B. This does not allow for a direct comparison. For models in such a large regime the loss differential will naturally be much smaller. It may be the competing loss terms are uniformly small enough that our theoretical framework, as currently applied, is less useful in this setting. However, in the smaller (but equally valid) regime of 125M models, where loss can range from 2.5 - 4.0, we consider an absolute error of $\pm 0.05$ to be accurate and compelling evidence that our theory is able to produce predictions reflecting deep learning phenomena in practice.
> > > >
> > > > We additionally remark that although our theory is not as precisely accurate as empirical measurements, it is more accurate than any other theoretical predictions we are aware of. This is because we importantly leverage the intrinsic dimension of the training distribution which is known to have a strong impact on model performance [5]. We reiterate that our experiments are not meant to suggest replacing current empirical predictions with theoretical ones but rather to validate the correctness of our theory which is the **core** contribution of this paper. Further, the experiments demonstrate for text a recurring observation made in other modalities: intrinsic dimension of data strongly impacts the learning process. We are able to rigorously quantify this impact with our theory.
> > > >
> > > > - I understand that it [our esimtate of ID] is a standard practice but it doesn't make it a reasonable one. You are making a claim about an intrinsic property of the data-generating process yet the actual computation relies on an arbitrarily chosen model. Real data is most likely not generated by a transformer so your measure of intrinsic dimension cannot be correct, so it is not clear to me what I can take away from empirical validation that is based on an erroneous foundation. Note that this is independent of the theoretical framework.
> > > >
> > > > We note the model used to measure ID is **not** arbitrarily chosen: it is trained on the distribution of data we aim to estimate the ID of. In general, the practice of training a model to learn a good representation in order to measure quantities of interest in data is extremely common (see e.g. [3,4,5]). Although the approximation offered may have some noise, often it is still useful in making predictions. Empirically, this has been demonstrated many times [4,5,6,7]. For these reasons, we (as well as many others) believe this method of measuring ID to be an entirely reasonable choice.
> > > >
> > > > Reference
> > > >
> > > > [1] Fantastic Generalization Measures are Nowhere to be Found. Gastpar et al.
> > > >
> > > > [2] llama 2: Open Foundation and Fine-Tuned Chat Model.
> > > >
> > > > [3] DoReMi: Optimizing Data Mixtures Speeds Up Language Model Pretraining, https://arxiv.org/abs/2305.10429
> > > >
> > > > [4]  Scaling Laws from the Data Manifold Dimension, https://jmlr.org/papers/v23/20-1111.html
> > > >
> > > > [5] The Intrinsic Dimension of Images and Its Impact on Learning, https://arxiv.org/abs/2104.08894
> > > >
> > > > [6] The Shape of Learning: Anisotropy and Intrinsic Dimensions in Transformer-Based Models, https://arxiv.org/abs/2311.05928
> > > >
> > > > [7] An Intrinsic Dimension Perspective of Transformers for Sequential Modeling, https://openreview.net/forum?id=0UzYWLzPBjA

---

> > > > > ### Comment · Reviewer_TAmh · 2024-08-12
> > > > >
> > > > > Thank you for the reply.
> > > > >
> > > > > > The pre-embeded input $x \in \mathbb{R}^D$ can be viewed as a sequence components [...]
> > > > >
> > > > > Thank you for the clarification. So if I understand correctly, there is no sequential aspect to the data under this model? Wouldn't this make it even less realistic?
> > > > >
> > > > > > This does not allow for a direct comparison [...]
> > > > >
> > > > > I am focusing on the final loss value, which I believe is comparable since the downstream performances are largely a function of the final loss according to the prevailing scaling hypothesis. What I am concerned about is that the prediction is systematically off, so it is hard for it to convince me that the theory *predicts* what happens in reality.
> > > > >
> > > > > >  it is trained on the distribution of data we aim to estimate the ID of [...]
> > > > >
> > > > > But if you train different models I assume you'd get different measurements of ID, so it's clearly not just noise. But this is a minor point due to lack of better alternative, so I would not include this in my final decision.

---

> > > > > > ### Author Response · Authors · 2024-08-12
> > > > > > **Rebuttal cont.**
> > > > > >
> > > > > > We thank the reviewer for their response.
> > > > > >
> > > > > > - Thank you for the clarification. So if I understand correctly, there is no sequential aspect to the data under this model? Wouldn't this make it even less realistic?
> > > > > >
> > > > > > Data with an explicit sequential structure is included in our formulation, making our formulation more general. Additionally, we remark transformers are often applied in practice to data which is not inherently sequential (such as images).
> > > > > >
> > > > > > - I am focusing on the final loss value, which I believe is comparable since the downstream performances are largely a function of the final loss according to the prevailing scaling hypothesis. What I am concerned about is that the prediction is systematically off, so it is hard for it to convince me that the theory predicts what happens in reality.
> > > > > >
> > > > > > To reiterate our previous point, we are simply saying that comparing the raw loss differential between 125M parameter models to the raw loss differential of 34B - 70B parameter models is not a fair comparison. The loss differential for 125M models will be much larger, so one should not judge the accuracy of our theory in this way.
> > > > > >
> > > > > > We admit that our predictions at this larger scale of model (34B - 70B) may not be as relatively precise as they are for smaller models. However, we are more accurate for smaller models ($\pm 0.05$ loss differential), and believe that this accuracy meaningfully demonstrates the practical correctness of our theory (when compared to other theoretical predictions).
> > > > > >
> > > > > > Additionally, you say you are concerned our predictions are systemically off. Do you mind being more specific? Do you believe there is a significant source of error even in the smaller model regime?
> > > > > >
> > > > > > - But if you train different models I assume you'd get different measurements of ID, so it's clearly not just noise. But this is a minor point due to lack of better alternative, so I would not include this in my final decision.
> > > > > >
> > > > > > In Figure 4 we demonstrate the estimated ID is stable across a number of model hyperparameters (model depth, embedding dimension, size, and context length) with the ID changing by at most $\pm 5$ (with the ambient dimension being as large as **1536**). Changing our scaling predictions to use another of the estimated IDs would result in at most $\pm 0.03$ additional error.

---

> > > > > > > ### Comment · Reviewer_TAmh · 2024-08-12
> > > > > > >
> > > > > > > > Data with an explicit sequential structure is included in our formulation,
> > > > > > >
> > > > > > > How do you incorporate sequential structure? Isn't $x\in \mathbb{R}^D$ is just a single vector.
> > > > > > >
> > > > > > > > we are more accurate for smaller models [...]
> > > > > > >
> > > > > > > It is ok that a theoretical framework is not accurate but the claim you are making in the paper is "predicting" and "explaining". These are tall orders and I don't think the empirical results sufficiently support your claims. By systematically off I just mean that they are already diverging somewhat significantly for SQL and Tiny story whereas the deviation in Openwebtext is much more reasonable. Scaling law is only interesting precisely because one can extrapolate from it and I don't think we can extrapolate from the theoretical prediction.
> > > > > > >
> > > > > > > > In Figure 4 we demonstrate the estimated ID is stable across a number of model hyperparameter [...]
> > > > > > >
> > > > > > > I agree that in Figure 4 the measurement seems somewhat stable but they are all measured by transformers (so are your references I believe), but my point is that how can one be sure that it is actually the intrinsic dimension of the data? If I use an RNN to measure the intrinsic dimension would I still get the same quantities? I think what this is actually measuring is some notion of simplicity as measured by *transformers trained by SGD*. In any case, this is somewhat of a philosophical question so I won't include it in my decision.

---

> ### Author Response · Authors · 2024-08-13
> **Rebuttal cont.**
>
> - How do you incorporate sequential structure?
>
> A vector $x \in \mathbb{R}^D$ is very general. Each component (or sub-sequence of components) could be taken to be time-series measurements or a sequence of pixels (as is done in ViT). We then use a position encoding to embed this sequential structure.
>
> - It is ok that a theoretical framework is not accurate but the claim you are making in the paper is "predicting" and "explaining". These are tall orders and I don't think the empirical results sufficiently support your claims. By systematically off I just mean that they are already diverging somewhat significantly for SQL and Tiny story whereas the deviation in Openwebtext is much more reasonable. Scaling law is only interesting precisely because one can extrapolate from it and I don't think we can extrapolate from the theoretical prediction.
>
> If you strongly object to our characterization of theory as "predicting" scaling laws we would be willing to use a softer characterization (perhaps "estimating"?) in an updated version of the paper.

---

> > ### Comment · Reviewer_TAmh · 2024-08-13
> >
> > >  Each component (or sub-sequence of components) could be taken to be time-series measurements or a sequence of pixels [...]
> >
> > I see. That is an interesting interpretation. I did not get this picture from the paper. Apologies if I missed it but if I didn't miss it, I think it would be good to highlight it.
> >
> > > our characterization of theory as "predicting" scaling laws [...]
> >
> > I think "estimating" would be better to use or at least greatly highlight that the prediction made is better but still far from reality so intrinsic dimension is only a part of a puzzle. I guess the bottom line is it's clear that deep learning can only work if the data are "low-dimension" in nature so it's natural that the same would apply to transformer. It is not surprising that things are a straight line on a log-log curve since even PAC bounds would look like a straight line on the log-log curve so how well the slope matches is the only thing that matters for the prediction. In any case, I think a large portion of my questions and concerns have been addressed and I think the theoretical construction is quite interesting and worthy of follow-up so I am raising my score to 6. Thank you for the discussion.

---

> > > ### Author Response · Authors · 2024-08-13
> > > **Rebuttal cont.**
> > >
> > > We thank the reviewer improving their score and for the helpful discussion!

---

### Decision · Program_Chairs · 2024-09-25

**Decision:**

Accept (poster)

**Comment:**

This paper develops a theoretical analysis of scaling laws for transforms for intrinsically low-dimensional data and validates the theory via empirical observation of training LLMS on text data. The topic is salient and will be of interest to the community. The reviewers found the work novel and highly non-trivial, opening a window into the differences between transformers and simpler architectures. I recommend acceptance.